# Distillation of Large Language Models via Concrete Score Matching

**Yeongmin Kim**[1], **Donghyeok Shin**[1], **Mina Kang**[1], **Byeonghu Na**[1], **Il-Chul Moon**[1,2]

[1]Korea Advanced Institute of Science and Technology (KAIST), [2]summary.ai

{alsdudrla10,tlsehdgur0,kasong13,byeonghu.na,icmoon}@kaist.ac.kr

## Abstract

Large language models (LLMs) deliver remarkable performance but are costly to deploy, motivating knowledge distillation (KD) for efficient inference. Existing KD objectives typically match student and teacher probabilities via softmax, which blurs valuable logit information. While direct logit distillation (DLD) mitigates softmax smoothing, it fails to account for logit shift invariance, thereby restricting the solution space. We propose *Concrete Score Distillation* (CSD), a discrete score-matching objective that overcomes both softmax-induced smoothing and restrictions on the optimal solution set. We resolve the training instability and quadratic complexity of discrete score-matching in autoregressive LLMs, and the resulting CSD objective aligns relative logit differences across all vocabulary pairs between student and teacher with flexible weighting. We provide both mode-seeking and mode-covering instances within our framework and evaluate CSD on task-agnostic instruction-following, task-specific, and general chat capability distillation using GPT-2-1.5B, OpenLLaMA-7B, Gemma-7B-IT, Qwen2.5-7B-IT, and Gemma2-9B-IT teachers. Experiments show that CSD consistently surpasses recent KD objectives, achieves favorable fidelity–diversity trade-offs, demonstrating its scalability and effectiveness for LLM distillation. Code: https://github.com/aailab-kaist/CSD.

## 1 Introduction

Large language models (LLMs) have demonstrated remarkable generative capabilities across a wide range of tasks (Achiam et al., 2023; Dubey et al., 2024; Liu et al., 2024; Comanici et al., 2025). Such progress has been primarily driven by the vast amount of training data and the unprecedented scale of model parameters (Kaplan et al., 2020). However, when deploying such LLMs in real-world applications, the recurring inference cost becomes prohibitively expensive. Consequently, research into reducing the parameter size of LLMs while preserving performance has become particularly crucial for enabling efficient inference. In this context, knowledge distillation (KD) (Hinton et al., 2015) has emerged as a promising approach for LLMs, as it allows a smaller student model to inherit the capabilities of a large teacher model, thereby enabling more efficient inference.

The common paradigm in KD for LLMs is to align the per-token probability distributions of the student with those of the teacher. Kullback–Leibler (KL) divergence was initially the most widely adopted objective, and the search for more effective probability matching losses has since become a central topic of research. Alternative objectives have been proposed within the framework of $f$-divergence (Wen et al., 2023; Gu et al., 2024; Agarwal et al., 2024), as well as its smoothed variants (Ko et al., 2024; Shing et al., 2025; Ko et al., 2025). However, existing distillation losses primarily targeted the estimated probabilities obtained through the softmax transformation, instead of directly utilizing the raw neural network outputs (logits) from either the teacher or the student. As illustrated in Figure 1b, even when the teacher's logit values differ substantially, their corresponding probability values can be nearly indistinguishable. Such smoothing hinders the student from faithfully capturing the teacher's knowledge, a challenge further exacerbated in modern LLMs with large vocabularies, where most tokens are assigned near-zero probabilities (See Figure 1a).

In traditional KD, direct logit distillation (DLD) (Ba & Caruana, 2014; Urban et al., 2017) has been proposed as an alternative strategy, with advantages in generalization capability and in removing the

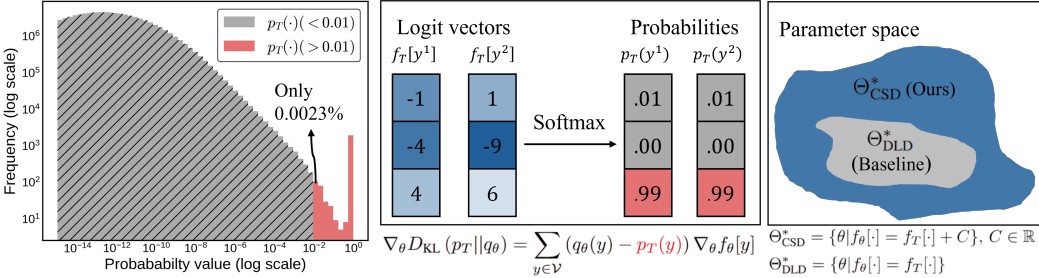

(a) Probability value statistics.    (b) Softmax hides teacher's knowledge.    (c) Optimal solution sets.

Figure 1: Motivation for logit-level distillation and limitations of prior work. (a) Statistics of per-token probabilities for every vocabulary for 16 input–output sequences from the teacher model (`GPT-2-1.5B`). The probabilities are highly sparse, with only 0.0023% being greater than 0.01. (b) Despite large differences in logits (e.g., $[-1, -4, 4]$ vs. $[1, -9, 6]$), softmax yields nearly identical probabilities and gradients. (c) Prior direct logit distillation restricts the solution set.

softmax smoothing (Kim et al., 2021). However, such approaches have not been thoroughly explored in the context of LLMs. This paper identifies a key drawback of DLD: its restriction on the optimal solution set as described in Figure 1c. Considering the softmax activation in inference, it is sufficient for the teacher's and student's logits to agree up to an additive constant, but the previous solutions of DLD fail to accommodate such an acceptable slack constant, a.k.a. *logit shift invariance*. Such a restriction on the solution set may hinder the discovery of optimal solutions in distillation, particularly when the teacher and student models have a large capacity gap, as is often the case with LLMs. Therefore, the goal of this paper is to establish a design space of distillation losses that overcome both the softmax-induced smoothing of teacher knowledge and the restriction on the solution set.

This paper adopts the idea from energy-based models (Song & Kingma, 2021), which design objectives that avoid the constraint of probabilistic models (sum-to-one) by using the score-matching objective (Hyvärinen & Dayan, 2005). We propose *Concrete Score Distillation* (CSD), a discrete form of the score-matching objective (Meng et al., 2022) adapted for autoregressive LLM distillation. We address training instability and computational overhead arising when applying the score-matching objective to LLMs, and provide theoretical guarantees of optimality, showing that its solution set is broader than that of DLD from both theoretical and empirical perspectives. The resulting objective reduces to matching the relative logit differences across all pairs of vocabulary items between the student and teacher, while allowing flexible weighting across all vocabulary pairs in linear time with respect to vocabulary size. Furthermore, we present instances within our framework that exhibit both mode-seeking and mode-covering properties.

In our experiments, we conducted task-agnostic instruction-following distillation, task-specific distillations (summarization, mathematics, and translation), and general chat capability distillation using GPT-2 (Radford et al., 2019), OpenLLaMA (Geng & Liu, 2023), Gemma (Team et al., 2024a), Qwen2.5 (Team et al., 2024b), and Gemma2 (Team et al., 2024a) backbones. The proposed CSD consistently outperformed recent probability-matching objectives as well as direct logit distillation. By appropriately choosing weighting functions, we further demonstrated that our method resides on the frontier of the diversity–fidelity trade-off. Finally, we observed complementary performance gains when integrating our loss with on-policy techniques.

## 2 PRELIMINARIES

### 2.1 KNOWLEDGE DISTILLATION OF LARGE LANGUAGE MODELS

We consider autoregressive large language models (LLMs), consisting of a teacher $p_T$ and a student $q_\theta$ with $\theta \in \Theta$, where the student is a smaller and more efficient model. Given an input context $\mathbf{c}$, the student generates an output sequence $\mathbf{y} = (y_1, y_2, \ldots, y_L)$ with probability $q_\theta(\mathbf{y}|\mathbf{c}) = \prod_{t=1}^{L} q_\theta(y_t|\mathbf{c}, \mathbf{y}_{<t})$, where $L$ denotes the sequence length, and the teacher's probability is defined analogously. Each token $y_t$ is drawn from the fixed vocabulary set $\mathcal{V} := \{v_1, v_2, \ldots\}$. As in prior works (Lin et al., 2020; Ko et al., 2024), we assume the teacher and student share the same vocabulary set. To compute the token probability $q_\theta(y_t|\mathbf{c}, \mathbf{y}_{<t})$, an LLM typically adopts a parametric function $f_\theta : \mathcal{V}^{|\mathbf{c}|} \times \mathcal{V}^{t-1} \to \mathbb{R}^{|\mathcal{V}|}$, which maps the input $(\mathbf{c}, \mathbf{y}_{<t})$ to a logit vector $f_\theta(\mathbf{c}, \mathbf{y}_{<t}) \in \mathbb{R}^{|\mathcal{V}|}$. The

logit corresponding to token $y_t$ is denoted by $f_\theta(\mathbf{c}, \mathbf{y}_{<t})[y_t]$. For brevity of notation, the input arguments of the function $f_\theta$ will be omitted hereafter. Let $f_T$ be the parametric function of the teacher. Accordingly, the probability of each token is calculated through the softmax transformation:

$$q_\theta(y_t|\mathbf{c}, \mathbf{y}_{<t}) = \frac{\exp(f_\theta[y_t])}{\sum_{x \in \mathcal{V}} \exp(f_\theta[x]))}, \qquad p_T(y_t|\mathbf{c}, \mathbf{y}_{<t}) = \frac{\exp(f_T[y_t])}{\sum_{x \in \mathcal{V}} \exp(f_T[x]))}. \tag{1}$$

**Problem definition:** The goal of knowledge distillation for LLMs is to align the student's per-token probability distribution with that of the teacher, so that the student inherits the teacher's capabilities. We assume access to input–output sequence pairs $(\mathbf{c}, \mathbf{y}) \sim \mathcal{D}$, obtained either from a fixed dataset or from samples generated by the student or teacher (Lin et al., 2020; Ko et al., 2024). For each selected instance $(\mathbf{c}, \mathbf{y})$, distillation is performed by selecting a specific discrepancy metric $D$ and minimizing the discrepancy between the per-token probability distributions with respect to $\theta$:

$$\mathbb{E}_{(\mathbf{c}, \mathbf{y}) \sim \mathcal{D}} \left[ \frac{1}{L} \sum_{t=1}^{L} D\left( p_T\left(\cdot|\mathbf{c}, \mathbf{y}_{<t}\right) || q_\theta\left(\cdot|\mathbf{c}, \mathbf{y}_{<t}\right)\right) \right]. \tag{2}$$

**Prior work and motivation**: In previous studies, $D$ is most commonly chosen as the KL divergence (Hinton et al., 2015), which is formulated as follows (the input of the probability is omitted):

$$D_{\mathrm{KL}}\left(p_T || q_\theta\right) = \sum_{y_t \in \mathcal{V}} p_T(y_t|\mathbf{c}, \mathbf{y}_{<t}) \log \frac{p_T(y_t|\mathbf{c}, \mathbf{y}_{<t})}{q_\theta(y_t|\mathbf{c}, \mathbf{y}_{<t})}. \tag{3}$$

However, $D_{\mathrm{KL}}$ focuses on the teacher's probabilities and is constrained by the softmax. As shown in Figure 1b, although the teacher carries rich knowledge across all vocabulary items at the logit level, much of it is lost after softmax, and the teacher provides nearly identical gradient signals to most minor tokens. Accordingly, in classical KD studies (Ba & Caruana, 2014; Urban et al., 2017), *direct logit distillation* (DLD) has been widely adopted as a logit-level mean squared error (MSE) loss:

$$\mathcal{L}_{\mathrm{DLD}}\left(\theta; p_T, w\right) = \frac{1}{2} \sum_{y_t \in \mathcal{V}} w(y_t) \left(f_\theta[y_t] - f_T[y_t]\right)^2, \tag{4}$$

where $w(\cdot)$ is a strictly positive weighting function[1]. Kim et al. (2021) showed that $\mathcal{L}_{\mathrm{DLD}}$ provides better generalization and representation capability by taking minority indices into account. Since faithfully distilling logit information is crucial for large-vocabulary LLMs, we investigated the use of DLD for LLM distillation. However, we found that its optimal solution does not permit logit constant invariance, thereby severely restricting the solution set. This observation motivated us to develop a logit-level distillation loss that does not restrict the optimal solution.

## 2.2 SCORE MATCHING FOR A DISCRETE RANDOM VARIABLE

Score-matching (SM) (Hyvärinen & Dayan, 2005) was originally proposed in energy-based models (Song & Kingma, 2021) with continuous variables $\mathbf{x} \in \mathbb{R}^d$. An energy function $E_\theta : \mathbb{R}^d \to \mathbb{R}$ maps $\mathbf{x}$ to a scalar. The corresponding probability and the score-matching objective are given by:

$$q_\theta(\mathbf{x}) = \frac{\exp(-E_\theta(\mathbf{x}))}{Z_\theta}, \quad \mathcal{L}_{\mathrm{SM}}(\theta; p_{\mathrm{data}}, w) = \mathbb{E}_{w(\mathbf{x})} \left[||\nabla_{\mathbf{x}} \log q_\theta(\mathbf{x}) - \nabla_{\mathbf{x}} \log p_{\mathrm{data}}(\mathbf{x})||_2^2\right], \tag{5}$$

where $p_{\mathrm{data}}$ is the data distribution, $Z_\theta = \int_{\mathbf{x}} \exp(-E_\theta(\mathbf{x}))\mathrm{d}\mathbf{x}$ is the partition function, and $w(\cdot)$ is a weighting function. The term $\nabla_{\mathbf{x}} \log q_\theta(\mathbf{x}) = -\nabla_{\mathbf{x}} E_\theta(\mathbf{x})$ is known as the *Stein score*, which uniquely identifies the probability distribution without requiring the computation of $Z_\theta$. $\mathcal{L}_{\mathrm{SM}}$ facilitates the design of losses without considering the normalization constraint of probabilistic models. The probability computation $q_\theta$ here follows, analogously, the form of the LLM probabilities in Eq. (1). The difference is that an LLM outputs energy values $f_\theta$ over all finite states at once, whereas an EBM handles continuous variables, so that each input to $E_\theta$ yields only a single scalar output.

Inspired by how EBMs design losses beyond the normalized structure of a probabilistic model through score-matching, we extend this idea to construct logit-level distillation losses for LLMs. However, because the Stein score is defined through derivatives, it cannot be directly applied to discrete random

---

[1]Throughout this paper, we assume each weighting function sums to one over the vocabulary for simplicity.

variables. Meng et al. (2022) proposed a generalized score function, applicable to both continuous and discrete variables, named the *concrete score*: $s_\theta(y) := \left[\frac{q_\theta(x)}{q_\theta(y)}\right]_{x \in \mathcal{V}}$. Similar to the Stein score, the concrete score characterizes local changes at the current state, but replaces them with probability ratios between all other point masses. This term is also uniquely identifiable with the underlying distribution. The corresponding concrete score-matching objective is then defined as:

$$\mathcal{L}_{\text{CSM}}(\theta; p_{\text{data}}, w) = \frac{1}{2}\left[\sum_{y \in \mathcal{V}}\sum_{x \in \mathcal{V}} w(y, x)\left(\frac{q_\theta(x)}{q_\theta(y)} - \frac{p_{\text{data}}(x)}{p_{\text{data}}(y)}\right)^2\right], \tag{6}$$

where $p_{\text{data}}$ is the data distribution defined over a discrete state, and $w(\cdot, \cdot)$ is a positive weighting function. Previous work on language models (Lou et al., 2024) typically adopted this loss by directly parameterizing the concrete score (also known as discrete diffusion models) to mimic the data distribution. In contrast, we take this concept as a starting point to design logit-level distillation losses for autoregressive-type language models, which are more dominant in real-world applications.

## 3 METHOD

This section introduces the proposed *Concrete Score Distillation* (CSD) objective for knowledge distillation (KD) in autoregressive large language models (LLMs). Section 3.1 discusses the challenges of directly applying $\mathcal{L}_{\text{CSM}}$ to LLMs, so we propose a modified objective with theoretical guarantees of optimality and compare the objective with $\mathcal{L}_{\text{DLD}}$. Section 3.2 presents an efficient analytic gradient computation for CSD, analyzes its gradient structure, and compares it with that of $D_{\text{KL}}$.

### 3.1 CONCRETE SCORE DISTILLATION FOR LARGE LANGUAGE MODELS

**Tackling training instability:** We observe that optimizing the student model $q_\theta$ by minimizing $\mathcal{L}_{\text{CSM}}(\theta; p_T, w)$ leads to training instability, as the likelihood ratio $\frac{q_\theta(x)}{q_\theta(y_t)}$ can diverge as the denominator approaches zero. In the discrete diffusion model (Lou et al., 2024), a single vocabulary item is fed into the neural network $s_\theta$, which directly outputs the ratios over the other vocabulary items, thereby avoiding instability. In contrast, autoregressive LLMs compute probabilities for each vocabulary item separately and then take their ratios, making this issue specific to autoregressive LLMs.

Training instability is a well-known issue in likelihood ratio estimation (Rhodes et al., 2020). Following Higuchi & Suzuki (2025), we address it by applying a monotonically increasing function to the concrete scores. In particular, we adopt the logarithm, which yields the following objective:

$$\mathcal{L}_{\text{CSD}}(\theta; p_T, w) := \frac{1}{2}\left[\sum_{y_t \in \mathcal{V}}\sum_{x \in \mathcal{V}} w(y_t, x)\left(\log\frac{q_\theta(x|\mathbf{c}, \mathbf{y}_{<t})}{q_\theta(y_t|\mathbf{c}, \mathbf{y}_{<t})} - \log\frac{p_T(x|\mathbf{c}, \mathbf{y}_{<t})}{p_T(y_t|\mathbf{c}, \mathbf{y}_{<t})}\right)^2\right] \tag{7}$$

$$= \frac{1}{2}\sum_{y_t \in \mathcal{V}}\sum_{x \in \mathcal{V}} w(y_t, x)\left(f_\theta[x] - f_\theta[y_t] - f_T[x] + f_T[y_t]\right)^2. \tag{8}$$

The choice of the logarithm function provides two benefits: (1) it yields an MSE loss between logits (i.e., neural network outputs), ensuring stability by avoiding the likelihood ratio computation; and (2) it naturally leads to the logit-level loss design, which aligns with our motivation.

**Logit distillation with intra-vocabulary relationships:**
Unlike $\mathcal{L}_{\text{DLD}}$, which directly matches student and teacher logits for the same vocabulary item, $\mathcal{L}_{\text{CSD}}$ aligns the logit residuals across different vocabulary items between the student and the teacher. This allows the student not only to be compared against the teacher but also to perform relative comparisons among its own vocabulary items. In contrast to $D_{\text{KL}}$, where softmax normalization implicitly adjusts each vocabulary item relative to all others, our loss explicitly controls the pairwise relationships between student vocabulary items $y_t$ and $x$ through the weighting function $w(y_t, x)$. Figure 2 illustrates how a logit vector $f_\theta(\mathbf{c}, \mathbf{y}_{<t}) \in \mathbb{R}^{|\mathcal{V}|}$ (e.g., [1, 4,

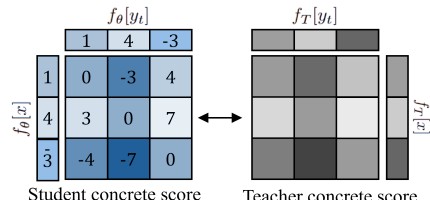

Figure 2: Schematic for $\mathcal{L}_{\text{CSD}}$ (Eq. (8)).

---

**Algorithm 1:** Gradient computation of *Concrete Score Distillation*

---

**Input:** Student $f_\theta$, teacher $f_T$, prompt $\mathbf{c}$, prefix $\mathbf{y}_{<t}$, function $w(\cdot,\cdot) = w_1(\cdot)w_2(\cdot)$.

1  Compute the student logit $f_\theta[y_t] = f_\theta(\mathbf{c}, \mathbf{y}_{<t})[y_t], \forall y_t \in \mathcal{V}$.

2  **with** *no_grad:*

3     Compute the teacher logit $f_T[y_t] = f_T(\mathbf{c}, \mathbf{y}_{<t})[y_t], \forall y_t \in \mathcal{V}$.

4     Compute the weighted average logits:

5     $\bar{f}_\theta^{w_1} = \sum_{y_t \in \mathcal{V}}[w_1(y_t) \times f_\theta[y_t].\texttt{detach}], \bar{f}_\theta^{w_2} = \sum_{y_t \in \mathcal{V}}[w_2(y_t) \times f_\theta[y_t].\texttt{detach}]$

6     $\bar{f}_T^{w_1} = \sum_{y_t \in \mathcal{V}}[w_1(y_t) \times f_T[y_t]], \bar{f}_T^{w_2} = \sum_{y_t \in \mathcal{V}}[w_2(y_t) \times f_T[y_t]]$

7     Compute the weighted normalized logits:

8     $\tilde{f}_\theta^{w_1}[y_t] = f_\theta[y_t] - \bar{f}_\theta^{w_1}, \tilde{f}_\theta^{w_2}[y_t] = f_\theta[y_t] - \bar{f}_\theta^{w_2}, \forall y_t \in \mathcal{V}$.

9     $\tilde{f}_T^{w_1}[y_t] = f_T[y_t] - \bar{f}_T^{w_1}, \tilde{f}_T^{w_2}[y_t] = f_T[y_t] - \bar{f}_T^{w_2}, \forall y_t \in \mathcal{V}$.

10    $w_{\text{grad}}(y_t) = \left[ w_1(y_t) \left[ \tilde{f}_\theta^{w_2}[y_t] - \tilde{f}_T^{w_2}[y_t] \right] + w_2(y_t) \left[ \tilde{f}_\theta^{w_1}[y_t] - \tilde{f}_T^{w_1}[y_t] \right] \right], \forall y_t \in \mathcal{V}$

11 $\nabla_\theta \mathcal{L}_{\text{CSD}}(\theta; p_T, w) = \sum_{y_t \in \mathcal{V}} [w_{\text{grad}}(y_t) \nabla_\theta f_\theta[y_t]]$

12 **return** $\nabla_\theta \mathcal{L}_{\text{CSD}}(\theta; p_T, w)$

---

–3]) produces a concrete score and how it is matched with the teacher's concrete score. The following theorems provide the theoretical guarantee of the proposed objective function.

**Proposition 1.** *(Consistency) Given context* $\mathbf{c}$ *and prefix* $\mathbf{y}_{<t}$*, assume model capacity* $|\Theta| \to \infty$*. For any* $w(\cdot, \cdot) > 0$*, define the set of optimal parameters as* $\Theta_{CSD}^* = \arg\min_{\theta \in \Theta} \mathcal{L}_{CSD}(\theta; p_T, w)$*. Then, for any* $\theta^* \in \Theta_{CSD}^*$*, we have* $\mathcal{L}_{CSD}(\theta^*; p_T, w) = 0$*, and the following holds for all* $y_t \in \mathcal{V}$*:*

$$q_{\theta^*}(y_t | \mathbf{c}, \mathbf{y}_{<t}) = p_T(y_t | \mathbf{c}, \mathbf{y}_{<t}).$$

Please refer to Section A.1 for the proof. Proposition 1 shows that consistency holds when matching the log-transformed concrete scores of the student and teacher, and guarantees that our objective leads the student to converge to the target teacher.

**Theorem 2.** *(Solution Superset) Assume model capacity* $|\Theta| \to \infty$*, let the set of optimal parameters* $\Theta_{CSD}^* = \arg\min_{\theta \in \Theta} \mathcal{L}_{CSD}(\theta; p_T, w)$ *and* $\Theta_{DLD}^* = \arg\min_{\theta \in \Theta} \mathcal{L}_{DLD}(\theta; p_T, w)$*, then following holds:*

$$\Theta_{CSD}^* \supsetneq \Theta_{DLD}^*.$$

Please see Section A.2 for the proof. Theorem 2 implies that all solutions obtainable by $\mathcal{L}_{\text{DLD}}$ can also be recovered by $\mathcal{L}_{\text{CSD}}$. This is because $\mathcal{L}_{\text{CSD}}$ is invariant to constant shifts in logits; for example, when $f_\theta[y_t] = f_T[y_t] + C$ for all $y_t \in \mathcal{V}$, the probabilities are identical and the $\mathcal{L}_{\text{CSD}}$ is zero, whereas the $\mathcal{L}_{\text{DLD}}$ is not optimal. This advantage could be pronounced under limited model capacity, where the larger solution set of $\mathcal{L}_{\text{CSD}}$ enables more faithful approximation of the teacher's knowledge.

## 3.2 GRADIENT COMPUTATION AND ANALYSIS

The remaining challenge of the proposed objective $\mathcal{L}_{\text{CSD}}$ in Eq. (8) lies in its computational cost of $\mathcal{O}(|\mathcal{V}|^2)$. Unlike $D_{\text{KL}}$ and $D_{\text{DLD}}$, $D_{\text{CSD}}$ requires a double summation over the vocabulary set $\mathcal{V}$. This formulation is infeasible to implement in standard computational environments due to memory constraints. Nevertheless, we show that the gradient of this objective can be computed in linear time:

**Theorem 3.** *(Efficient Gradient Computation) Assume* $w(y_t, x) = w_1(y_t)w_2(x)$*, then the gradient of* $\mathcal{L}_{CSD}(\theta; p_T, w)$ *with respect to* $\theta$ *could be computed in* $\mathcal{O}(|\mathcal{V}|)$ *as:*

$$\nabla_\theta \mathcal{L}_{CSD}(\theta; p_T, w) = \sum_{y_t \in \mathcal{V}} \mathbf{w}(y_t)^T \left( \tilde{\mathbf{f}}_\theta[y_t] - \tilde{\mathbf{f}}_T[y_t] \right) \nabla_\theta f_\theta[y_t], \tag{9}$$

*where* $\mathbf{w}(y_t) = (w_1(y_t), w_2(y_t))^T$, $\tilde{\mathbf{f}}_\theta[y_t] = \left( \tilde{f}_\theta^{w_2}[y_t], \tilde{f}_\theta^{w_1}[y_t] \right)^T$, $\tilde{\mathbf{f}}_T[y_t] = \left( \tilde{f}_T^{w_2}[y_t], \tilde{f}_T^{w_1}[y_t] \right)^T$, *with* $\tilde{f}_\theta^w[y_t] = f_\theta[y_t] - \mathbb{E}_{w(x)}[f_\theta[x]]$, $\tilde{f}_T^w[y_t] = f_T[y_t] - \mathbb{E}_{w(x)}[f_T[x]]$ *are normalized logits.*

The proof is provided in Section A.3. For the actual training time and memory usage, please refer to Table 9 in Section D. These results follow from factorizing the independent variables. Algorithm 1

further details the gradient computation of Eq. (9) step by step, with each step requiring only linear time over the vocabulary. An alternative approach is to use Monte Carlo estimation as described in Algorithm 2 of Section C. Instead of taking a weighted sum over all possible states of $y_t$ with $w_1$, one can draw a single sample of $y_t$ according to probability $w_1$ and compute the loss in expectation. The Monte Carlo estimation, unlike the analytic gradient form, does not require assuming independence between the two variables of $w$, allowing it to model the joint weighting function space directly. However, defining a joint weighting function over two discrete vocabulary spaces is generally difficult. We found that independent weighting functions capture various behaviors in Section 4.4. In these cases, Monte Carlo estimation increases the variance within batched samples, which slightly slows the convergence compared to the analytic computation (see Figure 5c).

**Gradient analysis:** The gradient of $\mathcal{L}_{\mathrm{CSD}}$ in Eq. (9) has a structure similar to that of $D_{\mathrm{KL}}$. For intuitive understanding, let us consider the case where the weighting function of CSD is the uniform distribution $U$. Then, the gradient of each loss becomes:

$$\nabla_\theta D_{\mathrm{KL}}\left(p_T||q_\theta\right) = \sum_{y_t \in \mathcal{V}} \left( \underbrace{\frac{\exp(f_\theta[y_t])}{\sum_{x\in\mathcal{V}} \exp(f_\theta[x])}}_{\text{normalized student logit}} - \underbrace{\frac{\exp(f_T[y_t])}{\sum_{x\in\mathcal{V}} \exp(f_T[x])}}_{\text{normalized teacher logit}} \right) \nabla_\theta f_\theta[y_t],$$

$$\nabla_\theta \mathcal{L}_{\mathrm{CSD}}\left(\theta; p_T, U\right) = \sum_{y_t \in \mathcal{V}} \frac{2}{|\mathcal{V}|} \left( \underbrace{\left( f_\theta[y_t] - \frac{\sum_{x\in\mathcal{V}} f_\theta[x]}{|\mathcal{V}|} \right)}_{\text{normalized student logit}} - \underbrace{\left( f_T[y_t] - \frac{\sum_{x\in\mathcal{V}} f_T[x]}{|\mathcal{V}|} \right)}_{\text{normalized teacher logit}} \right) \nabla_\theta f_\theta[y_t].$$

In gradient descent, both losses decrease the student's logit $f_\theta[y_t]$ where the student's normalized logits are large, and increase $f_\theta[y_t]$ where the teacher's normalized logits are large. The only difference lies in how the logit coefficients are normalized over the vocabulary set: $D_{\mathrm{KL}}$ inherits the softmax form, which, as noted in Figure 1b, poses a major problem for transferring the teacher's knowledge. In contrast, our $\mathcal{L}_{\mathrm{CSD}}$ uses centering normalization, allowing the student to directly capture the teacher's logit information. Moving beyond the uniform weighting case study, the formulation in Eq. (9) further provides a design space for logit normalization through $(w_1, w_2)$, where $w_1$ controls the weighting of vocabulary tokens during gradient updates and $w_2$ governs coefficient normalization, with their roles applied again in reverse order $(w_2, w_1)$.

## 4 EXPERIMENTS

This section comprehensively validates the effectiveness of the proposed *Concrete Score Distillation* (CSD) across various experimental setups. Section 4.1 shows results on task-agnostic instruction-following distillation, comparing CSD with alternative loss functions and assessing its performance when combined with on-policy methods. Section 4.2 further examines task-specific settings, including math, summarization, and translation, to evaluate the applicability of CSD. Section 4.3 evaluates scalability by examining whether general conversational abilities can also be distilled. Finally, Section 4.4 establishes the contribution of each component in CSD through ablation studies.

### 4.1 TASK-AGNOSTIC INSTRUCTION-FOLLOWING DISTILLATION

We follow the training setup of DistiLLM (Ko et al., 2024). For the distillation dataset $\mathcal{D}$, we use `databricks-dolly-15k` (Conover et al., 2023). We first fine-tune the teacher on this dataset and then distill it into student models. We use the detached student probability as the default choice for both $w_1$ and $w_2$, and we apply it similarly to the weights in DLD. Please refer to Section C.1 for further details on backbone, training configuration, baseline, and the evaluation protocol.

**Loss-level comparison:** To purely analyze the effect of the distillation loss itself, this comparison excludes the use of pretraining losses, initialization with an SFT-tuned student, and any on-policy techniques. Table 1 shows that the proposed CSD objective outperforms the other nine objectives, ranking first on three of the five benchmarks, second on one, and achieving the highest average score. SKL (Ko et al., 2024) and AB (Wang et al., 2025) exhibit slightly lower performance than previously reported, likely due to their reliance on pretraining losses or on-policy techniques. Figure 3a shows the fidelity–diversity trade-off based on ROUGE-L and Self-BLEU scores. Traditionally, KL favors diversity, whereas RKL favors mode-seeking. Within this trade-off, SKL, SRKL, TV, and AB

Table 1: Comparison of loss functions for distilling `GPT-2-1.5B` into `GPT-2-0.1B`. Every result is from our implementation with the same teacher, purely using the distillation objective. ROUGE-L scores were averaged over five random seeds; best scores are **boldfaced**, second-best underlined.

| Loss | Dolly Eval | Self-Instruct | Vicuna Eval | Super-NI | UnNI | Avg. (↑) |
|---|---|---|---|---|---|---|
| Teacher | $27.00_{\pm 0.19}$ | $14.07_{\pm 0.37}$ | $16.31_{\pm 0.32}$ | $26.46_{\pm 0.41}$ | $31.10_{\pm 0.06}$ | 22.99 |
| KL | $23.52_{\pm 0.25}$ | $10.02_{\pm 0.58}$ | $14.57_{\pm 0.32}$ | $16.76_{\pm 0.17}$ | $18.55_{\pm 0.13}$ | 16.68 |
| RKL (Gu et al., 2024) | $24.26_{\pm 0.11}$ | $11.19_{\pm 0.17}$ | $15.80_{\pm 0.26}$ | $20.17_{\pm 0.15}$ | $22.99_{\pm 0.14}$ | 18.88 |
| Sym-KL | $23.29_{\pm 0.20}$ | $10.24_{\pm 0.31}$ | $15.25_{\pm 0.43}$ | $17.46_{\pm 0.11}$ | $20.60_{\pm 0.08}$ | 17.37 |
| Jeffrey | $23.00_{\pm 0.38}$ | $10.82_{\pm 0.44}$ | $15.00_{\pm 0.50}$ | $18.19_{\pm 0.11}$ | $20.07_{\pm 0.11}$ | 17.42 |
| TV (Wen et al., 2023) | $23.88_{\pm 0.30}$ | $11.03_{\pm 0.51}$ | $15.13_{\pm 0.44}$ | $\underline{24.58}_{\pm 0.25}$ | $25.24_{\pm 0.06}$ | 19.97 |
| GJS (0.9) (Agarwal et al., 2024) | $24.10_{\pm 0.24}$ | $11.40_{\pm 0.39}$ | $\mathbf{16.02}_{\pm 0.57}$ | $20.28_{\pm 0.13}$ | $22.55_{\pm 0.12}$ | 18.87 |
| SKL (0.1) (Ko et al., 2024) | $24.17_{\pm 0.24}$ | $11.21_{\pm 0.53}$ | $15.29_{\pm 0.24}$ | $22.65_{\pm 0.14}$ | $24.69_{\pm 0.11}$ | 19.60 |
| SRKL (0.1) (Ko et al., 2024) | $\underline{24.53}_{\pm 0.21}$ | $\mathbf{12.19}_{\pm 0.29}$ | $15.63_{\pm 0.22}$ | $23.37_{\pm 0.27}$ | $24.28_{\pm 0.18}$ | $\underline{20.00}$ |
| AB (0.2, 0.7) (Wang et al., 2025) | $24.20_{\pm 0.12}$ | $11.82_{\pm 0.29}$ | $\underline{15.87}_{\pm 0.36}$ | $21.44_{\pm 0.20}$ | $25.59_{\pm 0.09}$ | 19.78 |
| CSD (Ours) | $\mathbf{24.94}_{\pm 0.29}$ | $\underline{12.06}_{\pm 0.46}$ | $15.78_{\pm 0.49}$ | $\mathbf{24.60}_{\pm 0.31}$ | $\mathbf{25.88}_{\pm 0.13}$ | $\mathbf{20.65}$ |

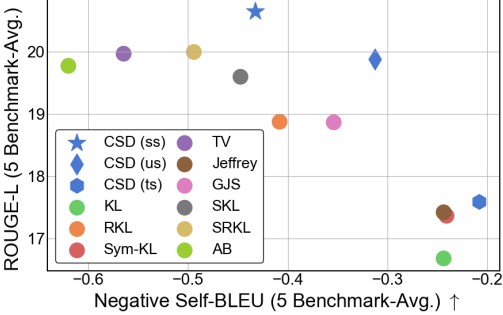

(a) Fidelity vs. Diversity trade-off.

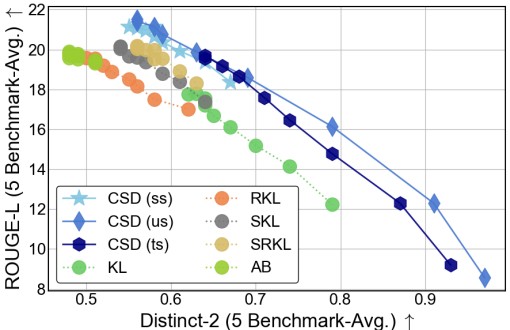

(b) Decoding temperature ablation in $[0.2, 1.8]$.

Figure 3: An in-depth analysis of the distributional behavior of different loss functions.

achieve higher ROUGE-L scores than RKL, but at the cost of reduced diversity, reflecting a stronger emphasis on fidelity. Diversity, however, remains an important aspect of user experience in instruction-following, and it becomes a valuable metric as it enhances performance when combined with best-of-N sampling. The proposed CSD provides an additional lever to control the fidelity–diversity trade-off. By default, using the detached student probabilities $(S, S)$ yields the highest fidelity. Replacing one side with uniform $(U, S)$ or with the teacher $(T, S)$ gradually increases diversity. This is likely because the $(S, S)$ makes the model focus only on regions where the student already assigns a high likelihood, limiting its exploratory ability. The trade-off offered by CSD envelopes those of existing losses, and we expect that even better operating points may exist within the design space of $w_1$ and $w_2$. Figure 3b presents an ablation on temperature, which enables easy adjustment of the trade-off during inference. Even within a reasonable range of decoding temperature, CSD achieves better trade-off points than other losses. In particular, CSD $(U, S)$ demonstrates a well-balanced exchange between diversity and fidelity through temperature adjustment.

**Orthogonal improvement with recent on-policy advances:** Table 2 reports the performance of recent distillation baselines augmented with the CSD loss, demonstrating its orthogonal applicability. We applied the CSD $(S, S)$ and DLD $(S)$ loss to ImitKD (Lin et al., 2020), GKD (Agarwal et al., 2024), and DistiLLM (Ko et al., 2024). DLD-mean refers to the mean-centered DLD variant, as described in Section A.4. The primary distinction among these methods, apart from their losses, lies in the choice of dataset $\mathcal{D}$: ImitKD uses purely student-generated on-policy data, GKD combines fixed data with student outputs, and DistiLLM adaptively selects between them based on validation loss. As a result, the average ROUGE-L score improved for both `GPT-2-0.1B` and `GPT-2-0.3B` students in all settings, compared to both the baseline and the corresponding DLD versions. The best result on each benchmark was also achieved by our method, with particularly strong performance under pure on-policy settings. We also evaluated using GPT-4 as the judge in Figure 4, where our best model was judged superior to other baselines. There may exist CSD variants other than $(S, S)$ that perform better for specific $\mathcal{D}$, but we leave this exploration to future work. Finally, applying our best setting to a larger `OpenLLaMA` also outperformed baselines, demonstrating the scalability of CSD with respect to model size. We provide comparisons with more baselines in Table 7 of Section D.

Table 2: Instruction-following performance of CSD with on-policy techniques for various backbones. $\mathcal{D}$ denotes the distillation dataset. ROUGE-L scores are averaged over five random seeds, with the best score for each student highlighted in **bold**. CSD and DLD use the student probability weighting.

| Method | Loss | $\mathcal{D}$ | Dolly Eval | Self-Instruct | Vicuna Eval | Super-NI | UnNI | Avg. (↑) |
|---|---|---|---|---|---|---|---|---|
| Teacher (`GPT-2-1.5B`) | | | $27.00_{\pm 0.19}$ | $14.07_{\pm 0.37}$ | $16.31_{\pm 0.32}$ | $26.46_{\pm 0.41}$ | $31.10_{\pm 0.06}$ | 22.99 |
| Teacher (`OpenLLaMA-7B`) | | | $27.60_{\pm 0.34}$ | $18.17_{\pm 0.80}$ | $17.85_{\pm 0.48}$ | $31.05_{\pm 0.31}$ | $32.40_{\pm 0.28}$ | 25.41 |
| `GPT-2-1.5B → GPT-2-0.1B` | | | | | | | | |
| GKD (Agarwal et al., 2024) | GJS | Mix | $22.48_{\pm 0.20}$ | $10.08_{\pm 0.67}$ | $15.61_{\pm 0.08}$ | $13.88_{\pm 0.21}$ | $16.59_{\pm 0.13}$ | 15.73 |
| DistiLLM (Ko et al., 2024) | SKL | Ada | $25.28_{\pm 0.28}$ | $12.04_{\pm 0.49}$ | $16.66_{\pm 0.34}$ | $22.13_{\pm 0.31}$ | $24.32_{\pm 0.14}$ | 20.09 |
| ImitKD (Lin et al., 2020) | KL | On | $21.79_{\pm 0.18}$ | $10.25_{\pm 0.37}$ | $14.65_{\pm 0.62}$ | $17.35_{\pm 0.12}$ | $19.43_{\pm 0.13}$ | 16.69 |
| GKD + DLD | DLD | Mix | $25.29_{\pm 0.50}$ | $\mathbf{12.51}_{\pm 0.62}$ | $16.59_{\pm 0.28}$ | $20.87_{\pm 0.37}$ | $22.63_{\pm 0.14}$ | 19.58 |
| DistiLLM + DLD | DLD | Ada | $24.23_{\pm 0.23}$ | $11.86_{\pm 0.51}$ | $\mathbf{17.69}_{\pm 0.18}$ | $19.60_{\pm 0.13}$ | $22.77_{\pm 0.22}$ | 19.23 |
| ImitKD + DLD | DLD | On | $24.69_{\pm 0.28}$ | $12.10_{\pm 0.40}$ | $16.77_{\pm 0.55}$ | $21.58_{\pm 0.36}$ | $23.93_{\pm 0.08}$ | 19.81 |
| GKD + Ours | CSD | Mix | $25.50_{\pm 0.34}$ | $12.03_{\pm 0.65}$ | $16.65_{\pm 0.45}$ | $21.39_{\pm 0.14}$ | $23.48_{\pm 0.03}$ | 19.81 |
| DistiLLM + Ours | CSD | Ada | $25.34_{\pm 0.27}$ | $11.93_{\pm 0.36}$ | $16.99_{\pm 0.29}$ | $\mathbf{22.96}_{\pm 0.24}$ | $24.72_{\pm 0.09}$ | 20.39 |
| ImitKD + Ours | CSD | On | $\mathbf{25.70}_{\pm 0.23}$ | $12.40_{\pm 0.48}$ | $17.18_{\pm 0.52}$ | $22.91_{\pm 0.46}$ | $\mathbf{25.47}_{\pm 0.17}$ | **20.73** |
| `GPT-2-1.5B → GPT-2-0.3B` | | | | | | | | |
| GKD (Agarwal et al., 2024) | GJS | Mix | $25.15_{\pm 0.41}$ | $11.22_{\pm 0.33}$ | $16.45_{\pm 0.48}$ | $17.35_{\pm 0.29}$ | $22.25_{\pm 0.05}$ | 18.48 |
| DistiLLM (Ko et al., 2024) | SRKL | Ada | $26.92_{\pm 0.23}$ | $13.75_{\pm 0.29}$ | $16.90_{\pm 0.25}$ | $26.12_{\pm 0.27}$ | $29.65_{\pm 0.14}$ | 22.67 |
| ImitKD (Lin et al., 2020) | KL | On | $23.61_{\pm 0.34}$ | $12.37_{\pm 0.26}$ | $15.53_{\pm 0.27}$ | $20.20_{\pm 0.20}$ | $24.42_{\pm 0.29}$ | 19.23 |
| GKD + DLD | DLD | Mix | $26.06_{\pm 0.31}$ | $13.51_{\pm 0.35}$ | $16.94_{\pm 0.24}$ | $23.91_{\pm 0.50}$ | $27.33_{\pm 0.09}$ | 21.55 |
| DistiLLM + DLD | DLD | Ada | $25.43_{\pm 0.37}$ | $12.64_{\pm 0.40}$ | $16.91_{\pm 0.61}$ | $22.69_{\pm 0.24}$ | $25.60_{\pm 0.10}$ | 20.65 |
| ImitKD + DLD | DLD | On | $25.82_{\pm 0.37}$ | $13.64_{\pm 0.33}$ | $\mathbf{17.55}_{\pm 0.16}$ | $25.51_{\pm 0.21}$ | $29.07_{\pm 0.08}$ | 22.32 |
| GKD + Ours | CSD | Mix | $27.11_{\pm 0.42}$ | $13.71_{\pm 0.45}$ | $16.98_{\pm 0.29}$ | $25.49_{\pm 0.35}$ | $30.16_{\pm 0.13}$ | 22.69 |
| DistiLLM + Ours | CSD | Ada | $26.77_{\pm 0.18}$ | $13.96_{\pm 0.62}$ | $17.05_{\pm 0.34}$ | $\mathbf{26.29}_{\pm 0.08}$ | $29.56_{\pm 0.09}$ | 22.72 |
| ImitKD + Ours | CSD | On | $\mathbf{27.14}_{\pm 0.28}$ | $\mathbf{14.85}_{\pm 0.66}$ | $16.88_{\pm 0.18}$ | $26.28_{\pm 0.21}$ | $\mathbf{30.43}_{\pm 0.04}$ | **23.12** |
| `OpenLLaMA-7B → OpenLLaMA-3B` | | | | | | | | |
| TAID (Shing et al., 2025) | tKL | Ada | $26.53_{\pm 0.23}$ | $17.73_{\pm 0.69}$ | $18.14_{\pm 0.39}$ | $31.93_{\pm 0.23}$ | $31.55_{\pm 0.12}$ | 25.18 |
| DistiLLM (Ko et al., 2024) | SKL | Ada | $28.63_{\pm 0.28}$ | $20.20_{\pm 0.66}$ | $19.15_{\pm 0.32}$ | $35.31_{\pm 0.19}$ | $34.74_{\pm 0.10}$ | 27.61 |
| DistiLLM (Ko et al., 2024) | SRKL | Ada | $28.83_{\pm 0.41}$ | $20.76_{\pm 0.37}$ | $19.37_{\pm 0.15}$ | $\mathbf{36.82}_{\pm 0.14}$ | $35.76_{\pm 0.13}$ | 28.31 |
| ImitKD + DLD | DLD | On | $29.07_{\pm 0.43}$ | $20.07_{\pm 0.60}$ | $20.05_{\pm 0.37}$ | $36.30_{\pm 0.41}$ | $35.71_{\pm 0.14}$ | 28.24 |
| ImitKD + DLD-mean | DLD | On | $28.13_{\pm 0.36}$ | $19.91_{\pm 0.50}$ | $19.58_{\pm 0.55}$ | $35.85_{\pm 0.50}$ | $35.49_{\pm 0.12}$ | 27.79 |
| ImitKD + Ours | CSD | On | $\mathbf{29.63}_{\pm 0.40}$ | $\mathbf{21.81}_{\pm 0.47}$ | $\mathbf{20.37}_{\pm 0.51}$ | $36.49_{\pm 0.13}$ | $\mathbf{36.86}_{\pm 0.10}$ | **29.03** |

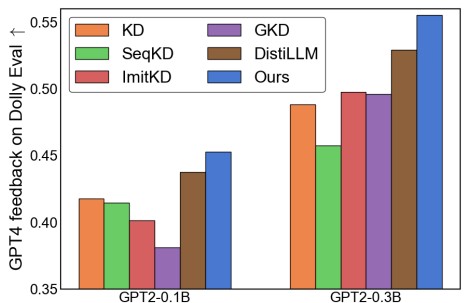

Figure 4: GPT-4 feedback performance, showing the proportion of responses judged correct relative to the golden answers. The teacher's score is 0.61.

Table 3: Task-specific distillation from `Gemma-7B-IT` to `Gemma-2B-IT`.

| | Summarization | Translation | GSM8K |
|---|---|---|---|
| Loss | ROUGE-L | COMET | Accuracy |
| Teacher | 37.09 | 79.23 | 60.27 |
| KL | 35.02 | 73.96 | 24.03 |
| JS | 35.60 | 74.05 | 23.73 |
| TV | 27.49 | 73.73 | 0.00 |
| Jeffrey | 35.29 | 74.02 | 23.28 |
| SKL | 25.86 | 59.65 | 0.00 |
| SRKL | 26.68 | 73.10 | 0.00 |
| RKL | 0.00 | 45.02 | 0.00 |
| DLD $(S)$ | 0.00 | 21.52 | 0.00 |
| DLD-max $(T)$ | 32.54 | 65.28 | 17.74 |
| CSD $(T, S)$ | **35.67** | **74.14** | **25.78** |

## 4.2 TASK-SPECIFIC DISTILLATION

We evaluate the effectiveness of CSD across dialogue summarization, low-resource translation, and arithmetic reasoning tasks. Distillation was conducted on 1,000 samples from the DialogSum (Chen et al., 2021), Flores-200 (Costa-Jussà et al., 2022), and GSM8K (Cobbe et al., 2021) datasets, following the experimental setup of Xu et al. (2025) (Please refer to Section C.2 for further details). Table 3 compares performance across the three tasks against baseline loss functions. Under identical experimental conditions, the proposed CSD objective achieved the best results on all tasks. For the arithmetic reasoning task, we observed several cases in which certain losses yielded zero accuracy. A

Table 4: Benchmarking result for general chat capability. The best score is highlighted in **bold**.

| Benchmark | Qwen2.5-7B-IT → Qwen2.5-1.5B-IT | | | Gemma2-9B-IT → Gemma2-2B-IT | | |
|---|---|---|---|---|---|---|
| | MT-Bench (0-10) | | AlpacaEval (WR) | MT-Bench (0-10) | | AlpacaEval (WR) |
| Judge | GPT4 | GPT4-Turbo | GPT4-Turbo | GPT4 | GPT4-Turbo | GPT4-Turbo |
| Teacher | 8.59 | 7.52 | 88.69 | 8.91 | 7.66 | 94.60 |
| DPKD (Li et al., 2024) | 1.04 | 1.09 | 0.32 | 6.30 | 4.89 | 71.18 |
| DistiLLM-2 (Ko et al., 2025) | 7.28 | 5.75 | **70.42** | 7.81 | 6.45 | 89.91 |
| DLD ($T$) | 7.25 | 5.56 | 69.80 | 5.85 | 4.45 | 31.24 |
| DLD ($S$) | 7.28 | 5.74 | 67.67 | 7.58 | 6.53 | 89.84 |
| CSD ($T, S$) | 7.42 | 5.90 | **70.42** | **7.85** | **6.55** | 89.92 |
| CSD ($S, S$) | **7.69** | **5.95** | 69.64 | 7.77 | 6.43 | **90.05** |

case study in Tables 11 to 15 of Section D shows that these models often produce excessively long chains of thought without arriving at a final answer, indicating a failure to learn proper formatting; furthermore, much of the reasoning itself is incorrect. As illustrated in Figure 3a, the RKL, TV, SKL, and SRKL losses exhibit mode-seeking tendencies, which we conjecture may have caused collapses into suboptimal modes under these limited data distillation settings. Similarly, CSD $(S, S)$, which also shows mode-seeking behavior, performed poorly as 21.09, 63.78, and 0.00 on summarization, translation, and GSM8K, respectively. CSD achieved stable performance in this case by using the $(T, S)$ weighting. DLD likewise performed poorly under student weighting and became only marginally stable when teacher weighting was applied. A full ablation is provided in Table 10 of Section D for DLD variants. Across all cases, DLD remained significantly weaker than CSD, likely due to its restricted solution space being more detrimental under limited-data distillation.

## 4.3 General Chat Capability Distillation

To evaluate general chat performance, we conducted distillation experiments using the latest instruction-tuned models, Qwen2.5-Instruct (Team et al., 2024b) and Gemma2-Instruct (Team et al., 2024a). We followed DistiLLM-2 (Ko et al., 2025) and performed distillation using the 50K subset of the UltraChat dataset (Ding et al., 2023) (Please refer to Section C.3 for further details). Table 4 shows that CSD outperformed both DistiLLM-2 and DLD on MT-Bench (Zheng et al., 2023) and AlpacaEval (Li et al., 2023) win rate (against `text-davinci-003`), demonstrating superior performance. These results further demonstrate the scalability of CSD.

## 4.4 Ablation Studies

**CSD vs DLD:** Table 5 presents comprehensive ablation studies on the weighting function choices in CSD and compares them with direct logit distillation. DLD accommodates only a single weighting function. When comparing DLD with $T$, $U$, and $S$ against CSD with $(T, T)$, $(U, U)$, and $(S, S)$, respectively, our method consistently achieved higher average scores. As shown in Theorem 2, we hypothesize that the broader solution space positively contributed to this improvement. We also compared CSD against other shift-aware DLD variants such as DLD-min and DLD-max, as well as ranking-matching variants including DLD-std (Sun et al., 2024) and DLD-range. However, none of these methods outperformed the naive DLD baseline. Figure 5a shows the effect of temperature scaling on the weighting function of DLD (s). Under the same temperature scaling, CSD consistently outperformed DLD. Figure 9 in Section D shows that DLD restricts solutions to those with a residual constant of zero; CSD adapts residual constants per token, providing evidence that it explores a broader solution set.

**Design choice of CSD:** CSD provides a more flexible loss design space through two weighting functions. While $(S, S)$ provides high-fidelity generation, $(U, S)$ and $(T, S)$ have its own benefits. As illustrated in Figure 3a, replacing $(S, S)$ with $(U, S)$ or $(T, S)$ reduces ROUGE-L but increases diversity, highlighting that CSD can adapt to tasks requiring either mode-covering or mode-seeking properties. As shown in Figure 13 in Section D, the $(U, S)$ weighting substantially reduces the gradient concentration on a small subset of vocabulary tokens that is induced by softmax. This allows all vocabulary items to be learned more evenly, which in turn produces the pattern in Figure 9c where the logits residuals are tightly centered around their offsets. When minority-vocabulary logits are well learned, performance improves noticeably in situations where their contribution

Table 5: Ablation on the logit-level loss design space using `GPT-2-0.1B` student. $T$, $U$, and $S$ denote teacher, uniform, and detached student probabilities. ROUGE-L scores are averaged over five seeds; best scores are in **bold**. Please refer to Section A.4 for DLD variants details.

| Loss | $w_1(\cdot)$ | $w_2(\cdot)$ | Dolly Eval | Self-Instruct | Vicuna Eval | Super-NI | UnNI | Avg. ($\uparrow$) |
|---|---|---|---|---|---|---|---|---|
| DLD | $T$ | - | $0.09_{\pm 0.02}$ | $0.07_{\pm 0.02}$ | $0.18_{\pm 0.03}$ | $0.07_{\pm 0.01}$ | $0.06_{\pm 0.00}$ | 0.09 |
| | $U$ | - | $11.25_{\pm 0.30}$ | $5.55_{\pm 0.63}$ | $9.10_{\pm 0.27}$ | $9.02_{\pm 0.14}$ | $8.24_{\pm 0.07}$ | 8.63 |
| | $S$ | - | $24.22_{\pm 0.24}$ | $12.01_{\pm 0.40}$ | $15.42_{\pm 0.31}$ | $\mathbf{25.44}_{\pm 0.19}$ | $24.88_{\pm 0.19}$ | 20.39 |
| DLD-min | $T$ | - | $1.14_{\pm 0.01}$ | $0.93_{\pm 0.05}$ | $1.65_{\pm 0.13}$ | $0.85_{\pm 0.01}$ | $0.87_{\pm 0.01}$ | 1.09 |
| | $U$ | - | $7.16_{\pm 0.13}$ | $4.53_{\pm 0.20}$ | $7.49_{\pm 0.19}$ | $7.20_{\pm 0.04}$ | $5.97_{\pm 0.05}$ | 6.47 |
| | $S$ | - | $23.89_{\pm 0.38}$ | $11.11_{\pm 0.17}$ | $15.43_{\pm 0.36}$ | $23.78_{\pm 0.24}$ | $25.87_{\pm 0.11}$ | 20.02 |
| DLD-max | $T$ | - | $0.39_{\pm 0.02}$ | $0.32_{\pm 0.03}$ | $0.73_{\pm 0.05}$ | $0.22_{\pm 0.01}$ | $0.21_{\pm 0.01}$ | 0.37 |
| | $U$ | - | $6.47_{\pm 0.06}$ | $4.81_{\pm 0.07}$ | $6.67_{\pm 0.24}$ | $6.80_{\pm 0.06}$ | $5.17_{\pm 0.01}$ | 5.98 |
| | $S$ | - | $9.65_{\pm 0.28}$ | $5.81_{\pm 0.11}$ | $8.66_{\pm 0.43}$ | $11.73_{\pm 0.24}$ | $11.62_{\pm 0.12}$ | 9.49 |
| DLD-std | $T$ | - | $5.98_{\pm 0.18}$ | $4.80_{\pm 0.14}$ | $7.98_{\pm 0.15}$ | $4.93_{\pm 0.06}$ | $4.85_{\pm 0.05}$ | 5.71 |
| | $U$ | - | $21.45_{\pm 0.29}$ | $10.55_{\pm 0.59}$ | $\mathbf{15.90}_{\pm 0.17}$ | $18.85_{\pm 0.26}$ | $20.82_{\pm 0.09}$ | 17.51 |
| | $S$ | - | $9.74_{\pm 0.22}$ | $5.67_{\pm 0.11}$ | $12.29_{\pm 0.27}$ | $7.07_{\pm 0.11}$ | $6.97_{\pm 0.07}$ | 8.35 |
| DLD-range | $T$ | - | $0.85_{\pm 0.03}$ | $0.73_{\pm 0.04}$ | $1.44_{\pm 0.06}$ | $0.53_{\pm 0.02}$ | $0.51_{\pm 0.00}$ | 0.81 |
| | $U$ | - | $10.84_{\pm 0.12}$ | $7.49_{\pm 0.14}$ | $12.82_{\pm 0.30}$ | $9.97_{\pm 0.04}$ | $7.89_{\pm 0.01}$ | 9.80 |
| | $S$ | - | $8.90_{\pm 0.17}$ | $4.74_{\pm 0.15}$ | $7.70_{\pm 0.56}$ | $8.70_{\pm 0.07}$ | $8.90_{\pm 0.04}$ | 7.79 |
| CSD (Ours) | $T$ | $T$ | $6.82_{\pm 0.16}$ | $4.24_{\pm 0.12}$ | $9.16_{\pm 0.25}$ | $4.53_{\pm 0.02}$ | $4.83_{\pm 0.02}$ | 5.91 |
| | $U$ | $U$ | $17.21_{\pm 0.30}$ | $8.08_{\pm 0.39}$ | $14.27_{\pm 0.40}$ | $13.19_{\pm 0.27}$ | $14.07_{\pm 0.04}$ | 13.37 |
| | $S$ | $S$ | $\mathbf{24.94}_{\pm 0.29}$ | $12.06_{\pm 0.46}$ | $15.78_{\pm 0.49}$ | $24.60_{\pm 0.31}$ | $\mathbf{25.88}_{\pm 0.13}$ | **20.65** |
| | $U$ | $S$ | $24.15_{\pm 0.55}$ | $\mathbf{12.25}_{\pm 0.47}$ | $15.25_{\pm 0.41}$ | $22.55_{\pm 0.09}$ | $25.19_{\pm 0.12}$ | 19.88 |
| | $T$ | $S$ | $22.77_{\pm 0.25}$ | $10.62_{\pm 0.32}$ | $14.06_{\pm 0.25}$ | $18.81_{\pm 0.40}$ | $21.71_{\pm 0.18}$ | 17.59 |

(a) Weight scaling  (b) Calibration error  (c) Training curve

Figure 5: Ablation studies for logit-level distillation loss design space.

becomes more significant, such as under high-temperature sampling. This explains why $(U, S)$ performs exceptionally well in the high-diversity region of Figure 3b. Figure 5b measures probability calibration performance. While $(S, S)$ provides high-fidelity generation, it becomes overconfident. Therefore, in scenarios where probability calibration is preferred, we recommend using the $(T, S)$ weighting. Better calibration can potentially improve training stability in small-data settings where the optimization landscape is more difficult, as in Section 4.2. Figure 5c compares the method to resolve $\mathcal{O}(|\mathcal{V}|^2)$ computational cost: 1) using analytic gradient computation from Theorem 3 and 2) Monte Carlo sampling. In both cases, the performance is far superior to KL, but the analytic CSD shows slightly faster training and better convergence. Thus, we recommend using the analytic gradient whenever possible. However, the Monte Carlo variant allows extensions such as joint weighting functions or using Lp loss instead of L2, making it a viable option for more complex modeling.

## 5 CONCLUSION

We introduced *Concrete Score Distillation* (CSD), a novel design space for distillation losses in large language models. CSD simultaneously addresses the challenges of softmax-induced smoothing and restrictions on the optimal solution set, which prior methods have failed to resolve together. Within this framework, we presented instances of both mode-covering and mode-seeking, and demonstrated scalability by consistently surpassing prior work across diverse tasks and model backbones up to 7B parameters. We anticipate that even better instances can be discovered within the proposed design space, particularly by refining $w_1$ and $w_2$ and adapting them to the type of data (fixed or on-policy). This points to promising directions for future exploration of improved instances.

ACKNOWLEDGMENTS

This work was supported by the IITP (Institute of Information & Communications Technology Planning & Evaluation)-ITRC (Information Technology Research Center) grant funded by the Korea government (Ministry of Science and ICT) (IITP-2026-RS-2024-00437268). (50 %) This research was supported by AI Technology Development for Commonsense Extraction, Reasoning, and Inference from Heterogeneous Data(IITP) funded by the Ministry of Science and ICT(RS-2022-II220077) (50 %)

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

# A PROOFS AND DERIVATIONS

## A.1 PROOF OF PROPOSITION 1

**Proposition 1.** *(Consistency) Given context* $\mathbf{c}$ *and prefix* $\mathbf{y}_{<t}$*, assume model capacity* $|\Theta| \to \infty$*. For any* $w(\cdot, \cdot) > 0$*, define the set of optimal parameters as* $\Theta^*_{CSD} = \arg\min_{\theta \in \Theta} \mathcal{L}_{CSD}(\theta; p_T, w)$*. Then, for any* $\theta^* \in \Theta^*_{CSD}$*, we have* $\mathcal{L}_{CSD}(\theta^*; p_T, w) = 0$*, and the following holds for all* $y_t \in \mathcal{V}$*:*

$$q_{\theta^*}(y_t | \mathbf{c}, \mathbf{y}_{<t}) = p_T(y_t | \mathbf{c}, \mathbf{y}_{<t}).$$

*Proof.* We have the following objective:

$$\mathcal{L}_{\text{CSD}}(\theta; p_T, w) := \frac{1}{2} \left[ \sum_{y_t \in \mathcal{V}} \sum_{x \in \mathcal{V}} w(y_t, x) \left( \log \frac{q_\theta(x | \mathbf{c}, \mathbf{y}_{<t})}{q_\theta(y_t | \mathbf{c}, \mathbf{y}_{<t})} - \log \frac{p_T(x | \mathbf{c}, \mathbf{y}_{<t})}{p_T(y_t | \mathbf{c}, \mathbf{y}_{<t})} \right)^2 \right] \tag{10}$$

$$= \frac{1}{2} \sum_{y_t \in \mathcal{V}} \sum_{x \in \mathcal{V}} w(y_t, x) \left( f_\theta[x] - f_\theta[y_t] - f_T[x] + f_T[y_t] \right)^2. \tag{11}$$

Since the objective is a weighted sum of squares with strictly positive weights $w(\cdot, \cdot) > 0$, the loss attains its minimum if and only if each squared term vanishes, i.e.

$$f_{\theta^*}[x] - f_{\theta^*}[y_t] = f_T[x] - f_T[y_t], \quad \forall\, y_t, x \in \mathcal{V}. \tag{12}$$

Then, the probability of a student satisfying the following:

$$q_{\theta^*}(y_t | \mathbf{c}, \mathbf{y}_{<t}) = \frac{\exp(f_{\theta^*}[y_t])}{\sum_{x \in \mathcal{V}} \exp(f_{\theta^*}[x]))} = \frac{\exp(f_{\theta^*}[y_t])}{\sum_{x \in \mathcal{V}} \exp(f_{\theta^*}[y_t] + f_T[x] - f_T[y_t]))} \tag{13}$$

$$= \frac{\exp(f_T[y_t])}{\sum_{x \in \mathcal{V}} \exp(f_T[x]))} = p_T(y_t | \mathbf{c}, \mathbf{y}_{<t}). \tag{14}$$

$\square$

## A.2 PROOF OF THEOREM 2

**Theorem 2.** *(Solution Superset) Assume model capacity* $|\Theta| \to \infty$*, let the set of optimal parameters* $\Theta^*_{CSD} = \arg\min_{\theta \in \Theta} \mathcal{L}_{CSD}(\theta; p_T, w)$ *and* $\Theta^*_{DLD} = \arg\min_{\theta \in \Theta} \mathcal{L}_{DLD}(\theta; p_T, w)$*, then following holds:*

$$\Theta^*_{CSD} \supsetneq \Theta^*_{DLD}.$$

*Proof.* We have the following objective for direct logit distillation (DLD):

$$\mathcal{L}_{\text{DLD}}(\theta; p_T, w) = \frac{1}{2} \sum_{y_t \in \mathcal{V}} w(y_t) \left( f_\theta[y_t] - f_T[y_t] \right)^2, \tag{15}$$

Since the loss is expressed as a strictly positive weighted sum of squares, it achieves its minimum value only when all squared terms are individually zero, i.e.,

$$f_{\theta^*_{\text{DLD}}}[y_t] = f_T[y_t], \quad \forall\, y_t \in \mathcal{V}. \tag{16}$$

Unlike DLD, the optimality condition of our loss is more relaxed. Specifically, it is sufficient for $\theta^*$ to satisfy the condition in Eq. (12), i.e.,

$$f_{\theta^*}[y_t] = f_T[y_t] + C, \quad \forall\, y_t \in \mathcal{V}, \quad C \in \mathbb{R}. \tag{17}$$

At $C = 0$, our objective recovers the solution set of DLD; for an arbitrary choice of $C$, it yields a strictly larger optimal solution set. This arises from the fact that the softmax mapping used to express probabilities is invariant under additive constants, whereas DLD explicitly constrains this constant to coincide with that of the teacher, which consequently reduces the solution set.

$\square$

## A.3 PROOF OF THEOREM 3

**Theorem 3.** *(Efficient Gradient Computation) Assume $w(y_t, x) = w_1(y_t)w_2(x)$, then the gradient of $\mathcal{L}_{CSD}(\theta; p_T, w)$ with respect to $\theta$ could be computed in $\mathcal{O}(|\mathcal{V}|)$ as:*

$$\nabla_\theta \mathcal{L}_{CSD}(\theta; p_T, w) = \sum_{y_t \in \mathcal{V}} \mathbf{w}(y_t)^T \left( \tilde{\mathbf{f}}_\theta[y_t] - \tilde{\mathbf{f}}_T[y_t] \right) \nabla_\theta f_\theta[y_t], \tag{9}$$

*where $\mathbf{w}(y_t) = (w_1(y_t), w_2(y_t))^T$, $\tilde{\mathbf{f}}_\theta[y_t] = \left( \tilde{f}_\theta^{w_2}[y_t], \tilde{f}_\theta^{w_1}[y_t] \right)^T$, $\tilde{\mathbf{f}}_T[y_t] = \left( \tilde{f}_T^{w_2}[y_t], \tilde{f}_T^{w_1}[y_t] \right)^T$, with $\tilde{f}_\theta^w[y_t] = f_\theta[y_t] - \mathbb{E}_{w(x)}[f_\theta[x]]$, $\tilde{f}_T^w[y_t] = f_T[y_t] - \mathbb{E}_{w(x)}[f_T[x]]$ are normalized logits.*

*Proof.* We have the following objective:

$$\mathcal{L}_{CSD}(\theta; p_T, w) = \frac{1}{2} \sum_{y_t \in \mathcal{V}} \sum_{x \in \mathcal{V}} w_1(y_t) w_2(x) \left( \log \frac{q_\theta(x|\mathbf{c}, \mathbf{y}_{<t})}{q_\theta(y_t|\mathbf{c}, \mathbf{y}_{<t})} - \log \frac{p_T(x|\mathbf{c}, \mathbf{y}_{<t})}{p_T(y_t|\mathbf{c}, \mathbf{y}_{<t})} \right)^2 \tag{18}$$

$$= \frac{1}{2} \sum_{y_t \in \mathcal{V}} \sum_{x \in \mathcal{V}} w_1(y_t) w_2(x) \left( f_\theta[x] - f_\theta[y_t] - f_T[x] + f_T[y_t] \right)^2. \tag{19}$$

And its gradient is given by:

$$\nabla_\theta \mathcal{L}_{CSD}(\theta; p_T, w) = \sum_{y_t \in \mathcal{V}} \sum_{x \in \mathcal{V}} w_1(y_t) w_2(x) \left( f_\theta[x] - f_\theta[y_t] - f_T[x] + f_T[y_t] \right) \nabla_\theta (f_\theta[x] - f_\theta[y_t])$$

$$= \underbrace{\sum_{y_t \in \mathcal{V}} \sum_{x \in \mathcal{V}} w_1(y_t) w_2(x) \left( f_\theta[x] - f_T[x] \right) \nabla_\theta (f_\theta[x])}_{\textcircled{1}} - \underbrace{\sum_{y_t \in \mathcal{V}} \sum_{x \in \mathcal{V}} w_1(y_t) w_2(x) \left( f_\theta[x] - f_T[x] \right) \nabla_\theta (f_\theta[y_t])}_{\textcircled{2}}$$

$$+ \underbrace{\sum_{y_t \in \mathcal{V}} \sum_{x \in \mathcal{V}} w_1(y_t) w_2(x) \left( -f_\theta[y_t] + f_T[y_t] \right) \nabla_\theta (f_\theta[x])}_{\textcircled{3}} - \underbrace{\sum_{y_t \in \mathcal{V}} \sum_{x \in \mathcal{V}} w_1(y_t) w_2(x) \left( -f_\theta[y_t] + f_T[y_t] \right) \nabla_\theta (f_\theta[y_t])}_{\textcircled{4}}$$

$$\textcircled{1} = \sum_{y_t \in \mathcal{V}} w_1(y_t)^{\nearrow 1} \times \sum_{x \in \mathcal{V}} w_2(x) \left( f_\theta[x] - f_T[x] \right) \nabla_\theta (f_\theta[x]) = \sum_{y_t \in \mathcal{V}} w_2(y_t) \left( f_\theta[y_t] - f_T[y_t] \right) \nabla_\theta (f_\theta[y_t])$$

$$\textcircled{4} = \sum_{x \in \mathcal{V}} w_2(x)^{\nearrow 1} \times \sum_{y_t \in \mathcal{V}} w_1(y_t) \left( f_\theta[y_t] - f_T[y_t] \right) \nabla_\theta (f_\theta[y_t]) = \sum_{y_t \in \mathcal{V}} w_1(y_t) \left( f_\theta[y_t] - f_T[y_t] \right) \nabla_\theta (f_\theta[y_t])$$

$$\textcircled{2} = - \left\{ \sum_{x \in \mathcal{V}} w_2(x) \left( f_\theta[x] - f_T[x] \right) \right\} \times \left\{ \sum_{y_t \in \mathcal{V}} w_1(y_t) \nabla_\theta (f_\theta[y_t]) \right\}$$

$$= -\mathbb{E}_{w_2(x)} \left[ f_\theta[x] - f_T[x] \right] \times \sum_{y_t \in \mathcal{V}} w_1(y_t) \nabla_\theta (f_\theta[y_t])$$

$$\textcircled{3} = - \left\{ \sum_{y_t \in \mathcal{V}} w_1(y_t) \left( f_\theta[y_t] - f_T[y_t] \right) \right\} \times \left\{ \sum_{x \in \mathcal{V}} w_2(x) \nabla_\theta (f_\theta[x]) \right\}$$

$$= -\mathbb{E}_{w_1(y_t)} \left[ f_\theta[y_t] - f_T[y_t] \right] \times \sum_{x \in \mathcal{V}} w_2(x) \nabla_\theta (f_\theta[x])$$

$$= -\mathbb{E}_{w_1(x)} \left[ f_\theta[x] - f_T[x] \right] \times \sum_{y_t \in \mathcal{V}} w_2(y_t) \nabla_\theta (f_\theta[y_t])$$

$$④ + ② = \sum_{y_t \in \mathcal{V}} w_1(y_t) \left( f_\theta[y_t] - f_T[y_t] - \mathbb{E}_{w_2(x)}[f_\theta[x] - f_T[x]] \right) \nabla_\theta(f_\theta[y_t])$$

$$= \sum_{y_t \in \mathcal{V}} w_1(y_t) \left( \tilde{f}_\theta^{w_2}[y_t] - \tilde{f}_T^{w_2}[y_t] \right) \nabla_\theta(f_\theta[y_t])$$

$$① + ③ = \sum_{y_t \in \mathcal{V}} w_2(y_t) \left( f_\theta[y_t] - f_T[y_t] - \mathbb{E}_{w_1(x)}[f_\theta[x] - f_T[x]] \right) \nabla_\theta(f_\theta[y_t])$$

$$= \sum_{y_t \in \mathcal{V}} w_2(y_t) \left( \tilde{f}_\theta^{w_1}[y_t] - \tilde{f}_T^{w_1}[y_t] \right) \nabla_\theta(f_\theta[y_t])$$

$$① + ② + ③ + ④ = \sum_{y_t \in \mathcal{V}} \mathbf{w}(y_t)^T \left( \tilde{\mathbf{f}}_\theta[y_t] - \tilde{\mathbf{f}}_T[y_t] \right) \nabla_\theta(f_\theta[y_t])$$

$\square$

## A.4 THE VARIANTS OF DIRECT LOGIT DISTILLATION

We define the DLD variants used for our comparisons in Table 2, Table 3, and Table 5.

$$\mathcal{L}_{\text{DLD}}^{\min}(\theta; p_T, w) = \frac{1}{2} \sum_{y_t \in \mathcal{V}} w(y_t) \left( (f_\theta[y_t] - \min_x f_\theta[x]) - (f_T[y_t] - \min_x f_T[x]) \right)^2,$$

$$\mathcal{L}_{\text{DLD}}^{\max}(\theta; p_T, w) = \frac{1}{2} \sum_{y_t \in \mathcal{V}} w(y_t) \left( (f_\theta[y_t] - \max_x f_\theta[x]) - (f_T[y_t] - \max_x f_T[x]) \right)^2,$$

$$\mathcal{L}_{\text{DLD}}^{\text{std}}(\theta; p_T, w) = \frac{1}{2} \sum_{y_t \in \mathcal{V}} w(y_t) \left( \frac{f_\theta[y_t] - \frac{1}{|\mathcal{V}|} \sum_{x \in \mathcal{V}} f_\theta[x]}{\sqrt{\frac{1}{|\mathcal{V}|} \sum_{x \in \mathcal{V}} \left( f_\theta[x] - \frac{1}{|\mathcal{V}|} \sum_{x' \in \mathcal{V}} f_\theta[x'] \right)^2}} - \frac{f_T[y_t] - \frac{1}{|\mathcal{V}|} \sum_{x \in \mathcal{V}} f_T[x]}{\sqrt{\frac{1}{|\mathcal{V}|} \sum_{x \in \mathcal{V}} \left( f_T[x] - \frac{1}{|\mathcal{V}|} \sum_{x' \in \mathcal{V}} f_T[x'] \right)^2}} \right)^2,$$

$$\mathcal{L}_{\text{DLD}}^{\text{range}}(\theta; p_T, w) = \frac{1}{2} \sum_{y_t \in \mathcal{V}} w(y_t) \left( \left( \frac{2 \left( f_\theta[y_t] - \min_x f_\theta[x] \right)}{\max_x f_\theta[x] - \min_x f_\theta[x]} - 1 \right) - \left( \frac{2 \left( f_T[y_t] - \min_x f_T[x] \right)}{\max_x f_T[x] - \min_x f_T[x]} - 1 \right) \right)^2,$$

$$\mathcal{L}_{\text{DLD}}^{\text{mean}}(\theta; p_T, w) = \frac{1}{2} \sum_{y_t \in \mathcal{V}} w(y_t) \left( \left( f_\theta[y_t] - \frac{1}{|V|} \sum_{x \in \mathcal{V}} f_\theta[x] \right) - \left( f_T[y_t] - \frac{1}{|V|} \sum_{x \in \mathcal{V}} f_T[x] \right) \right)^2,$$

$$\mathcal{L}_{\text{DLD}}^{\text{w-mean}}(\theta; p_T, w) = \frac{1}{2} \sum_{y_t \in \mathcal{V}} w(y_t) \left( \left( f_\theta[y_t] - \sum_{x \in \mathcal{V}} w(x) f_\theta[x] \right) - \left( f_T[y_t] - \sum_{x \in \mathcal{V}} w(x) f_T[x] \right) \right)^2.$$

Here, $\mathcal{L}_{\text{DLD}}^{\text{w-mean}}$ is recovered by CSD as a special case shown by the following remark.

**Remark 4.** *Let CSD objective as $\mathcal{L}_{CSD}(\theta; p_T, w)$ with $w(y_t, x) = w_1(y_t) w_2(x)$. If $w_1(\cdot) = w_2(\cdot)$, $\nabla_\theta \mathcal{L}_{CSD}(\theta; p_T, w) = \nabla_\theta \mathcal{L}_{DLD}^{w\text{-}mean}(\theta; p_T, w_1)$.*

*Proof.* We use $\sum_{y_t \in \mathcal{V}} w_1(y_t) \tilde{f}_\theta^{w_1}[y_t] = \sum_{y_t \in \mathcal{V}} w_1(y_t) \left( f_\theta[y_t] - \sum_{x \in \mathcal{V}} w_1(x) f_\theta[x] \right) = 0.$

$$\nabla_\theta \mathcal{L}_{\text{DLD}}^{\text{w-mean}}(\theta; p_T, w_1) = \sum_{y_t \in \mathcal{V}} w_1(y_t) \left( \tilde{f}_\theta^{w_1}[y_t] - \tilde{f}_T^{w_1}[y_t] \right) \left( \nabla_\theta f_\theta[y_t] - \sum_{x \in \mathcal{V}} w_1(x) \nabla_\theta f_\theta[x] \right)$$

$$= \sum_{y_t \in \mathcal{V}} w_1(y_t) \left( \tilde{f}_\theta^{w_1}[y_t] - \tilde{f}_T^{w_1}[y_t] \right) \nabla_\theta f_\theta[y_t] - \underbrace{\left( \sum_{y_t \in \mathcal{V}} w_1(y_t) \left( \tilde{f}_\theta^{w_1}[y_t] - \tilde{f}_T^{w_1}[y_t] \right) \right)}_{0} \left( \sum_{x \in \mathcal{V}} w_1(x) \nabla_\theta f_\theta[x] \right)$$

$$= \nabla_\theta \mathcal{L}_{\text{CSD}}(\theta; p_T, w)$$

$\square$

### A.5 Weighting for Divergence-based Loss

Unlike the L2 loss, a divergence-based loss does not guarantee convergence to the target distribution when a weighting is applied. Here, we examine the effect of applying $w$ weighting to the KL divergence. Define $p_T^w(y_t) = \frac{p_T(y_t) \times w(y_t)}{Z_{wT}}$, where $Z_{wT} = \sum_{y_t \in \mathcal{V}}(p_T(y_t) \times w(y_t))$ is a partition function. Then we have:

$$
\begin{aligned}
D_{\text{KL}}^w\left(p_T || q_\theta\right) &:= \sum_{y_t \in \mathcal{V}} w(y_t) p_T(y_t) \log \frac{p_T(y_t)}{q_\theta(y_t)}. \\
&= \sum_{y_t \in \mathcal{V}} w(y_t) p_T(y_t) \left( \log \frac{w(y_t) p_T(y_t)}{q_\theta(y_t)} - \log w(y_t) \right). \\
&= Z_{wT} \times \sum_{y_t \in \mathcal{V}} \frac{w(y_t) p_T(y_t)}{Z_{wT}} \left( \log \frac{w(y_t) p_T(y_t)}{Z_{wT} \times q_\theta(y_t)} + \log Z_{wT} - \log w(y_t) \right) \\
&= Z_{wT} \times \sum_{y_t \in \mathcal{V}} p_T^w(y_t) \left( \log \frac{p_T^w(y_t)}{q_\theta(y_t)} + \log Z_{wT} - \log w(y_t) \right) \\
&= Z_{wT} \times \sum_{y_t \in \mathcal{V}} p_T^w(y_t) \log \frac{p_T^w(y_t)}{q_\theta(y_t)} + C \\
&= Z_{wT} D_{\text{KL}}\left(p_T^w || q_\theta\right) + C,
\end{aligned}
$$

where $C$ is constant with respect to $\theta$. Thus, in this case, the student $q_\theta$ does not converge to the teacher distribution but instead converges to $p_T^w$, meaning that the target distribution is altered by the weighting $w$. In contrast, the proposed CSD theoretically guarantees convergence to the target distribution for any choice of $w$ proved by Proposition 1.

## B    Related Works

**Distillation loss:** The choice of discrepancy metric between teacher and student probability distributions is central to knowledge distillation for large language models (LLMs). Prior work has predominantly employed either forward KL divergence (Hinton et al., 2015) or reverse KL divergence (Gu et al., 2024). These divergences, however, exhibit distinct biases: forward KL is inherently mode-covering, while reverse KL is mode-seeking. Consequently, optimization under either measure imposes an unavoidable trade-off between fidelity and diversity. To address this limitation, recent studies have explored alternative measures, including (generalized) Jensen–Shannon divergence (Wen et al., 2023; Agarwal et al., 2024), adaptive KL divergence (Wu et al., 2025), and $\alpha$–$\beta$ divergence (Wang et al., 2025). Complementarily, Ko et al. (2024) introduced skew KL and skew reverse KL divergences to improve optimization stability. Beyond the KL family, total variation distance has also been investigated (Wen et al., 2023). AMiD (Shin et al., 2026) extends the distillation objective by aligning either the student or teacher distribution with a generalized assistant distribution under the arbitrary divergence framework. Broadly, existing approaches extend in two directions: (i) instantiating different generating functions within the $f$-divergence family, or (ii) constructing hybrid objectives that combine multiple divergences. In contrast, we propose a novel logit-level distillation framework grounded in concrete-score matching (Meng et al., 2022), which departs from the $f$-divergence family and offers both extensibility and originality. Furthermore, we introduce a loss design space with multiple instances, including instances that envelope the diversity–fidelity trade-off exhibited by previous methods.

**Distillation dataset:** Concurrently, a complementary line of work has examined dataset composition to mitigate the distribution mismatch between training and inference. Several studies have explored on-policy strategies, either using only student-generated outputs (Lin et al., 2020) or combining them with a fixed dataset (Agarwal et al., 2024) and teacher-generated outputs (Gu et al., 2024; Xu et al., 2025). To reduce the computational overhead of on-policy training, Ko et al. (2024) proposed an adaptive off-policy method with a replay buffer. By contrast, our contribution focuses on developing a novel discrepancy metric, which is orthogonal to these dataset composition strategies and can be seamlessly integrated with them as shown in Table 2.

**Autoregressive model meets score function:** BPO (Kim et al., 2025b) generalizes the preference optimization objective for LLMs by interpreting it as sequence-level concrete score matching. Unlike CSD, which formulates concrete score matching at the per-token level over the entire vocabulary to match a student model to a teacher model, BPO defines concrete scores at the sequence level between preferred and dispreferred responses and matches them to preference data. To enable tractable learning from such data, BPO further extends the objective within a Bregman divergence framework. ARD (Kim et al., 2025a), in contrast, extends the distillation of score-based diffusion models to an autoregressive model. Specifically, it employs an autoregressive student to learn, through regression, the integration of the ODE defined by the score function.

## C  EXPERIMENTAL DETAILS

### C.1  TASK-AGNOSTIC INSTRUCTION FOLLOWING DISTILLATION IN SECTION 4.1.

**Experimental setup:** We follow the training setup of DistiLLM (Ko et al., 2024). For the distillation dataset $\mathcal{D}$, we use `databricks-dolly-15k` (Conover et al., 2023), containing about 14,000 samples for training, with 500 held out for validation and 500 for evaluation. For comparison with the baseline, we optionally add a pretraining loss using the pretraining dataset `OpenWebText` (Gokaslan & Cohen, 2019) in some cases of Table 2. We first fine-tune the `GPT-2-1.5B` (Radford et al., 2019) teacher on the dataset, and then distill it into `GPT-2-0.1B` and `GPT-2-0.3B` students. Similarly, we distill `OpenLLaMA-7B` (Geng & Liu, 2023) into `OpenLLaMA-3B`. We determined the learning rate and batch size by referring to the search ranges used in prior studies (Gu et al., 2024; Ko et al., 2024). We use the detached student probability as the default choice for both $w_1$ and $w_2$, and analyze alternative choices through ablation studies.

All experiments were conducted primarily on four RTX 3090 GPUs. We searched learning rates in [5e-4, 1e-4, 5e-5] and batch sizes in [8, 16, 32]. Each configuration was trained for 20 epochs, saving a checkpoint at every epoch, and evaluated using the checkpoint with the highest validation ROUGE-L score. We used the same five evaluation seeds [10, 20, 30, 40, 50] as in prior work to compute the mean and standard deviation of the evaluation metric. The baselines in Table 2 were run with the official code settings of prior work (Ko et al., 2024), with additional tuning for the batch size. In the OpenLLaMA experiments, all baselines and ours were standardized to a batch size of 8, the maximum supported in our environment. Baselines used the learning rates from their official code, while we fixed the learning rate to 1e-4 (commonly effective for GPT-2) with CSD, without further tuning. For ablation studies in Table 5 and Figure 5, we used the same configuration: learning rate 1e-4 and batch size 8. For GPT-4 feedback in Figure 4, we use the following templates following prior work (Zheng et al., 2023; Ko et al., 2024) as shown below. We computed the ratio between the model answer and the golden answer for each of the 500 samples from Dolly Eval, and reported the average over all samples. We provide the reference implementation for CSD in Code 1.

**Baselines:** Since our main focus is on the loss function, we compared our method with existing objectives using the same teacher checkpoint. The baselines include KL, reverse KL (RKL) (Gu et al., 2024), symmetric KL (the mean of KL and RKL), Jeffrey's divergence, Total Variation (Wen et al., 2023), Generalized Jensen–Shannon (GJS) (Agarwal et al., 2024) with smoothing parameter 0.9, Skewed KL (SKL) (Ko et al., 2024), Skewed reverse KL (SRKL) (Ko et al., 2024) with smoothing parameter 0.1, and $\alpha$–$\beta$ divergence (AB) (Wang et al., 2025) with parameters (0.2, 0.7). We followed the hyperparameter choices reported in each paper and the implementation of DistiLLM. For KL, we performed a full-range hyperparameter search, as in our method. For losses not specified in prior work, we adopted the same settings as for KL.

**Evaluation metrics and setups:** We evaluated on five instruction-following benchmarks: 1) the test set of Dolly, 2) Self-Instruct (Wang et al., 2023), 3) Vicuna Eval (Chiang et al., 2023), 4) Super-Natural Instructions (Super-NI) (Wang et al., 2022), and 5) Unnatural Instructions (UnNI) (Honovich et al., 2023). ROUGE-L (Lin, 2004), which measures similarity to the golden answer, was used as the primary metric. We additionally employed Self-BLEU (Zhu et al., 2018) and Distinct-N (Li et al., 2016) as diversity metrics. Furthermore, GPT-4 feedback (Zheng et al., 2023) was used as a proxy for human judgment. Checkpoints were saved at each epoch, with evaluation performed on the one achieving the best validation ROUGE-L. The decoding temperature was set to 1 by default, and following prior work, reduced to 0.7 for GPT-judge evaluation.

## C.2 Task-specific Distillation in Section 4.2.

**Experimental setup:** We verify the effectiveness of CSD across diverse tasks, including dialogue summarization, low-resource translation, and arithmetic reasoning. Distillation was conducted on DialogSum (Chen et al., 2021), Flores-200 (Costa-Jussà et al., 2022), and GSM8K (Cobbe et al., 2021) datasets, following the experimental setup of Xu et al. (2025) with a fixed dataset. We used `Gemma-7B-IT` (Team et al., 2024a), fine-tuned with SFT as the teacher and `Gemma-2B-IT` as the student. We compared with the baselines using the same teacher, changing only the loss function.

For teacher SFT, we trained summarization and arithmetic reasoning for 3 epochs and translation for 10 epochs, using the full datasets. Model evaluation was performed every 16 steps, and the checkpoint with the lowest validation loss was selected. The batch size was fixed to 128 for all tasks, with the learning rate set to 1e-5. For each task in distillation, we distilled both the baselines and our method from the same teacher checkpoint with a fixed learning rate of 1e-5 and batch size of 8, using about 1,000 samples. We trained for 3 epochs on summarization and arithmetic reasoning, and 10 epochs on translation. For the baselines, checkpoints were saved every 25 steps, and the one with the lowest validation loss was used for evaluation. For CSD, since the loss itself cannot be directly computed and training relies on its gradient, validation loss was unavailable; thus, we evaluated using the final checkpoint. For all tasks, we set $w_1$ and $w_2$ using the teacher's and student's probabilities. For evaluation, we used task-specific metrics: COMET (Rei et al., 2022) for translation, ROUGE-L (Lin, 2004) for summarization, and answer accuracy for arithmetic reasoning, all evaluated on each task's test dataset.

## C.3 General Chat Capability Distillation in Section 4.3.

**Experimental setup:** We closely followed the official code of DistiLLM-2 (Ko et al., 2025) for our distillation setup. For 50k UltraChat prompts, we generated samples from both the student and the teacher. We then applied the CSD and DLD matching losses to each pair of generated samples. Although methods such as DPKD (Li et al., 2024) and DistiLLM-2 define different losses depending on which model produced the sample, CSD could also benefit from adopting such a strategy, suggesting room for further improvement. For evaluation, we followed the SimPO (Meng et al., 2024) protocol. We used the same learning rate and batch size as DistiLLM-2.

---

**Algorithm 2:** Monte Carlo estimation to compute $\mathcal{L}_{\text{CSD}}$ in Eq. (8)

---

**Input:** Student $f_\theta$, teacher $f_T$, prompt $\mathbf{c}$, prefix $\mathbf{y}_{<t}$, function $w(\cdot, \cdot) = w_1(\cdot)w_2(\cdot|\cdot)$.

1 Compute the student logit $f_\theta[y_t] = f_\theta(\mathbf{c}, \mathbf{y}_{<t})[y_t], \forall y_t \in \mathcal{V}$.

2 Compute the teacher logit $f_T[y_t] = f_T(\mathbf{c}, \mathbf{y}_{<t})[y_t], \forall y_t \in \mathcal{V}$.

3 Sample $y_t$ according to $w_1(\cdot)$.

4 $\mathcal{L}_{\text{CSD}}^{\text{MC}}(\theta; p_T, w) = \sum_{x \in \mathcal{V}}[w_2(x|y_t) \times (f_\theta[x] - f_\theta[y_t] - f_T[x] + f_T[y_t])^2]$

5 **return** $\nabla_\theta \mathcal{L}_{\text{CSD}}^{\text{MC}}(\theta; p_T, w)$

---

> **GPT-4 feedback template**
>
> [System] Please act as an impartial judge and evaluate the quality of the response provided by an AI assistant to the user question displayed below. Your evaluation should consider factors such as the helpfulness, relevance, accuracy, depth, creativity, and level of detail of the response. Begin your evaluation by providing a short explanation. Be as objective as possible. After providing your explanation, please rate the response on a scale of 1 to 10 by strictly following this format: "[[rating]]", for example: "Rating: [[5]]".
>
> [Question]
> {question}
>
> [The Start of Assistant's Answer]
> {answer}
> [The End of Assistant's Answer]

```python
import torch
import torch.nn.functional as F

def CSD_loss(student_logits, teacher_logits, mode):
    student_probs = F.softmax(student_logits, dim=-1)
    teacher_probs = F.softmax(teacher_logits, dim=-1)

    if mode == "SS":
        loss = (student_logits - teacher_logits - torch.sum(student_probs * (student_logits -
            teacher_logits), dim=-1,keepdim=True)).detach() * student_probs.detach() *
            student_logits

    elif mode == "TS":
        loss1 = (student_logits - teacher_logits - torch.sum(teacher_probs * (student_logits -
            teacher_logits), dim=-1,keepdim=True)).detach() * student_probs.detach() *
            student_logits
        loss2 = (student_logits - teacher_logits - torch.sum(student_probs * (student_logits -
            teacher_logits), dim=-1,keepdim=True)).detach() * teacher_probs * student_logits
        loss = (loss1 + loss2) / 2

    distil_loss = torch.sum(loss, dim=-1) ## summation over vocab

    return distil_loss
```

Code 1: CSD loss function implementation

# D   ADDITIONAL EXPERIMENTAL RESULTS

This section presents additional experimental results. Figure 6 shows the logit and probability statistics of the GPT-2-1.5B teacher, corresponding to Figure 1. Figure 7 illustrates further fidelity–diversity trade-offs using Distinct-N metrics, corresponding to Figure 3a. Figure 8 presents validation ROUGE-L scores during training, corresponding to Table 1. CSD not only converges to a higher point but also achieves faster performance gains in the early stages. Table 7 provides comparisons with additional baselines corresponding to Table 2, and Table 8 compares CSD with the MSE probability-matching objective under different weighting schemes. Finally, Tables 11 to 15 present case studies of model generations for math questions.

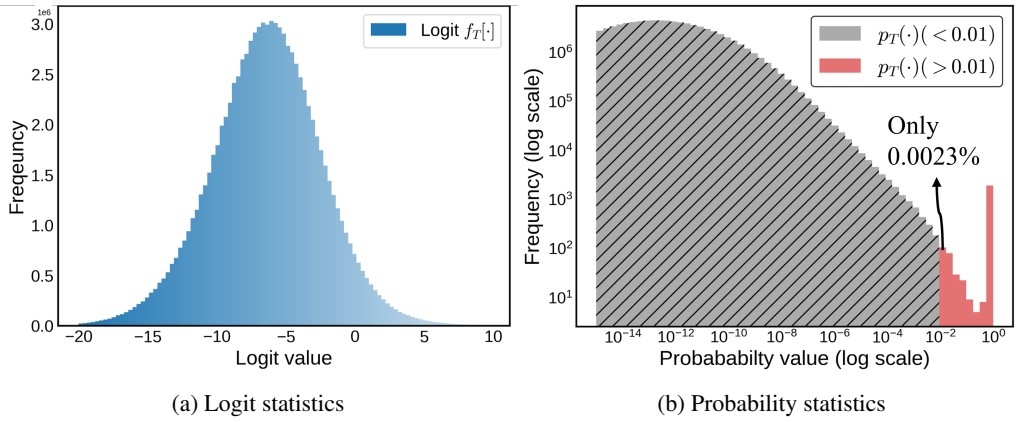

(a) Logit statistics                         (b) Probability statistics

Figure 6: Comparison between teacher's logit and probability statistics. While the logits span a wide range from $-20$ to $5$ and convey rich information, the probabilities are mostly concentrated near zero.

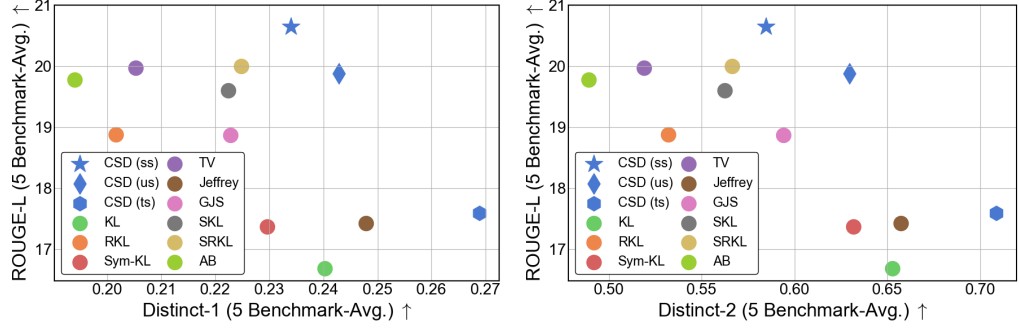

Figure 7: Fidelity vs. Diversity trade-off with more metrics.

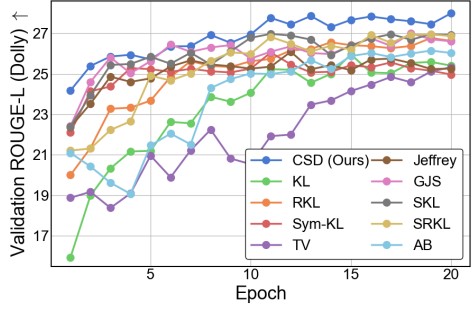

Figure 8: Validation ROUGE-L scores over training epochs.

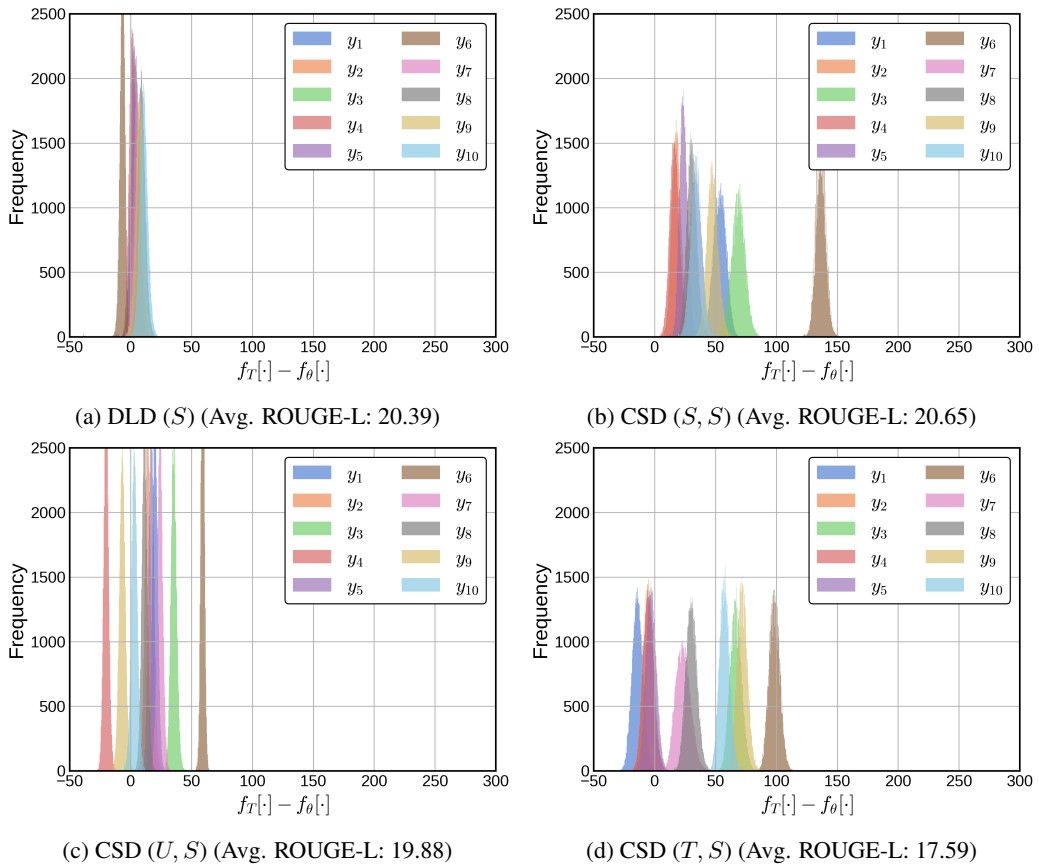

Figure 9: Solution set restriction of direct logit distillation (DLD) and the flexible selection of logit residual constants in *Concrete Score Distillation* (CSD). CSD finds a broader solution space.

### D.1 ANALYSIS ON THE LOGIT OFFSETS.

Figure 9 shows the logit offsets between the teacher and the student for 10 consecutive tokens within a sentence. This demonstrates that DLD converges only to solutions with zero residual constants, whereas CSD learns token-dependent residual constants. In other words, CSD explores a wide solution space during the training. Figure 11 also shows how the offset for the same token changes across training epochs. We observe that an appropriate offset for each token is determined early in training, after which the model consistently refines its solution around that offset.

Figure 9c shows that CSD $(U, S)$ is more tightly centered. As analyzed in Figure 13, vocabulary items are learned more uniformly, causing the logits to cluster around the token-wise offset. This indicates that minority vocabulary items are also well learned, which helps explain why the method performs exceptionally well under the high-temperature sampling setting of Figure 3b, where the contribution of minority vocabulary becomes more significant.

Figure 10 presents the averaged KL, probability MSE, and ECE errors across training epochs, and Table 6 shows the per-instance values corresponding to Figure 9. Since the probabilities of specific vocabularies (those with high probabilities in the student or teacher) are more important than the full vocabulary in these metrics, CSD $(T, S)$ performs well because it learns with probability weighting from the teacher and student.

In contrast, the generative performance was highest with CSD $(S, S)$. This is because generative performance only needs good quality in the regions favored by the student, which is often negatively correlated with probability calibration (Achiam et al., 2023; Wang et al., 2024).

$$D_{\mathrm{KL}}\left(p_T\|q_\theta\right) := \sum_{y_t \in \mathcal{V}} p_T(y_t) \log \frac{p_T(y_t)}{q_\theta(y_t)}.$$

$$\mathrm{MSE}\left(p_T, q_\theta\right) := \sum_{y_t \in \mathcal{V}} \left(p_T(y_t) - q_\theta(y_t)\right)^2.$$

$$\mathrm{ECE}\left(p_T, q_\theta\right) := \sum_{y_t \in \mathcal{V}} q_\theta(y_t)|p_T(y_t) - q_\theta(y_t)|.$$

Table 6: Instance-wise probability calibration results for various logit distillation methods correspond to Figure 9.

| | $y_1$ | $y_2$ | $y_3$ | $y_4$ | $y_5$ | $y_6$ | $y_7$ | $y_8$ | $y_9$ | $y_{10}$ | AVG |
|---|---|---|---|---|---|---|---|---|---|---|---|
| **KL Divergence** | | | | | | | | | | | |
| DLD $(S)$ | 0.61 | 5.40 | 16.99 | 0.00 | 0.58 | 0.00 | 9.49 | 10.14 | 0.00 | 17.41 | 6.06 |
| CSD $(S, S)$ | 8.98 | 11.25 | 12.37 | 0.02 | 0.29 | 0.00 | 11.57 | 12.57 | 0.01 | 12.16 | 6.92 |
| CSD $(U, S)$ | 6.41 | 9.90 | 12.00 | 0.03 | 0.48 | 0.00 | 10.38 | 13.84 | 0.04 | 12.08 | 6.52 |
| CSD $(T, S)$ | 8.67 | 0.70 | 2.61 | 5.22 | 0.47 | 0.00 | 0.01 | 0.01 | 1.59 | 0.02 | 1.93 |
| **Mean Squared Error** | | | | | | | | | | | |
| DLD $(S)$ | 0.25 | 1.01 | 1.61 | 0.00 | 0.29 | 0.00 | 1.04 | 1.50 | 0.00 | 2.00 | 0.77 |
| CSD $(S, S)$ | 1.13 | 1.17 | 1.94 | 0.00 | 0.00 | 0.00 | 1.32 | 2.00 | 0.00 | 2.00 | 0.96 |
| CSD $(U, S)$ | 0.60 | 1.03 | 1.39 | 0.00 | 0.01 | 0.00 | 1.01 | 1.48 | 0.00 | 1.96 | 0.75 |
| CSD $(T, S)$ | 0.84 | 0.02 | 0.08 | 0.39 | 0.01 | 0.00 | 0.00 | 0.00 | 0.03 | 0.00 | 0.14 |
| **Expected Calibration Error** | | | | | | | | | | | |
| DLD $(S)$ | 0.29 | 0.03 | 0.61 | 0.00 | 0.34 | 0.00 | 0.04 | 0.50 | 0.00 | 1.00 | 0.28 |
| CSD $(S, S)$ | 0.39 | 0.18 | 0.94 | 0.00 | 0.05 | 0.00 | 0.32 | 1.00 | 0.00 | 1.00 | 0.39 |
| CSD $(U, S)$ | 0.28 | 0.04 | 0.39 | 0.01 | 0.08 | 0.00 | 0.01 | 0.48 | 0.00 | 0.96 | 0.22 |
| CSD $(T, S)$ | 0.25 | 0.11 | 0.20 | 0.36 | 0.08 | 0.00 | 0.00 | 0.00 | 0.14 | 0.00 | 0.11 |

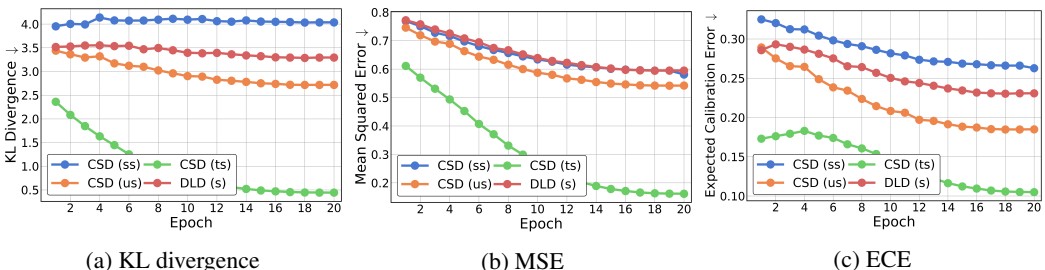

(a) KL divergence       (b) MSE       (c) ECE

Figure 10: Averaged probability calibration results during the training.

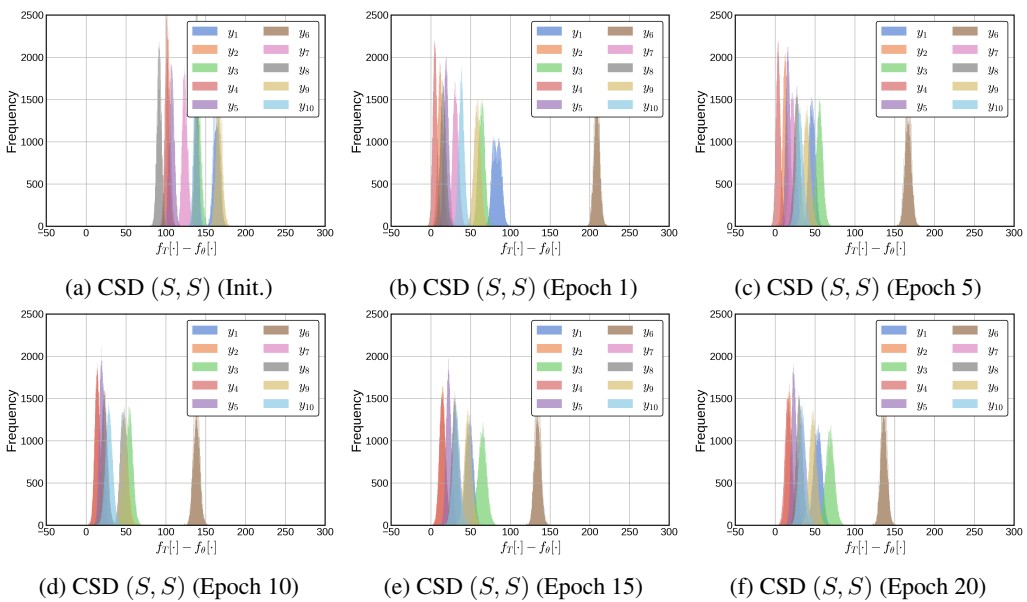

Figure 11: Logit offsets dynamics during the training of CSD.

## D.2 ADAPTIVE LOSS WEIGHTING

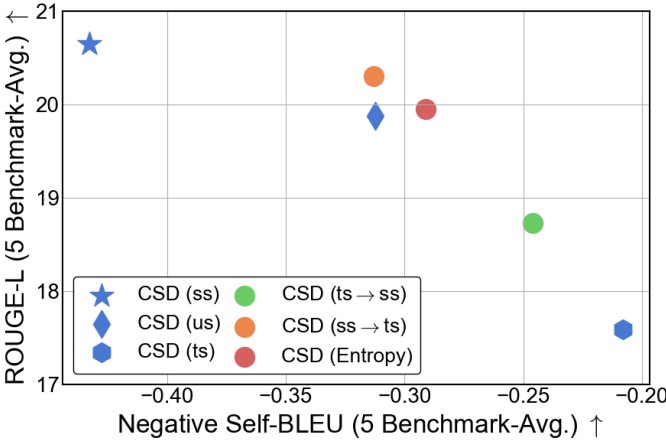

Figure 12: Adaptive loss weighting

Diversity and fidelity form an inherent trade-off in generative modeling; a well-balanced default option is also important. Because an adaptive loss can better reconcile this trade-off, we conducted additional experiments on the adaptive loss weighting. We provide the results of two naïve scheduling and one confidence-based adaptive loss that interpolates CSD $(S, S)$ and CSD $(T, S)$ by defining $w_1$ as an interpolation of $p_S$ and $p_T$ using $\alpha$. We found that the following geometric interpolation performs better than linear interpolation in balancing the fidelity–diversity trade-off:

$$w_1(x) \propto p_s(x)^\alpha p_T(x)^{1-\alpha}, \quad w_2(x) = p_s(x).$$

$$\text{CSD } (TS \to SS): \alpha = \frac{\text{Current Epoch}}{\text{Total Epoch}}$$

$$\text{CSD } (SS \to TS): \alpha = \frac{\text{Total Epoch} - \text{Current Epoch}}{\text{Total Epoch}}$$

$$\text{CSD (Entropy)}: \alpha = \text{clip}\left(\frac{H(p_s(x)) - H(p_T(x))}{H(p_s(x))}, 0, 1\right)$$

CSD ($TS \to SS$), CSD ($SS \to TS$), and CSD (Entropy) combine the strengths of both CSD ($S, S$) and CSD ($T, S$), and therefore achieve better performance at intermediate trade-off points as shown in Figure 12. Because the learning rate is high in the early stages and decreases over time, the loss used at the beginning of training tends to have a stronger influence on the final trade-off position. For example, CSD ($TS \to SS$) behaves similarly to CSD ($T, S$) because its early-stage loss is closer to CSD($T, S$).

Unlike other epoch-based scheduling, CSD (Entropy) adaptively sets $\alpha$ at each token every step. Early in training, the entropy $H(p_s)$ is typically larger than $H(p_T)$, so $\alpha$ becomes close to 1, making the loss similar to CSD ($S, S$). Since $p_s$ is more diverse than $p_T$ at the early stage, the CSD ($S, S$) weighting provides richer feedback over a larger set of vocabulary indices compared to CSD ($T, S$).

In the later stages of training, $p_s$ ideally becomes closer to $p_T$, making $\alpha$ close to 0. In a well-trained situation, CSD ($S, S$) and CSD ($T, S$) will show similar behavior, so the exact value of $\alpha$ becomes less important. However, when training does not progress well and $p_s$ becomes overconfident without matching $p_T$, focusing solely on CSD ($S, S$) weighting is undesirable. In such cases, stronger teacher guidance from CSD ($T, S$) is needed, which is why CSD (Entropy) is designed as above. With this design, the model showed more balanced performance.

### D.3 GRADIENT COEFFICIENT DIVERSITY

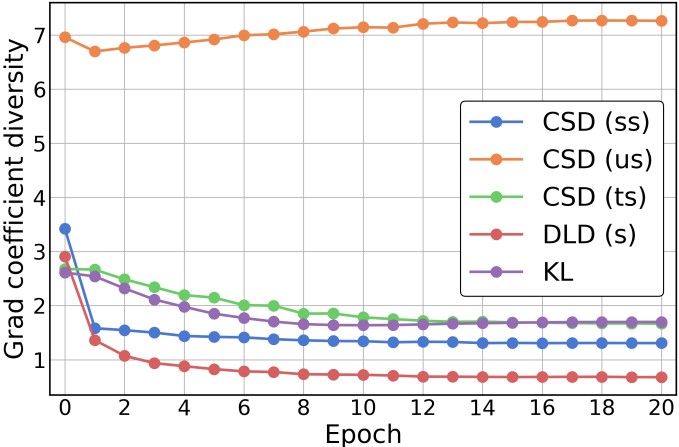

Figure 13: Gradient coefficient diversity.

The limitation of softmax-based divergence losses, as pointed out in Figures 1a and 1b, is that they provide almost no learning signal for minority vocabulary items. In this section, we analyze how broadly CSD learns across different vocabulary items. For CSD, we know the gradient coefficient for each vocabulary item from Eq. (9), which is given as follows:

$$\text{Coeff}(y_t) = \mathbf{w}(y_t)^T \left( \tilde{\mathbf{f}}_\theta[y_t] - \tilde{\mathbf{f}}_T \cdot [y_t] \right)$$

We take the absolute value of this coefficient, normalize it by dividing by a constant so that the values sum to one, and then measure its entropy across training epochs. Figure 13 shows that when both weightings come from either the teacher's or the student's probabilities, the model learns only a small subset of vocabulary items, similar to KL. In contrast, CSD ($U, S$) learns from a much broader range of vocabulary.

This demonstrates that expanding the loss-design space beyond the smoothing behavior imposed by softmax can be effective, which was the main motivation of this work. Because CSD ($U, S$) learns uniformly across all vocabulary items, the logits for all vocabularies are well centered around their respective offsets, as shown in Figure 9c. This also explains its strong performance under high-temperature sampling (where minority vocabulary contributions become more important) as shown in Figure 3b.

Table 7: Comparison with more baselines corresponds to Table 2.

| Method | Loss | $\mathcal{D}$ | Dolly Eval | Self-Instruct | Vicuna Eval | Super-NI | UnNI | Avg. (↑) |
|---|---|---|---|---|---|---|---|---|
| Teacher (GPT-2-1.5B) | | | $27.00_{\pm 0.19}$ | $14.07_{\pm 0.37}$ | $16.31_{\pm 0.32}$ | $26.46_{\pm 0.41}$ | $31.10_{\pm 0.06}$ | 22.99 |
| GPT-2-1.5B → GPT-2-0.1B | | | | | | | | |
| SFT | SFT | Fix | $23.49_{\pm 0.25}$ | $10.56_{\pm 0.29}$ | $15.09_{\pm 0.48}$ | $17.13_{\pm 0.12}$ | $19.97_{\pm 0.08}$ | 17.25 |
| SeqKD (Kim & Rush, 2016) | SFT | $p_T$ | $23.86_{\pm 0.49}$ | $11.67_{\pm 0.80}$ | $14.73_{\pm 0.37}$ | $21.04_{\pm 0.19}$ | $23.55_{\pm 0.11}$ | 18.97 |
| KD (Hinton et al., 2015) | KL | Fix | $23.52_{\pm 0.25}$ | $10.02_{\pm 0.58}$ | $14.57_{\pm 0.32}$ | $16.76_{\pm 0.17}$ | $18.55_{\pm 0.13}$ | 16.68 |
| Ours | CSD | Fix | $24.94_{\pm 0.29}$ | $12.06_{\pm 0.46}$ | $15.78_{\pm 0.49}$ | $\mathbf{24.60}_{\pm 0.31}$ | $\mathbf{25.88}_{\pm 0.13}$ | 20.65 |
| Ours | CSD | On | $\mathbf{25.70}_{\pm 0.23}$ | $\mathbf{12.40}_{\pm 0.48}$ | $\mathbf{17.18}_{\pm 0.52}$ | $22.91_{\pm 0.46}$ | $25.47_{\pm 0.17}$ | $\mathbf{20.73}$ |
| GPT-2-1.5B → GPT-2-0.3B | | | | | | | | |
| SFT | SFT | Fix | $25.09_{\pm 0.62}$ | $12.23_{\pm 0.79}$ | $16.24_{\pm 0.40}$ | $23.42_{\pm 0.11}$ | $26.99_{\pm 0.13}$ | 20.79 |
| SeqKD (Kim & Rush, 2016) | SFT | $p_T$ | $24.79_{\pm 0.26}$ | $11.03_{\pm 0.95}$ | $15.27_{\pm 0.30}$ | $18.91_{\pm 0.29}$ | $21.78_{\pm 0.10}$ | 18.36 |
| KD (Hinton et al., 2015) | KL | Fix | $25.41_{\pm 0.52}$ | $11.15_{\pm 0.20}$ | $15.83_{\pm 0.26}$ | $20.13_{\pm 0.38}$ | $23.57_{\pm 0.13}$ | 19.22 |
| Ours | CSD | On | $\mathbf{27.14}_{\pm 0.28}$ | $\mathbf{14.85}_{\pm 0.66}$ | $\mathbf{16.88}_{\pm 0.18}$ | $\mathbf{26.28}_{\pm 0.21}$ | $\mathbf{30.43}_{\pm 0.04}$ | $\mathbf{23.12}$ |

Table 8: Comparison with probability matching loss with various weighting functions.

| Loss | $w_1(\cdot)$ | $w_2(\cdot)$ | Dolly Eval | Self-Instruct | Vicuna Eval | Super-NI | UnNI | Avg. (↑) |
|---|---|---|---|---|---|---|---|---|
| Prob L2 | $T$ | - | $24.41_{\pm 0.09}$ | $11.45_{\pm 0.25}$ | $14.43_{\pm 0.68}$ | $24.08_{\pm 0.30}$ | $25.53_{\pm 0.04}$ | 19.98 |
| | $U$ | - | $15.62_{\pm 0.37}$ | $6.59_{\pm 0.49}$ | $10.63_{\pm 0.44}$ | $10.31_{\pm 0.34}$ | $12.51_{\pm 0.14}$ | 11.13 |
| | $S$ | - | $16.43_{\pm 0.14}$ | $6.51_{\pm 0.55}$ | $9.73_{\pm 0.17}$ | $10.94_{\pm 0.31}$ | $13.16_{\pm 0.20}$ | 11.35 |
| KL | $T$ | - | $23.65_{\pm 0.44}$ | $10.36_{\pm 0.19}$ | $15.10_{\pm 0.41}$ | $16.18_{\pm 0.36}$ | $19.64_{\pm 0.07}$ | 16.99 |
| | $U$ | - | $23.52_{\pm 0.25}$ | $10.02_{\pm 0.58}$ | $14.57_{\pm 0.32}$ | $16.76_{\pm 0.17}$ | $18.55_{\pm 0.13}$ | 16.68 |
| | $S$ | - | $23.18_{\pm 0.34}$ | $10.04_{\pm 0.43}$ | $15.06_{\pm 0.29}$ | $16.93_{\pm 0.22}$ | $19.78_{\pm 0.12}$ | 17.00 |
| TV | $T$ | - | $24.04_{\pm 0.33}$ | $10.99_{\pm 0.41}$ | $14.68_{\pm 0.19}$ | $\mathbf{25.40}_{\pm 0.06}$ | $25.24_{\pm 0.04}$ | 20.07 |
| | $U$ | - | $23.88_{\pm 0.30}$ | $11.03_{\pm 0.51}$ | $15.13_{\pm 0.44}$ | $24.58_{\pm 0.25}$ | $25.24_{\pm 0.06}$ | 19.97 |
| | $S$ | - | $3.21_{\pm 0.41}$ | $0.51_{\pm 0.10}$ | $0.97_{\pm 0.13}$ | $0.66_{\pm 0.06}$ | $0.69_{\pm 0.03}$ | 1.21 |
| SRKL | $T$ | - | $0.06_{\pm 0.01}$ | $0.04_{\pm 0.01}$ | $0.18_{\pm 0.02}$ | $0.03_{\pm 0.00}$ | $0.03_{\pm 0.00}$ | 0.07 |
| | $U$ | - | $24.53_{\pm 0.21}$ | $12.19_{\pm 0.29}$ | $15.63_{\pm 0.22}$ | $23.37_{\pm 0.27}$ | $24.28_{\pm 0.18}$ | 20.00 |
| | $S$ | - | $0.64_{\pm 0.04}$ | $0.49_{\pm 0.05}$ | $0.94_{\pm 0.08}$ | $0.53_{\pm 0.02}$ | $0.44_{\pm 0.00}$ | 0.61 |
| CSD (Ours) | $T$ | $T$ | $6.82_{\pm 0.16}$ | $4.24_{\pm 0.12}$ | $9.16_{\pm 0.25}$ | $4.53_{\pm 0.02}$ | $4.83_{\pm 0.02}$ | 5.91 |
| | $U$ | $U$ | $17.21_{\pm 0.30}$ | $8.08_{\pm 0.39}$ | $14.27_{\pm 0.40}$ | $13.19_{\pm 0.27}$ | $14.07_{\pm 0.04}$ | 13.37 |
| | $S$ | $S$ | $\mathbf{24.94}_{\pm 0.29}$ | $12.06_{\pm 0.46}$ | $15.78_{\pm 0.49}$ | $24.60_{\pm 0.31}$ | $\mathbf{25.88}_{\pm 0.13}$ | $\mathbf{20.65}$ |
| | $U$ | $S$ | $24.15_{\pm 0.55}$ | $\mathbf{12.25}_{\pm 0.47}$ | $15.25_{\pm 0.41}$ | $22.55_{\pm 0.09}$ | $25.19_{\pm 0.12}$ | 19.88 |
| | $T$ | $S$ | $22.77_{\pm 0.25}$ | $10.62_{\pm 0.32}$ | $14.06_{\pm 0.25}$ | $18.81_{\pm 0.40}$ | $21.71_{\pm 0.18}$ | 17.59 |

# E  THE USE OF LARGE LANGUAGE MODELS

In this work, LLMs were used only for minor writing assistance, such as grammar correction after drafting. In addition, since the research topic is LLM distillation, LLMs were employed as the subject of experiments and also as evaluation models for performance assessment.

Table 9: Distillation configuration, memory usage, and training speed for each teacher–student pair and distillation method. All measurements were obtained using a single A100 GPU.

| | | Teacher → Student | | | | |
| --- | --- | --- | --- | --- | --- | --- |
| | | GPT-2 1.5B → 0.1B | GPT-2 1.5B → 0.3B | OpenLlama 7B → 3B | Qwen2.5-IT 7B → 1.5B | Gemma2-IT 9B → 2B |
| **Configuration** | | | | | | |
| | Vocab | 50,257 | 50,257 | 32,000 | 151,665 | 256,000 |
| | Max sequence len. (prompt len.) | 512 (256) | 512 (256) | 512 (256) | 1024 (512) | 1024 (512) |
| | BatchSize (microbatch × accum.) | 32 | 32 | 32 | 32 (2 × 16) | 32 (1 × 32) |
| | LoRA | ✗ | ✗ | ✓ | ✓ | ✓ |
| **Efficiency (memory & training speed)** | | | | | | |
| DLD | Memory (MB) (↓) | 30489.98 | 50656.78 | 35341.80 | 45134.24 | 46203.35 |
| | Elapsed Time (sec / batch) (↓) | 0.758 | 1.033 | 4.035 | 26.62 | 44.50 |
| SKL | Memory (MB) (↓) | 39129.84 | 60082.85 | 41542.05 | 52234.71 | 52196.50 |
| | Elapsed Time (sec / batch) (↓) | 0.803 | 1.094 | 4.077 | 27.88 | 43.92 |
| KL | Memory (MB) (↓) | 32845.77 | 49870.58 | 35041.99 | 38032.67 | 40208.22 |
| | Elapsed Time (sec / batch) (↓) | 0.770 | 1.027 | 4.044 | 25.84 | 43.34 |
| CSD (Anal.) | Memory (MB) (↓) | 28919.70 | 49085.38 | 34542.05 | 42766.25 | 44205.31 |
| | Elapsed Time (sec / batch) (↓) | 0.764 | 1.033 | 4.041 | 26.64 | 42.12 |
| CSD (MC) | Memory (MB) (↓) | 30490.10 | 50656.91 | 35341.92 | 47502.30 | 48201.43 |
| | Elapsed Time (sec / batch) (↓) | 0.789 | 1.061 | 4.063 | 27.78 | 43.06 |

Table 10: Task-specific distillation performance from the `Gemma-7B-IT` teacher to the `Gemma-2B-IT` student. Please refer to Section A.4 for DLD variants details.

| | Summarization | Translation | GSM8K |
| --- | --- | --- | --- |
| Loss | ROUGE-L | COMET | Accuracy |
| Teacher | 37.09 | 79.23 | 60.27 |
| DLD (T) | 0.00 | 19.00 | 0.00 |
| DLD (U) | 0.00 | 18.98 | 0.00 |
| DLD (S) | 0.00 | 21.52 | 0.00 |
| DLD-min (T) | 0.46 | 48.05 | 0.00 |
| DLD-min (U) | 13.29 | 53.81 | 0.00 |
| DLD-min (S) | 15.91 | 52.98 | 0.00 |
| DLD-max (T) | 32.54 | 65.28 | 17.74 |
| DLD-max (U) | 15.75 | 24.23 | 0.00 |
| DLD-max (S) | 18.72 | 60.56 | 0.00 |
| DLD-std (T) | 0.00 | 35.71 | 0.00 |
| DLD-std (U) | 0.85 | 35.29 | 0.00 |
| DLD-std (S) | 18.97 | 58.07 | 0.00 |
| DLD-mean (T) | 0.00 | 34.21 | 0.00 |
| DLD-mean (S) | 18.78 | 43.98 | 0.00 |
| DLD-mean (U) | 0.03 | 32.03 | 0.00 |
| CSD $(T, S)$ | **35.67** | **74.14** | **25.78** |

Table 11: Qualitative comparison on the GSM8K dataset. Only **CSD (Ours)** produces the correct final answer; other students give incorrect results.

---

**Prompt for model:**

Q: A robe takes 2 bolts of blue fiber and half that much white fiber. How many bolts in total does it take? A:

**Correct Answer:**

It takes 2/2=«2/2=1»1 bolt of white fiber

So the total amount of fabric is 2+1=«2+1=3»3 bolts of fabric

#### 3

**Teacher sampled output:**

The robe takes 2 x 1/2 = «2*1/2=1»1 bolt of white fiber.

So it takes 2 + 1 = «2+1=3»3 bolts in total.

#### 3

---

**CSD (ours) student sampled output:**

The robe takes 2 x 1/2 = «2*1/2=1»1 bolt of white fiber.

So, it takes 2 + 1 = «2+1=3»3 bolts in total.

#### 3 (✓)

**KL student sampled output:**

2 bolts of blue fiber + 1 bolt of white fiber = «2+1=3»3 bolts in total

3 bolts of blue fiber + 2 bolts of white fiber = «3+2=5»5 bolts in total

#### 5 (✗)

**Jeffrey student sampled output:**

The robe takes 2 x 2 = «2*2=4»4 bolts of blue fiber.

The robe takes 4 / 2 = «4/2=2»2 bolts of white fiber.

The robe takes 4 + 2 = «4+2=6»6 bolts in total.

#### 6 (✗)

**JS student sampled_output:**

The robe takes 2*2=«2*2=4»4 bolts of blue fiber.

The robe takes 4/2=«4/2=2»2 bolts of white fiber.

The robe takes 4+2=«4+2=6»6 bolts in total.

#### 6 (✗)

---

Table 12: Qualitative comparison on a GSM8K dataset. The SRKL student's sampled output shows degenerative repetition and drifting arithmetic reasoning.

---

**Prompt for model:**

Q: A robe takes 2 bolts of blue fiber and half that much white fiber. How many bolts in total does it take? A:

**Correct Answer:**

It takes 2/2=«2/2=1»1 bolt of white fiber

So the total amount of fabric is 2+1=«2+1=3»3 bolts of fabric

#### 3

---

**SRKL student sampled output:**

The robe takes 2/2=1/2 bolts of white fiber.

So, it takes 1/2*2=1 bolt of blue fiber.

So, it takes 1+2=3 bolts in total.

So, it takes 3*1=3 bolts in total.

So, it takes 3*2=6 bolts in total.

So, it takes 6+1=7 bolts in total.

So, it takes 7*2=14 bolts in total.

So, it takes 14+3=17 bolts in total.

So, it takes 17*2=34 bolts in total.

So, it takes 34+1=35 bolts in total.

So, it takes 35 bolts in total.

So, it takes 35*2=70 bolts in total.

So, it takes 70+1=71 bolts in total.

So, it takes 71 bolts in total.

So, it takes 71*2=142 bolts in total.

So, it takes 142+3=145 bolts in total.

So, it takes 145*2=290 bolts in total.

So, it takes 290+1=291 bolts in total.

So, it takes 291 bolts in total.

So, it takes 291*2=582 bolts in total.

So, it takes 582 bolts in total.

So, it takes 582*2=1164 bolts in total.

So, it takes 1164 bolts in total.

So, it takes 1164*2=2328 bolts in total.

So, it takes 2328 bolts in total.

So, it takes 2328*2=4656 bolts in total.

So, it takes 4656 bolts in total.

So, it takes 4656*2=9312 bolts in total.

So, it takes 9312 bolts in total.

So, it takes 9312*2=18624 bolts in total.

So, it takes 18624 bolts

---

Table 13: Qualitative comparison on a GSM8K dataset. The TV student's sampled output shows degenerative repetition and drifting arithmetic reasoning.

---

**Prompt for model:**

Q: A robe takes 2 bolts of blue fiber and half that much white fiber. How many bolts in total does it take? A:

**Correct Answer:**

It takes 2/2=«2/2=1»1 bolt of white fiber

So the total amount of fabric is 2+1=«2+1=3»3 bolts of fabric

#### 3

---

**TV student sampled output:**

The robe takes 2 x .5 = 1 bolt of white fiber.

So it takes 2 + 1 = 3 bolts in total.

The robe takes 3 x 2 = 6 bolts in total.

So it takes 6 bolts in total.

The robe takes 6 bolts in total.

So it takes 6 bolts in total.

The robe takes 6 bolts in total.

So it takes 6 bolts in total.

The robe takes 6 bolts in total.

So it takes 6 bolts in total.

The robe takes 6 bolts in total.

So it takes 6 bolts in total.

The robe takes 6 bolts in total.

So it takes 6 bolts in total.

The robe takes 6 bolts in total.

So it takes 6 bolts in total.

... (repeats) ...

The robe takes 6 bolts in total.

So it takes 6 bolts in total.

The robe takes 6 bolts in total.

So it takes 6 bolts in total.

The robe takes 6 bolts in total.

So it takes 6 bolts in total.

The robe takes 6 bolts in total.

So it takes 6 bolts in total.

The robe takes 6 bolts in total.

So it takes 6 bolts in total.

The robe takes 6 bolts in total.

So it takes 6 bolts in total.

The robe takes 6 bolts in total.

So it takes 6 bolts in total.

The robe takes 6 bolts in total.

So it takes 6 bolts in total.

---

Table 14: Qualitative comparison on a GSM8K dataset. The SKL student's sampled output shows degenerative repetition and drifting arithmetic reasoning.

---

**Prompt for model:**

Q: A robe takes 2 bolts of blue fiber and half that much white fiber. How many bolts in total does it take? A:

**Correct Answer:**

It takes 2/2=«2/2=1»1 bolt of white fiber

So the total amount of fabric is 2+1=«2+1=3»3 bolts of fabric

#### 3

---

**SKL student sampled output:**

The robe takes 2/2=1/2 bolts of white fiber.

So it takes 2+1/2=3/2 bolts of fiber in total.

So it takes 3/2=1.5 bolts of fiber in total.

So it takes 1.5*2=3 bolts of fiber in total.

So it takes 3+1=4 bolts of fiber in total.

So it takes 4*2=8 bolts of fiber in total.

So it takes 8+1=9 bolts of fiber in total.

So it takes 9+1=10 bolts of fiber in total.

So it takes 10+2=12 bolts of fiber in total.

So it takes 12+1=13 bolts of fiber in total.

So it takes 13+1=14 bolts of fiber in total.

So it takes 14+4=18 bolts of fiber in total.

So it takes 18+1=19 bolts of fiber in total.

So it takes 19+1=20 bolts of fiber in total.

So it takes 20+2=22 bolts of fiber in total.

So it takes 22+1=23 bolts of fiber in total.

So it takes 23+1=24 bolts of fiber in total.

So it takes 24+1=25 bolts of fiber in total.

So it takes 25+1=26 bolts of fiber in total.

So it takes 26+1=27 bolts of fiber in total.

So it takes 27+1=28 bolts of fiber in total.

So it takes 28+1=29 bolts of fiber in total.

So it takes 29+1=30 bolts of fiber in total.

So it takes 30+1=31 bolts of fiber in total.

So it takes 31+1=32 bolts of fiber in total.

So it takes 32+1=33 bolts of fiber in total.

So it takes 33+1=34 bolts of fiber in total.

So it takes 34+1=35 bolts of fiber in total.

So it takes 35+1=36 bolts of fiber in total.

---

Table 15: Qualitative comparison on a GSM8K dataset. The DLD (S) student's sampled output shows degenerative repetition and drifting arithmetic reasoning.

---

**Prompt for model:**

Q: A robe takes 2 bolts of blue fiber and half that much white fiber. How many bolts in total does it take? A:

**Correct Answer:**

It takes 2/2=«2/2=1»1 bolt of white fiber

So the total amount of fabric is 2+1=«2+1=3»3 bolts of fabric

#### 3

---

**DLD (S) student sampled output:**

reluct reluct reluct reluct reluct reluct
reluct reluct reluct reluct reluct reluct
reluct reluct reluct reluct reluct reluct
reluct reluct reluct reluct reluct reluct
reluct reluct reluct reluct reluct reluct
reluct reluct reluct reluct reluct reluct
reluct reluct reluct reluct reluct reluct
reluct reluct reluct reluct reluct reluct
reluct reluct reluct reluct reluct reluct
reluct reluct reluct reluct reluct reluct
... (repeats) ...
reluct reluct reluct reluct reluct reluct
reluct reluct reluct reluct reluct reluct
reluct reluct reluct reluct reluct reluct
reluct reluct reluct reluct reluct reluct
reluct reluct reluct reluct reluct reluct
reluct reluct reluct reluct reluct reluct
reluct reluct reluct reluct reluct reluct
reluct reluct reluct reluct reluct reluct
reluct reluct reluct reluct reluct reluct
reluct reluct reluct reluct reluct reluct
reluct reluct reluct reluct reluct reluct
reluct reluct reluct reluct reluct reluct
reluct reluct reluct reluct reluct reluct
reluct reluct reluct reluct reluct reluct

---

