# OpenReview forum: "Distillation of Large Language Models via Concrete Score Matching"
_ICLR.cc/2026/Conference — ICLR 2026 Poster_

### Official Review · Reviewer_5noZ · 2025-10-31

**Soundness:** 2
**Presentation:** 4
**Contribution:** 3
**Rating:** 4
**Confidence:** 4

**Summary:**

The paper introduces Concrete Score Distillation (CSD), a novel training objective for improved large language model's knowledge distillation. Based on the observation that softmax is invariant to constant addition, the proposed method extends direct logit distillation (DLD) by allowing student's logits to be matched to the teacher's logits shifted by an arbitrary constant. The paper further introduces and experiments with various weighting functions to balance the output accuracy and diversity of the trained student. Experiments show that the proposed method outperforms prior approaches compared to divergence-based loss function, and that it achieves complementary gains when used in combination with recent on-policy KD methods.

**Strengths:**

1. The paper, including the algorithm, theoretical analyses and proofs, is well-written and easy to follow.
1. The idea of enhancing logit matching distillation by accounting for the logit shift invariance is well-motivated and reasonable.
1. The author's extension to DLD theoretically and empirically achieves the target of logit shift invariance.
1. The experimental results show improvements compared to the baselines, across multiple model families and model sizes.
1. The paper is well placed in existing literature, and prior works considered are thorough.

**Weaknesses:**

1. The authors claim that one of their motivations is that vanilla KLD gives almost identical and very low probabilities and gradients to the low-probability tokens in the tail. While their method, with uniform weighting (w1,w2=U) would indeed achieve their target of "overcoming the softmax-induced smoothing of teacher knowledge", most of their results use w1,w2 as either student or teacher probabilities. As these values are very small and almost zero, a very similar smoothing is also present in the proposed method.
1. Comparison with DLD: since the method is an extension to DLD, I believe it would be more appropriate to compare the method with variants of logit matching methods. Currently, only the comparison between DLD and CSD on GPT2-0.1B for the five benchmarks is given. It would be great to see how CSD compares against DLD at larger model sizes, different datasets, etc.
1. Weighting function: not using weighting function (corresponding to the (u, u) setting in the paper) performs significantly worse compared to other divergence-based baselines, and the overall performance seems to be heavily impacted by the choice of the weighting function. Furthermore, with S as a weighing function, DLD performance seems to be only slightly worse than the proposed method. This makes me wonder if similar performance improvement can be made to the divergence-based methods by adding similar weighting function to those.
1. The method is only tested on relatively small datasets (eg. 15000 samples), its effectiveness in longer training horizon is unexplored.
1. Most of the experiments were performed on a relatively weak model (GPT2), with only some experiments on Openllama/gemma. It is uncertain if the results will hold with newer/stronger models.

**Questions:**

1. In figure 4, "Ours" refers to which exact model/setting for the two model sizes? Just CSD, or perhaps ImitKD+Ours, or something else? Could the authors share the GPT evaluation scores of vanilla CSD, GKD+ours, DistilLLM+ours,ImitKD+ours if these were also evaluated?
1. For results in Table 3, w1,w2 were used as (t,s) (as mentioned in line 879) compared to the "default"(as mentioned in line 312) of (s,s) elsewhere. Can the authors elaborate on why were w1,w2 changed for these experiments? If the authors tried (s,s) first, could they share these results?
1. In figure 8 (a) and (b), could the authors also share the KL-divergence/MSE/some other divergence metric between the teacher and student probabilities corresponding to y1 to y10? This will prove that the student probabilities with CSD maintain similarity to teacher inspite of logit shift.

---

> ### Author Response · Authors · 2025-11-21
> **Official Comment by Authors for Reviewer 5noZ (1/3)**
>
> We sincerely appreciate **reviewer 5noZ** for the helpful feedback. Thank you for the reviewer’s recognition of **writing quality**, **motivation regarding logit shift invariance**, **theoretical guarantee**, and **empirical performance**. We address the reviewer’s comments below.
>
> &nbsp;
>
> ---
>
> ### **Weakness 1 (Mitigation on the softmax smoothing)**
>
> *Q. The authors claim that one of their motivations is that vanilla KLD gives almost identical and very low probabilities and gradients to the low-probability tokens in the tail. While their method, with uniform weighting (w1,w2=U) would indeed achieve their target of "overcoming the softmax-induced smoothing of teacher knowledge", most of their results use w1,w2 as either student or teacher probabilities. As these values are very small and almost zero, a very similar smoothing is also present in the proposed method?*
>
> &nbsp;
>
> ### **Response to Weakness 1.**
>
> Unlike KL's softmax normalization, where smoothing is built-in, our weighting function $w$ is designed to avoid making smoothing an inherent part of the method. As you pointed out, when w1 and w2 are set to the student or teacher probabilities, the gradients tend to concentrate on a small subset of vocabulary tokens, similar to the KL divergence. However, this configuration was intentionally designed because it specializes in competing with other divergence-based methods that primarily focus on fidelity, such as Rouge-L performance. It is not the only instance we consider.
>
> Importantly, we also provide other instances that maintain good fidelity, significantly better than KL, while effectively mitigating softmax-induced smoothing. For example, CSD (us) produces much more diverse gradient coefficients across the vocabulary, as shown in **Figure 13** of Section D.3. The table below shows the gradient diversity across the vocab items at initialization and the end of training. Despite this improved diversity, it still achieves a substantially higher Rouge-L score (19.88) compared to KL (16.45).
> | Loss| Initialized | After training|
> |---------------|:------------:|:------------:|
> | KL | 2.61   | 1.67   |
> | CSD (us)| **6.96**  | **7.26**   |
>
> This advantage also translates into better trade-offs in temperature-adjustment scenarios, as demonstrated in **Figure 3b**. Under high-temperature settings, where minority vocabulary items have more opportunity to contribute, CSD (us) shows significantly stronger performance. This is because minority vocabulary items have been learned more uniformly. More detailed information can be found in the discussion of **Section D.3**.
>
>
> &nbsp;
>
> ---
>
> ### **Weakness 2, 4, 5 (Compare to DLD on stronger models)**
>
> *Q. Comparison with DLD: since the method is an extension to DLD, I believe it would be more appropriate to compare the method with variants of logit matching methods. Currently, only the comparison between DLD and CSD on GPT2-0.1B for the five benchmarks is given. It would be great to see how CSD compares against DLD at larger model sizes, different datasets, etc. // The method is only tested on relatively small datasets (eg. 15000 samples), its effectiveness in longer training horizon is unexplored. // Most of the experiments were performed on a relatively weak model (GPT2), with only some experiments on Openllama/gemma. It is uncertain if the results will hold with newer/stronger models.*
>
> &nbsp;
>
> ### **Response to Weakness 2, 4, 5.**
>
> We provide comprehensive experimental results comparing our method to DLD.
>
> First, we applied DLD to multiple student backbones (GPT-2-0.1B, GPT-2-0.3B, OpenLlama-3B) and datasets (on-policy, mix, adaptive) in the benchmark experiments presented in **Table 2**. Across all average scores, CSD consistently outperformed DLD. We present averaged Rouge-L scores as a summary below.
> | Loss|Student| Mix | Ada |On |
> |-----|:---------:|:---------:|:---------:|:-:|
> | DLD (S)   | GPT-2-0.1B |19.58      | 19.23      | 19.81     |
> | CSD (SS)  |  GPT-2-0.1B|**19.81**      | **20.39**       | **20.73**      |
> ||||||
> | DLD (S) |  GPT-2-0.3B|21.55      | 20.65      | 22.32      |
> | CSD (SS) |  GPT-2-0.3B|**22.69**      | **22.72**     | **23.12**    |
> ||||||
> | DLD (S) |  OpenLlama-3B|-     | -     |  28.24    |
> | DLD-mean (S) |  OpenLlama-3B|-     | -     |  27.79    |
> | CSD (SS) |  OpenLlama-3B|-     | -     | **29.03**    |

---

> ### Author Response · Authors · 2025-11-21
> **Official Comment by Authors for Reviewer 5noZ (2/3)**
>
> ### **Response to Weakness 2, 4, 5. - Continued**
>
> Second, we added DLD to task-specific distillation in **Table 3** using Gemma-7B-IT teacher. This task-specific setting is particularly challenging because each task uses only 1,000 training examples. Under this condition, the restricted solution set of DLD leads to even more severe behavior, resulting in significantly degraded performance across all tasks.
>
> | Loss          | Summarization $\uparrow$| Translation$\uparrow$ | GSM8K$\uparrow$ |
> |---------------|:--------------------------:|:----------------------:|:-------------------:|
> | DLD (S)       | 0.00                     | 21.52               | 0.00              |
> | DLD-max (T)   | 32.54                    | 65.28               | 17.74              |
> | CSD (T, S)    | **35.67**                | **74.14**           | **25.78**
>
> Third, we present new distillation experiments in **Table 4**, using recent strong teacher models (Qwen2.5-7B-IT, Gemma2-9B-IT) trained on the UltraChat-50K dataset. In these experiments, CSD outperforms both the recent divergence-based method (DistillLM2) and DLD.
> || | Qwen2.5-IT (7B$\rightarrow$1.5B) | | |Gemma2-IT (9B$\rightarrow$2B) | |
> |-|:-:|:-:|:-:|:-:|:-:|:-:|
> | Benchmark| MT-Bench| MT-Bench | AlpacaEval | MT-Bench | MT-Bench| AlpacaEval |
> | Judge| GPT4| GPT4-Turbo | GPT4-Turbo | GPT4| GPT4-Turbo | GPT4-Turbo |
> | | |  |  | |  |  |
> | Teacher | 8.59 | 7.52 | 88.69 | 8.91 | 7.66 | 94.60 |
> | | |  |  | |  |  |
> | DPKD (Li et al., 2024) | 1.04 | 1.09 | 0.32 | 6.30 | 4.89 | 71.18 |
> | DistiLLM2 (Ko et al., 2025) | 7.28 | 5.75 | **70.42** | 7.81 | 6.45 | 89.91 |
> | DLD (T) | 7.25 | 5.56 | 69.80 | 5.85 | 4.45 | 31.24 |
> | DLD (S) | 7.28 | 5.74 | 67.67 | 7.58 | 6.53 | 89.84 |
> | **CSD (T, S)** | 7.42 | 5.90 | **70.42** | **7.85** | **6.55** | 89.92 |
> | **CSD (S, S)** | **7.69** | **5.95** | 69.64 | 7.77 | 6.43 | **90.05** |
>
> Finally, in **Table 5**, we compare CSD to variants of DLD, as defined in Section A.4. CSD continues to outperform all DLD variants. Below is a summary table comparing each method’s best weighting.
> | Loss        | Avg. Rouge-L $\uparrow$ |
> |--|:-:|
> | DLD (S)       | 20.39       |
> | DLD-min (S)    | 20.02       |
> | DLD-max (S)   | 9.49        |
> | DLD-std (U)     | 17.51       |
> | DLD-range (U) | 9.80        |
> | CSD (S, S)  | **20.65**       |
>
> Some of these variants are also present in **Table 2** and **Table 3**, and **Table 10** provides a comprehensive evaluation of DLD variants in task-specific settings.
>
> ----
>
>
> ### **Weakness 3 (Weighting for divergence method)**
>
> *Q. Weighting function: not using weighting function (corresponding to the (u, u) setting in the paper) performs significantly worse compared to other divergence-based baselines, and the overall performance seems to be heavily impacted by the choice of the weighting function. Furthermore, with S as a weighing function, DLD performance seems to be only slightly worse than the proposed method. This makes me wonder if similar performance improvement can be made to the divergence-based methods by adding similar weighting function to those.*
>
> &nbsp;
>
> ### **Response to Weakness 3.**
>
> Unlike CSD or DLD, which are based on an L2 formulation, traditional divergence-based losses can exhibit different notions of optimality depending on the choice of weighting (See **Section A.5**, where we show that applying weighting to divergence losses induces a different form of optimality). As a result, the model distribution may become biased toward an unintended target. L2 can maintain its optimality even under different weightings, as shown in Proposition 1. **This is one of the other reasons why L2 provides a better design space for loss functions compared to divergence-based methods.** **Table 8** of Section D shows that using teacher or student weighting instead of uniform weighting to KL, TV, and SRKL. The results produced either negligible improvements or significant performance degradation. We also evaluated an L2 loss defined over probabilities, but in every case, it failed to outperform CSD. We provide the results summary in the table below.
> | Loss        | Avg. Rouge-L $\uparrow$ |
> |--|:-:|
> | Prob L2 (T)       | 19.98       |
> | Prob L2 (U)     | 11.35      |
> | Prob L2 (S)    | 11.13        |
> | KL (T)     | 16.99       |
> | KL (U)   | 16.68       |
> | KL (S)    | 17.00       |
> | TV (T)     | 20.07       |
> | TV (U)   | 19.97        |
> | TV (S)    | 1.21       |
> | SRKL (T)     | 0.07       |
> | SRKL (U)     | 20.00       |
> | SRKL (S)      | 0.61       |
> | CSD (S,S)      | **20.65**       |

---

> ### Author Response · Authors · 2025-11-21
> **Official Comment by Authors for Reviewer 5noZ (3/3)**
>
> ### **Question 1 (GPT score)**
>
> *Q. In figure 4, "Ours" refers to which exact model/setting for the two model sizes? Just CSD, or perhaps ImitKD+Ours, or something else? Could the authors share the GPT evaluation scores of vanilla CSD, GKD+ours, DistilLLM+ours,ImitKD+ours if these were also evaluated?*
>
> ### **Response to Question 1.**
>
> For the GPT-2-0.1B model, we evaluated only the ImitKD + CSD combination, since it achieved the best Rouge-L performance. For GPT-2-0.3B, ImitKD + Ours and DistiLLM showed comparable performance, so we additionally evaluated GKD + Ours and DistiLLM + Ours. Among these, DistiLLM + CSD performed the best, and we report that result in the main text. In all cases, our method provided orthogonal performance gains. The detailed results are as follows:
>
> ||**ImitKD**|**GKD**|**DistiLLM**|
> |-|:-:|:-:|:-:|
> | **Naive**| 0.4977 | 0.4963 | 0.5294 |
> | **+CSD**| **0.5389** | **0.5264** |**0.5552** |
>
>
> &nbsp;
>
> ---
>
>
> ### **Question 2 ( (s, s) weighing for task-specific distillation)**
>
> *Q. For results in Table 3, w1,w2 were used as (t,s) (as mentioned in line 879) compared to the "default"(as mentioned in line 312) of (s,s) elsewhere. Can the authors elaborate on why were w1,w2 changed for these experiments? If the authors tried (s,s) first, could they share these results?*
>
> &nbsp;
>
> ### **Response to Question 2.**
>
> The default $(s,s)$ weighting we used was originally defined for the instruction-following task in Section 4.1, and it was designed to mimic the behavior of mode-seeking divergences, which performed well in those experiments. Because this task-specific distillation setting uses only 1,000 training examples, it presents a highly challenging optimization landscape. In such cases, mode-seeking behavior often leads to failure, likely because it makes the model more prone to getting stuck in poor local minima. For this reason, even RKL performed extremely poorly, as shown in **Table 3**. Similarly, CSD (ss) also exhibited very poor results, reaching only (21.09, 63.78, 0.00) for (summarization, Translation, GSM8K). We added relevant discussions to the **lines 448-453** of the modified manuscript.
>
> Rather than proposing a single instance of a loss function, our paper introduces a flexible design space that allows losses to be adapted to a variety of scenarios. This design space enables us to identify loss formulations that perform well even under specific conditions, ultimately leading to strong empirical results.
>
>
> &nbsp;
>
> ---
>
> ### **Question 3 (Probability matching measurement for each logits)**
>
> *Q. In figure 8 (a) and (b), could the authors also share the KL-divergence/MSE/some other divergence metric between the teacher and student probabilities corresponding to y1 to y10? This will prove that the student probabilities with CSD maintain similarity to teacher inspite of logit shift*
>
> &nbsp;
>
> ### **Response to Question 3.**
>
> In **Figure 9** of our revised manuscript, we newly plot the logits of DLD and CSD under various weighting settings. **Table 6** presents the corresponding KL, MSE, and ECE values for these logits. Furthermore, **Figure 10** provides a more systematic comparison by averaging these metrics across multiple sequences and tokens for each training epoch.
>
> The results for generative performance and probability calibration metrics show contrasting trends. CSD (T, S) achieves the best calibration scores because the probabilities of specific vocabulary items (those assigned high probability by either the student or the teacher) are more important in the calibration score. CSD (T, S) benefits from leveraging probability weighting from both models. In contrast, generative performance is highest with CSD (S, S), because generative performance depends on the quality of outputs produced within the student model’s high-likelihood region. As a result, calibration performance does not necessarily correlate with generative performance [D1, D2]. This further illustrates that our loss design space can flexibly adapt to different perspectives (generative fidelity vs. probability calibration) depending on the choice of $w$. Please refer to **Section D.1** for the detailed discussion.
>
>
> [D1] Gpt-4 technical report, OpenAI
>
> [D2] [ICLR 2024] Beyond Reverse KL: Generalizing Direct Preference Optimization with Diverse Divergence Constraints
>
> ------
>
> Thanks again for taking the time to share your thoughtful feedback. I’m happy to answer any further questions you may have.

---

> > ### Comment · Reviewer_5noZ · 2025-11-24
> > **Rebuttal Acknowledgement**
> >
> > Thank you for the rebuttal.
> >
> > - Figure 10/11 clearly demonstrate that logit-shift invariance is being achieved without sacrificing MSE/divergence, and further comparisons in Table 10 highlight the superiority of this method over DLD.
> > - I also appreciate the comparisons of applying CSD-style extensions to divergence-based metrics, as well as the analysis in A5 that such extensions will necessarily bias the convergence optima.
> > - Experiments on Ultrachat with Qwen/Gemma teacher/student, evaluated on MT-Bench and AlpacaEval, are appreciated, and clearly show the method is equally applicable with newer models and IF datasets.
> >
> > I have hence increased my score.
> >
> > **For Table 4, can the authors share applying vanilla KL divergence to either of these two mdoels if possible?**
> >
> > **Regarding Softmax Smoothing**
> >
> > Figure 13 does indeed show that except for csd (us), csd variants (ss, ts) have similar or worse gradient diversity than KL divergence, and suffer from the same "softmax-induced smoothing" that the authors wanted to fix. The authors say that they address this by instead proposing a "flexible design space", specifically the choice of w1 and w2 as hyper-params, which leads me to my next concern.
> >
> > **Difficult to choose the right combination of w1/w2**
> >
> > The proposed method introduces the choice of w1 and w2 as hyper-params, and the right choice of these is difficult. u will usually give bad performance in hard tasks, s as w1 will be unstable, and PO/RLHF-inspired losses usually worsen calibration.
> >
> > - t,t - significantly worse in Table 5
> > - u,u - significantly worse in Table 5
> > - u,s - slightly worse in Table 5, good in figure 3b, bad calibration, good in Figure 13
> > - s,s - best in table 5, similar to ts in Table 4, diverged in Table 3, bad calibration/ECE, bad in Figure 13.
> > - t,s - best in table 10, worse in Table 4, good calibration/ECE, medium in Figure 13
> >
> > The authors also explore an adaptive strategy for these, which seems reasonable and well-motivated.
> >
> > I agree the proposed framework gives flexibility. The problem is unstable/inconsistent performance, with bad calibration, for any given choice from this space (except possibly for t,s).
> >
> > The above concerns prevent me from increasing my score further.

---

> ### Author Response · Authors · 2025-11-29
> **Additional Official Comment by Authors for Reviewer 5noZ (1/3)**
>
> We sincerely appreciate **reviewer 5noZ** for taking the time to read all our responses and the updated manuscript. Thank you for acknowledging that (i) CSD simultaneously achieves logit-shift invariance and probability matching, (ii) in contrast to CSD, divergence-based objectives inevitably introduce bias depending on the choice of weighting functions, (iii) CSD scales effectively to newer datasets and more powerful models, (iv) our loss design space provides the desired flexibility, and (v) the additional adaptive strategy within this design space is properly examined. Accordingly, we also noticed that you had positively updated your review score based on our previous responses. (It appears that the system has now reverted to the initial review state.)
>
> ```
> Soundness: 2 (fair) → 3 (good)
> Rating: 4 (marginally below the acceptance) → 6 (marginally above the acceptance)
> ```
>
> We provide our responses to the reviewer's additional questions below.
>
> ---
>
> $\quad$
>
> ### **Additional Question 1 (KL divergence baseline on large scale experiments)**
>
> $\quad$
>
> ### **Response to Additional Question 1**
>
> A. We present the vanilla KL-divergence results on Qwen2.5-7B-IT $\rightarrow$ Qwen2.5-1.5B-IT. To further offer a comparison across different student strengths, we also include results from a student model obtained by supervised fine-tuned on the Qwen2.5-1.5B-Base model. In both cases, CSD performs better than KL on various evaluation metrics.
>
> || | Qwen2.5-IT-7B $\rightarrow$ Qwen2.5-Base-1.5B-SFT | | | Qwen2.5-IT-7B $\rightarrow$ Qwen2.5-IT-1.5B | |
> |-|:-:|:-:|:-:|:-:|:-:|:-:|
> | Benchmark| MT-Bench| MT-Bench | AlpacaEval | MT-Bench | MT-Bench| AlpacaEval |
> | Judge| GPT4| GPT4-Turbo | GPT4-Turbo | GPT4| GPT4-Turbo | GPT4-Turbo |
> | | |  |  | |  |  |
> | Teacher | 8.59 | 7.52 | 88.69 | 8.59 | 7.52 | 88.69 |
> | | |  |  | |  |  |
> | Student | 5.83 | 4.56 | 38.15 | 6.99| 5.21| 57.67|
> | KL | 6.94| 5.62| 67.80| 7.45| 5.89| **70.47**|
> | DistiLLM2 | 7.13| 5.59| 67.74| 7.28 | 5.75 | 70.42 |
> | DLD (T) | 6.71| 5.01| 67.07| 7.25 | 5.56 | 69.80 |
> | DLD (S) | 6.62| 5.23| 66.32| 7.28 | 5.74 | 67.67 |
> | **CSD (T, S) (Ours)** | 7.12| 5.44| 67.97| 7.42 | 5.90 | 70.42 |
> | **CSD (S, S) (Ours)** | **7.54**| **6.02**| **72.46**| **7.69** | **5.95** | 69.64 |

---

> ### Author Response · Authors · 2025-11-29
> **Additional Official Comment by Authors for Reviewer 5noZ (2/3)**
>
> ### **Additional Question 2 (Right combination of w1/w2)**
>
> $\quad$
>
> ### **Response to Additional Question 2**
>
> Our goal is not to propose a single loss function, but to introduce a design space for loss construction. Accordingly, the main arguments of our paper, **Proposition 1**, **Theorem 2**, and **Theorem 3** are all established for a general weighting function $w$. As the reviewer acknowledged, this provides the desired flexibility.
>
> $\quad$
>
> **(General goal of loss design space research)** In research on loss design spaces, the goal is typically not to develop a single loss function that performs best across all tasks. For example, Generalized KD [E1] presents in Figure 1 that different tasks benefit from different losses: JSD (0.1) is most effective for translation, while forward KL performs better for summarization and reasoning. Similar observations appear in other work extending LLM losses. The f-divergence-based extension [E2] shows clear trade-offs between performance and diversity depending on the selected loss (Figure 3). Likewise, the Bregman divergence-based extension [E3] reports that the proper loss confidence parameter λ varies across settings, for example, λ=1.2 for in-domain dialogue generation and λ=-0.5 for instruction-following benchmarks. Taken together, these studies show that **different losses excel under different conditions**. The aim of proposing a loss design space is therefore not to identify a single universal loss, but to offer principled guidance that helps practitioners choose the most suitable loss for a given task or scenario.
>
> $\quad$
>
> **(People use task-specific models with different training loss)** We observe the same pattern in practical, production-scale systems. Instruction-following models [E4, E5] (e.g., Qwen2.5-Instruct), math-specialized models [E6, E7] (e.g., Qwen2.5-Math), and code-focused models [E8, E9] (e.g., Qwen2.5-Coder) are provided separately, all trained with different losses and training recipes. Recent Agentic AI systems [E10] also rely on such specialization, where different expert models are employed for the tasks that they perform best. This further highlights that providing situation-appropriate loss functions is important in practice.
>
> $\quad$
>
> **(We are clearly demonstrating the benefits of each instance)**
> Our instance-wise performance is not inconsistent; rather, all the behaviors noted by the reviewer can be explained by the principled characteristics summarized below. We recommend, as a default choice, setting at least one of $w_1$ or $w_2$ to the student’s probability distribution. The following summarizes the behavior observed when $w_1$ is set to alternative options:
> | $(w_1, w_2)$ | Fidelity | Calibration | Mitigates Smoothing|
> |--------|----------|-------------|--------------------|
> | $(S, S)$     | Good     | Poor        | Poor               |
> | $(U, S)$     | Middle   | Middle      | Good               |
> | $(T, S)$     | Poor     | Good        | Middle|
>
> Fidelity reflects how high-quality the samples generated are, and it typically increases when the model becomes more mode collapsed. Calibration improves when the model covers the full distributional modes while assigning accurate probabilities across all regions. The better calibration performance results in better sample diversity and stable training. The fidelity and calibration are generally known to exhibit a negative correlation [E2] (e.g., forward KL vs. reverse KL, reference models vs. RLHF-tuned models). The behaviors of $(S, S)$, $(U, S)$, and $(T, S)$ follow a gradual trade-off along this spectrum.
>
> Separately, $(U, S)$ has the desirable property of mitigating softmax smoothing, allowing minority vocabulary tokens to be learned more effectively. This becomes particularly beneficial in settings where temperature adjustment amplifies the importance of minority tokens (Figure 3.b). Because our loss design space is not constrained by softmax smoothing, it is able to capture such advantageous instances. **We use the above characteristics to clearly explain the results of the various experiments that the reviewer perceived as *inconsistent*, as summarized below.**
>
> - $(U, S)$: The reason why $(U, S)$ performs slightly worse in Table 5 is that its fidelity lies in the middle range. The strong performance in Figure 3b and Figure 13 arises from instances where softmax smoothing does not occur.
> - $(S, S)$: The strong performance of $(S, S)$ in Table 5 is due to its high fidelity, and the same effect explains its good results in Table 4. However, in Table 3, which evaluates distillation under limited training samples, $(S, S)$ diverges because its poor calibration leads to reduced training stability.
> - $(T, S)$: The strong performance of $(T, S)$ in Table 10 is due to its good calibration, which ensures stable optimization even under limited-sample distillation settings. In Table 4, its fidelity is lower, so a slight performance drop can occur.

---

> ### Author Response · Authors · 2025-11-29
> **Additional Official Comment by Authors for Reviewer 5noZ (3/3)**
>
> Many LLM distillation papers report results from only a single performance perspective. In contrast, we comprehensively evaluate the behaviors that different instances within our loss design space can exhibit, covering a wide range of conditions (so naturally different losses excel on different metrics):
>
> - Multiple student model sizes: 0.1B – 2B
> - Multiple teacher model sizes: 1.5B – 9B
> - Disillation tasks: task-agnostic instruction following, task-specific, and general chat capabilities
> - Number of distillation datasets: 1k – 50k
> - Evaluation metric: fidelity, diversity, calibration, and external benchmarks
>
>
> We believe we have transparently presented these characteristics across multiple dimensions, and **Section 4.4 (Ablation Studies — Design Choice of CSD)** provides our best effort to explain these factors in detail. To the best of our understanding, fidelity, calibration, and smoothing mitigation each represent distinct trade-offs. Our intention is to offer guidance that allows practitioners to flexibly select the most appropriate loss configuration depending on the specific scenario.
>
> ---
>
> [E1] [ICLR 2024] On-Policy Distillation of Language Models: Learning from Self-Generated Mistakes
>
> [E2] [ICLR 2024] Beyond reverse kl: Generalizing direct preference optimization with diverse divergence constraints
>
> [E3] [NeurIPS 2025] Preference Optimization by Estimating the Ratio of the Data Distribution
>
> [E4] Qwen2.5 Technical Report
>
> [E5] DeepSeek-V3 Technical Report
>
> [E6] Qwen2.5-Math Technical Report: Toward Mathematical Expert Model via Self-Improvement
>
> [E7] DeepSeekMath: Pushing the Limits of Mathematical Reasoning in Open Language Models
>
> [E8] Qwen2.5-Coder Technical Report
>
> [E9] DeepSeek-Coder: When the Large Language Model Meets Programming -- The Rise of Code Intelligence
>
> [E10] Small Language Models are the Future of Agentic AI

---

### Official Review · Reviewer_8hoT · 2025-11-02

**Soundness:** 3
**Presentation:** 3
**Contribution:** 3
**Rating:** 4
**Confidence:** 4

**Summary:**

This paper proposes Concrete Score Distillation (CSD), a knowledge-distillation objective for autoregressive LLMs that addresses two core issues: (1) probability-matching objectives (e.g., KL) lose fine-grained information at the logit level due to softmax smoothing; (2) while direct logit distillation (DLD) avoids softmax, its objective implicitly forces the student logits to match the teacher exactly, ignoring logit shift invariance (i.e., an additive constant does not change probabilities), which restricts the optimal solution set. CSD adapts score matching from energy-based modeling to a discrete setting and matches pairwise relative logit differences across the vocabulary, thereby being inherently shift-invariant and expanding the attainable solution set. The paper further addresses training instability and computational overhead when applying score matching to autoregressive LLMs by deriving an O(|V|) analytic gradient (from a naïve O(|V|2) formulation). Experiments show that CSD outperforms recent probability-matching and direct-logit objectives on both task-agnostic instruction following and task-specific distillation (summarization, math, and translation), offers a controllable fidelity–diversity trade-off, and yields complementary gains when combined with on-policy techniques, indicating good scalability.

**Strengths:**

1. Precise problem framing and strong motivation. The paper clearly identifies fundamental shortcomings of existing approaches—probability matching loses logit information; DLD over-constrains the solution set by ignoring shift invariance.
2. Theory-grounded contribution. CSD is derived from score matching with key guarantees: Proposition 1 (consistency) and Theorem 2 (solution set is a superset of DLD). This provides principled justification for the objective.
3. Clear, controllable objective design. Equation (9) reveals a gradient structure based on centering, factorized weighting, and role swapping; the (w1,w2) design smoothly tunes the fidelity–diversity trade-off (as illustrated in Figure 3).

**Weaknesses:**

1. Underpowered DLD baselines. DLD does not appear in the main comparison table and is shown only once in ablations (Table 4). Stronger DLD variants—e.g., mean-centered, temperature-scaled, or otherwise shift-aware formulations—are missing, as are several recent logit-distillation variants. This makes it hard to assess CSD’s margin over the strongest DLD baselines.
2. Practical computational burden. Although Theorem 3 reduces complexity from O(|V|2) to O(|V|), real-world LLMs with very large vocabularies (e.g., 50k+) still require computing and normalizing all token logits per step. The paper shows that a Monte-Carlo estimator performs worse, so the efficient implementation currently relies on full-vocabulary passes, which may become a bottleneck at scale.

**Questions:**

1. Stronger DLD comparisons. If DLD is augmented with mean-centering, temperature scaling, or an explicit shift-invariant formulation, does CSD still retain the advantages reported in Tables 1 and 3? Could you include strong DLD in the main results and report variance?
2. On the solution space. Since CSD allows a global additive shift on logits, how does this offset behave during training—does it converge to a stable value? Are there systematic differences across tokens? Does this flexibility introduce any extra instability compared to DLD?
3. Scaling characterization. For vocabularies of 32k–100k and sequence lengths of 4k–8k, what are the peak memory, step time, and convergence trade-offs of CSD (analytic vs. MC) versus KL/DLD? Is there a practical heuristic for choosing analytic vs. MC estimators at different scales?

---

> ### Author Response · Authors · 2025-11-21
> **Official Comment by Authors for Reviewer 8hoT (1/4)**
>
> We sincerely appreciate **reviewer 8hoT** for the valuable feedback. Thank you for the reviewer’s recognition of **strong motivation**, **theoretical strength**, and **controllable objective design**. We address the reviewer’s comments below.
>
> &nbsp;
>
> ---
>
> ### **Weakness 1 & Question 1 (Comparison to more DLD variants)**
>
> *Q. Underpowered DLD baselines. DLD does not appear in the main comparison table and is shown only once in ablations (Table 4). Stronger DLD variants—e.g., mean-centered, temperature-scaled, or otherwise shift-aware formulations—are missing, as are several recent logit-distillation variants. This makes it hard to assess CSD’s margin over the strongest DLD baselines. // Stronger DLD comparisons. If DLD is augmented with mean-centering, temperature scaling, or an explicit shift-invariant formulation, does CSD still retain the advantages reported in Tables 1 and 3? Could you include strong DLD in the main results and report variance?*
>
> &nbsp;
>
> ### **Response to Weakness 1.**
>
> Thank you for the helpful comment. In the context of LLM distillation, most prior work has focused on divergence-based objectives rather than DLD, so our comparisons primarily include divergence-based methods. We were unable to find DLD variants specifically designed for LLM distillation. The works cited in line 131 (Ba & Caruana, 2014; Urban et al., 2017) and line 136 (Kim et al., 2021) consider $w$ as a uniform weight. In fact, we generalize DLD by allowing arbitrary weighting functions, enabling us to identify more effective DLD variants as our baseline. However, in the context of our paper, we agree that it is more appropriate to make a more comprehensive comparison with DLD.
>
> **(Comparison to DLD variants)** We have conducted a thorough comparison with the DLD variants you suggested (We also describe DLD variants formulation in **Section A.4** of the modified manuscript):
> $$ L_{DLD}^{min} :=\frac{1}{2} \sum_{y_t} w(y_t) ( (f_{\theta}[y_t] - f_{\theta}^{min}) - (f_{T}[y_t] - f_{T}^{min} ))^2 $$
> $$ L_{DLD}^{max} :=\frac{1}{2} \sum_{y_t} w(y_t) ( (f_{\theta}[y_t] - f_{\theta}^{max}) - (f_{T}[y_t] - f_{T}^{max} ))^2 $$
> $$ L_{DLD}^{mean} :=\frac{1}{2} \sum_{y_t} w(y_t) ( (f_{\theta}[y_t] - f_{\theta}^{mean}) - (f_{T}[y_t] - f_{T}^{mean} ))^2 $$
> $$ L_{DLD}^{w-mean} :=\frac{1}{2} \sum_{y_t} w(y_t) ( (f_{\theta}[y_t] - f_{\theta}^{w-mean}) - (f_{T}[y_t] - f_{T}^{w-mean} ))^2 $$
> $$ f_{\theta}^{min}:= \min_{x} f_{\theta}[x], \quad f_{\theta}^{max}:= \max_{x} f_{\theta}[x], \quad f_{\theta}^{mean}:= \frac{1}{|V|}\sum_{x}f_{\theta}[x], \quad f_{\theta}^{w-mean}:= \sum_{x} w(x) f_{\theta}[x] $$
> $$ L_{DLD}^{std} :=\frac{1}{2} \sum_{y_t} w(y_t) \left( \frac{f_{\theta}[y_t] - f_{\theta}^{mean}}{f_{\theta}^{std}} - \frac{f_{T}[y_t] - f_{T}^{mean}}{f_{T}^{std}} \right)^2 $$
> $$L_{DLD}^{range} := \frac{1}{2} \sum_{y_t} w(y_t) \left( \frac{2(f_{\theta}[y_t] - f_{\theta}^{min})}{f_{\theta}^{max} - f_{\theta}^{min}} - 1- \frac{2(f_{T} [y_t] - f_{T}^{min})}{f_{T}^{max} - f_{T}^{min}} - 1  \right)^2 $$
>
> $ L_{DLD}^{min}, L_{DLD}^{max}, L_{DLD}^{mean}, L_{DLD}^{w-mean}$ are shift-aware formulations. CSD includes $L_{DLD}^{w-mean}$ as a special case when $w_1 = w_2$ (See **Remark 4** in Section A.4 of modified manuscript). Although there has been some progress on DLD in classical classification settings, these developments focus on the nature of classification rather than generative modeling. Consequently, they aim to distill the prediction ranking so that the student matches the teacher’s argmax outputs, rather than matching the full probability distribution as required in our setting, which is theoretically characterized by Proposition 1. For example, as in [C1], one can normalize both the student and teacher logits to have unit variance before matching, which preserves ranking but does not ensure probability matching. Following this idea, we also experimented with matching unit-variance normalized logits, as well as logits scaled to the range [-1,1].
>
> - **Table 5** of the modified manuscript compares DLD-min, DLD-max, DLD-std, and DLD-range with various weighting functions on GPT-2-1.5B $\rightarrow$ GPT-2-0.1B distillation. Below is a summary table comparing each method’s best weighting.
> | Loss        | Avg. Rouge-L $\uparrow$ |
> |-------------|:-:|
> | DLD (S)       | 20.39       |
> | DLD-min (S)    | 20.02       |
> | DLD-max (S)   | 9.49        |
> | DLD-std (U)     | 17.51       |
> | DLD-range (U) | 9.80        |
> | CSD (S, S)  | **20.65**       |
>
> ---
>
> [C1] [CVPR 24] Logit Standardization in Knowledge Distillation

---

> ### Author Response · Authors · 2025-11-21
> **Official Comment by Authors for Reviewer 8hoT (2/4)**
>
> ### **Response to Weakness 1. - Continued**
>
> **(Comparison to DLD variants - Continued)**
>
> - **Table 3** and **Table 10** compare DLD-min, DLD-max, DLD-std, and DLD-mean with various weighting functions on task-specific Gemma-7B-it $\rightarrow$ Gemma-2B-it distillation. Among the DLD variants, DLD-max (T) performed the best, and its comparison with CSD is shown below.
> | Loss          | Summarization $\uparrow$| Translation$\uparrow$ | GSM8K$\uparrow$ |
> |---------------|:--------------------------:|:----------------------:|:-------------------:|
> | DLD (S)       | 0.00                     | 21.52               | 0.00              |
> | DLD-max (T)   | 32.54                    | 65.28               | 17.74              |
> | CSD (T, S)    | **35.67**                | **74.14**           | **25.78**         |
>
>
> - **Table 2** compares to DLD-mean in OpenLlama-7B $\rightarrow$ OpenLlama-3B distillation.
> | Loss          | 	Avg. Rouge-L $\uparrow$|
> |---------------|:--------------------------:|
> | DLD (S)       |28.24             |
> | DLD-mean (S)   | 27.79       |
> | CSD (S, S)    | **29.03**|
>
> **(Comparison to DLD under various distillation scenarios)**
> - **Table 2** comprehensively compared the best-performing DLD loss across various student backbones (GPT-2-0.1B, GPT-2-0.3B, OpenLLama3B) and dataset settings (on-policy, mix, adaptive). The table below summarizes the averaged Rouge-L scores.
> | Loss|Student| Mix | Ada |On |
> |-----|:---------:|:---------:|:---------:|:-:|
> | DLD (S)   | GPT-2-0.1B |19.58      | 19.23      | 19.81     |
> | CSD (SS)  |  GPT-2-0.1B|**19.81**      | **20.39**       | **20.73**      |
> ||||||
> | DLD (S) |  GPT-2-0.3B|21.55      | 20.65      | 22.32      |
> | CSD (SS) |  GPT-2-0.3B|**22.69**      | **22.72**     | **23.12**    |
> - **Table 4** presents newly added experiments on distilling general chat capabilities with Qwen2.5-IT and Gemma2-IT and compares them with DLD.
> || | Qwen2.5-IT (7B$\rightarrow$1.5B) | | |Gemma2-IT (9B$\rightarrow$2B) | |
> |-|:-:|:-:|:-:|:-:|:-:|:-:|
> | Benchmark| MT-Bench| MT-Bench | AlpacaEval | MT-Bench | MT-Bench| AlpacaEval |
> | Judge| GPT4| GPT4-Turbo | GPT4-Turbo | GPT4| GPT4-Turbo | GPT4-Turbo |
> | | |  |  | |  |  |
> | Teacher | 8.59 | 7.52 | 88.69 | 8.91 | 7.66 | 94.60 |
> | | |  |  | |  |  |
> | DPKD (Li et al., 2024) | 1.04 | 1.09 | 0.32 | 6.30 | 4.89 | 71.18 |
> | DistiLLM2 (Ko et al., 2025) | 7.28 | 5.75 | **70.42** | 7.81 | 6.45 | 89.91 |
> | DLD (T) | 7.25 | 5.56 | 69.80 | 5.85 | 4.45 | 31.24 |
> | DLD (S) | 7.28 | 5.74 | 67.67 | 7.58 | 6.53 | 89.84 |
> | **CSD (T, S)** | 7.42 | 5.90 | **70.42** | **7.85** | **6.55** | 89.92 |
> | **CSD (S, S)** | **7.69** | **5.95** | 69.64 | 7.77 | 6.43 | **90.05** |
>
>
> **(Comparison to temperature-scaled DLD)** Temperature scaling is typically applied by dividing both the student and teacher logits by $\tau$ before computing the loss. While this can meaningfully alter the loss landscape for softmax-based objectives such as KL divergence, it becomes ineffective for L2 logit matching, since it only results in a constant rescaling of the loss without changing the optimization behavior. Instead, we applied temperature scaling to the weighting function $w$, producing new variants of DLD. This temperature scaling can also be applied to the weighting function of CSD, so we evaluated it for both methods. As shown in **Figure 5.a**, the performance gains are more pronounced for CSD. The table below highlights the sweetspot temperature region.
> | Loss        | Avg. Rouge-L $\uparrow$ |
> |-------------|:-:|
> | DLD (S) ($\tau=1$)      | 20.39       |
> | CSD (S, S) ($\tau=1$)  | **20.65**       |
> |||
> | DLD (S) ($\tau=1.25$)      | 20.80      |
> | CSD (S, S) ($\tau=1.25$)  | **21.17**       |
> |||
> | DLD (S) ($\tau=1.5$)      | 20.43       |
> | CSD (S, S) ($\tau=1.5$)  | **21.23**       |

---

> ### Author Response · Authors · 2025-11-21
> **Official Comment by Authors for Reviewer 8hoT (3/4)**
>
> ### **Weakness 2 & Question 3 (Efficiency comparison)**
>
> *Q. Practical computational burden. Although Theorem 3 reduces complexity from O(|V|2) to O(|V|), real-world LLMs with very large vocabularies (e.g., 50k+) still require computing and normalizing all token logits per step. The paper shows that a Monte-Carlo estimator performs worse, so the efficient implementation currently relies on full-vocabulary passes, which may become a bottleneck at scale. // Scaling characterization. For vocabularies of 32k–100k and sequence lengths of 4k–8k, what are the peak memory, step time, and convergence trade-offs of CSD (analytic vs. MC) versus KL/DLD? Is there a practical heuristic for choosing analytic vs. MC estimators at different scales?*
>
> &nbsp;
>
> ### **Response to Weakness 2 & Question 3.**
>
> Autoregressive LLMs output logits over the entire vocabulary in a single forward pass and normalizing these logits across all vocabulary items is a standard tensor operation. KL divergence similarly requires softmax normalization over all vocabulary logits, so the computational structure is comparable.  (You can refer to Code 1 in Appendix C for a concise PyTorch implementation of CSD, which requires only 3–4 lines of code.)
>
> We have added GPU memory usage and wall-clock training time measurements for the full distillation pipeline, including the teacher forward pass, student forward pass, loss computation, and student backward pass. The loss function can influence overall computational cost through both the complexity of its own computation and the structure of the gradient computational graph used during the student’s backward pass. We keep all other experimental settings fixed across the five backbone models. (We also added the results in **Table 9** of Section D in the revised manuscript.)
>
>
> ||||Configuration|||
> |-|:-:|:-:|:-:|:-:|:-:|
> |**Models**|**GPT-2**|**GPT-2**|**OpenLLaMA**|**Qwen2.5-Instruct**|**Gemma2-Instruct**|
> | **Teacher**| 1.5B | 1.5B | 7B | 7B | 9B |
> | **Student**|0.1B | 0.3B | 3B | 1.5B | 2B |
> | **LoRA**   | x | x| o| o| o|
> | **Vocab Size**    |50,257| 50,257| 32,000| 151,665| 256,000|
> | **Sequence Length**| 512  | 512  | 512| 1024| 1024|
> | **Microbatch$\times$Grad_Accu.**|32 (32$\times$1)|32 (32$\times$1)|32 (32$\times$1)|32 (2$\times$16)| 32 (1$\times$32)|
> |||||||
> ||||**Peak Memory (MB) ($\downarrow$)**|||
> | **SKL**| 39,130 | 60,083  | 41,542  | 52,235| 52,197|
> | **CSD (MC)**| 30,490 | 50,657 | 35,342 | 47,502 | 48,201|
> | **CSD (Analytic)**| **28,920** | **49,085** | **34,542** | 42,766 | 44,205|
> | **DLD**| 30,490 | 50,657 | 35,342 | 45,134 | 46,203|
> | **KL**| 32,846 | 49,871 | 35,042 | **38,033** |**40,208**|
> |||||||
> ||||**Elapsed Time (Sec. / batch) ($\downarrow$)**|||
> | **SKL**|  0.803|  1.094|  4.077 |  27.88 |  43.92 |
> | **CSD (MC)**|  0.789|  1.061|  4.063|  27.78| 43.06|
> | **CSD (Analytic)**|  0.764 |  1.033 |  4.041 |  26.64| **42.12**|
> | **DLD**|  **0.758**|  1.033|  **4.035**|  26.62| 44.50|
> | **KL**|  0.770|  **1.027**|  4.044|  **25.84** | 43.34|
>
> All losses considered are computed using standard linear-time tensor operations. Therefore, changing the loss generally results in negligible differences in training speed. However, SKL requires constructing intermediate distributions for both the student and teacher, which incurs slightly higher memory consumption. For CSD (Analytic), following Algorithm 1, the backward pass is computed directly with respect to the raw student logits, while the gradient coefficient is detached and applied afterward. This effectively precomputes part of the gradient of the L2 loss, reducing the size of the computational graph involved in the student’s backward pass. Consequently, CSD (Analytic) achieves better memory and time efficiency compared to CSD (MC), which performs backpropagation through the full L2 loss output.
>
> Regarding the performance of CSD (MC) and CSD (Analytic), as shown in the modified **Figure 5.c**, the analytic approach performs slightly better, but MC still outperforms other losses like KL. Therefore, the performance drop due to MC is not substantial. However, if possible, performing analytic gradient computation is preferable in terms of cost and performance. On the other hand, the advantage of MC, as mentioned in **lines 272-277**, is its ability to extend to joint weighting space modeling or Lp loss. This makes the MC loss beneficial, as it allows for more complex loss modeling.

---

> ### Author Response · Authors · 2025-11-21
> **Official Comment by Authors for Reviewer 8hoT (4/4)**
>
> ### **Question 2 (Logit offset behaviors)**
>
> *Q. On the solution space. Since CSD allows a global additive shift on logits, how does this offset behave during training—does it converge to a stable value? Are there systematic differences across tokens? Does this flexibility introduce any extra instability compared to DLD?*
>
> &nbsp;
>
> ### **Response to Question 2.**
>
> **Figure 11** of Section D.1 in the modified manuscript plots the logit offsets of the teacher and the student across 10 consecutive tokens within a sentence, evaluated every 5 epochs starting from the initialized student. Each cluster of points with the same color corresponds to all vocabulary logit offsets $(f_T[y_t] – f_{\theta}[y_t])$ for a single token across all vocabulary items $y_t \in V$. In other words, each token has its own offset, and this token-specific offset is largely determined early in training and remains stable throughout later stages. Unlike CSD, DLD forces the solution toward a fixed offset of zero for every token, as revealed in **Figure 9.a**. Section D.1 provides a more systematic analysis of the logit offsets.
>
> ---
>
> Thank you once again for your time and insightful feedback. We would be happy to answer any additional questions you may have

---

### Official Review · Reviewer_Pedf · 2025-11-02

**Soundness:** 3
**Presentation:** 3
**Contribution:** 3
**Rating:** 8
**Confidence:** 4

**Summary:**

This paper addresses the challenge of distilling large language models. To address some of the limitations of Direct Logit Distillation, the authors propose leveraging score matching for discrete variables. They thus propose a new objective for knowledge distillation named Concrete Score Distillation. The main idea is to replace the dataset data distribution ($p_{\textrm{data}$) in the score matching objective with the teacher probability $p_T$. To address training instability, they propose using a logarithm function. They also propose an efficient approach for computing gradients. A complete experimental validation is provided for different setups: task-agnostic instruction-following distillation and task-specific distillation.

**Strengths:**

**originality**
 + The idea of using the generalized score function for LLM knowledge distillation is new and original.
 + The proposed approach is technically sound, and the claims are supported by both theoretical aspects and by the experimental part.

**quality**
 + The effectiveness of the proposed approach is validated on multiple KD settings.

**clarity**
+ The paper is well-written, with a clear formalization of the proposed approach.
The approach is also well-motivated.
+ The annexes of the paper contain important additional information, such as proofs, derivations, and additional experimental details.

**significance**
+ LLM Knowledge distillation is a crucial issue for the real-world deployment of LLMs. The proposed approach clearly advances the state of the art on LLM Knowledge distillation.

**Weaknesses:**

+ Some minor remarks on equation 5 and 6 : $p_{\textrm{data}}$ is not defined.
+ Proposition 1 and Theorem 2 assume a sufficient model capacity. This should be detailed. What is a sufficient model capacity in this particular case?
+ In Theorem 3, an assumption is that we can write the weighting function $w(y_t,x) = w_1(y_t) w_2(x)$. The authors do not justify this assumption. This rewriting of the weighting function is important in the proposed approach and in the experimental part. The paper does not explain how to build $w_1$ and $w_2$.

**Questions:**

+ A key assumption of the proposed approach is that the teacher and student models share the same vocabulary. We know that, in practice, this hypothesis is not satisfied due to the differences between the different tokenizers. How can this issue be addressed in the proposed approach?

---

> ### Author Response · Authors · 2025-11-21
> **Official Comment by Authors for Reviewer Pedf (1/2)**
>
> We sincerely appreciate **reviewer Pedf** for the deep understanding of our work and providing helpful feedback. Thank you for the reviewer’s recognition of **originality**, **quality**, **clarity**, and **significance** of this paper. We address the reviewer’s comments below.
>
> &nbsp;
>
> ---
>
> ### **Weakness 1**
>
> *Q. Some minor remarks on equation 5 and 6 : p_data is not defined.*
>
> &nbsp;
>
> ### **Response to Weakness 1.**
> We thank the reviewer for pointing this out. We have now explicitly defined $p_{data}$ as the data distribution after Equations (5) and (6). This addition can be found at **lines 150 and 170** of the revised manuscript.
>
> &nbsp;
>
> ---
>
> &nbsp;
>
> ### **Weakness 2 (Model capacity assumption)**
>
> *Q. Proposition 1 and Theorem 2 assume a sufficient model capacity. This should be detailed. What is a sufficient model capacity in this particular case?*
>
> &nbsp;
>
> ### **Response to Weakness 2.**
>
> This assumption is commonly used to discuss optimality of objective function and often appears in prior work without explicit explanation. For example, recent works such as [B1] (Theorem 1), [B2] (Proposition 3.2), and [B3] (Theorem 2) make the same assumption. However, as you pointed out, clearly stating the assumption is important to ensure that readers understand it correctly.
>
> The sufficient model capacity refers to the assumption that our model distribution $q_{\theta}$ can represent an arbitrary conditional distribution $q_{\theta}(\cdot|c,y_{<t})$ for any context $c$ and prefix $y_{<t}$. This assumption means that the parameter space $\theta \in \Theta$ is sufficiently large to include at least one parameter that minimizes the CSD objective $L_{CSD}$ to zero in Proposition 1 (the same applies to Theorem 2). We have refined the description of sufficient model capacity in the manuscript by expressing it more explicitly as the limit $|\Theta| \rightarrow \infty$ (Please refer to the **Proposition 1** and **Theorem 2** of the modified manuscript).
>
> Although this assumption differs from the practical setting in which the parameter space is finite, it is important because it clarifies the target of our learning objective and connects to the fact that performance improves asymptotically as model capacity increases (See **Table 2** for the comparison between the GPT-2-0.1B and GPT-2-0.3B students).
>
>
> [B1] [AISTATS 2025] f-PO: Generalizing Preference Optimization with f-divergence Minimization
>
> [B2] [ICML 2024] Discrete Diffusion Modeling by Estimating the Ratios of the Data Distribution
>
> [B3] [NeurIPS 2025] Preference Optimization by Estimating the Ratio of the Data Distribution

---

> ### Author Response · Authors · 2025-11-21
> **Official Comment by Authors for Reviewer Pedf (2/2)**
>
> ### **Weakness 3 (Weighting function decomposition)**
>
> *Q. In Theorem 3, an assumption is that we can write the weighting function $w(y_t, x)=w_1(y_t) w_2(x)$. The authors do not justify this assumption. This rewriting of the weighting function is important in the proposed approach and in the experimental part. The paper does not explain how to build w1 and w2 .*
>
> &nbsp;
>
> ### **Response to Weakness 3.**
>
> Decomposing the joint weighting function into two independent functions assumption that is required to derive the analytic gradient in linear time. In fact, defining a joint weighting function over the vocabulary, whose size would be $|V|^2$, is generally difficult in practice, which makes it challenging to propose meaningful joint weighting instances. We found that important behaviors such as mode-seeking and mode-covering (as shown in **Figure 3**) can still be effectively captured within the independent design space. In addition, Monte Carlo estimation allows modeling of the full joint weighting space without this assumption (Please refer to **Algorithm 2** of Section C). We have also added the relevant discussion in **lines 272-277** of the revised manuscript.
>
> &nbsp;
>
> ---
>
> &nbsp;
>
> ### **Question 1 (Applicability to cross tokenizer setting)**
>
> *Q. A key assumption of the proposed approach is that the teacher and student models share the same vocabulary. We know that, in practice, this hypothesis is not satisfied due to the differences between the different tokenizers. How can this issue be addressed in the proposed approach?*
>
> &nbsp;
>
> ### **Response to Question 1.**
> Thank you for your insightful question. LLM distillation can generally be divided into two orthogonal directions: 1) loss-function research and 2) cross-tokenizer research. Prior work on loss functions has typically assumed a shared tokenizer, and many commercial LLM families (such as Qwen, LLaMA, Gemma, and GPT) provide multiple model sizes that share the same vocabulary. Consequently, shared-tokenizer distillation is commonly reported in the technical reports of companies that deploy LLM services. As mentioned in **Line 105**, our experiments also follow this setting, consistent with prior work (Li et al., 2020; Ko et al., 2024).
>
> That said, we agree that cross-tokenizer distillation is more practical and important for real-world applications. Accordingly, we also discuss how CSD can be applied in cross-tokenizer settings. A recent study [B4] shows that distillation can be performed by matching the likelihood of observed chunks that are shared between the teacher’s and student’s tokenizers. For example, when the input $(c, y_{<t})$ is “Say”, the student tokenizer may split the next shared chunk “GoodBye” into {“Good”, “Bye”}, whereas the teacher tokenizer may represent it as {“Goodbye”}. For such observed chunks (e.g., “GoodBye”), the loss in [B4] is formulated using a binarized f-divergence, a modified f-divergence designed to address the intractability of matching over all possible next-token states when tokenizers differ:
> $$D_{f} ({q_{\theta}(Good, Bye | Say), 1- q_{\theta}(Good, Bye | Say) } || {p_{T}(GoodBye | Say)), 1- p_{T}(GoodBye | Say))}$$
> In CSD as well, matching over all possible states $(x, y_t)$ is intractable under different tokenizers. However, unlike the f-divergence approach, which requires a binarized approximation, CSD instead defines $w_1$ and $w_2$ as the observation probabilities of shared chunks. By approximating the objective over $(x, y_t)$ via Monte Carlo estimation, we obtain the following formulation:
> $$ L_{CSD}= \frac{1}{2}\sum_{y_t}\sum_{x}w(y_t, x) \left[ \log{\frac{q_{\theta}(x | c, y_{<t})}{q_{\theta}(y_t | c, y_{<t})}} -  \log{\frac{p_{T}(x | c, y_{<t})}{p_{T}(y_t | c, y_{<t})}}  \right]^2  = \frac{1}{2}E_{w(y_t, x) } \left[ \log{\frac{q_{\theta}(x | c, y_{<t})}{q_{\theta}(y_t | c, y_{<t})}} -  \log{\frac{p_{T}(x | c, y_{<t})}{p_{T}(y_t | c, y_{<t})}}  \right]^2$$
> Similarly, if we consider that for the input $(c, y_{<t})$=“Say”, the chunks “GoodBye” and “Hello” are observed from the distribution $w(\cdot,\cdot)$, then the instance-level loss becomes:
> $$ l_{CSD}= \left[ \log{\frac{q_{\theta}(Good, Bye | Say)}{q_{\theta}(He, llo | Say)}} -  \log{\frac{p_{T}(GoodBye | Say)}{p_{T}(Hello | Say)}}  \right]^2$$
> Overall, while our main experiments follow the shared-tokenizer setting consistent with prior literature, CSD naturally extends to cross-tokenizer scenarios through likelihood-matching over shared observed chunks, as demonstrated above.
>
>
> [B4] [NeurIPS 2025] Universal Cross-Tokenizer Distillation via Approximate Likelihood Matching
>
> ---
>
> Thank you once again for your time and valuable feedback. Please let us know if you have any additional questions.

---

> > ### Comment · Reviewer_Pedf · 2025-11-26
> >
> > I appreciate the detailed rebuttals from the authors.  They carefully answer, with details, to my main concerns. It confirms that the proposed paper is a strong contribution supported by both experimental validation and a theoretical perspective. I maintain my grade.

---

> > > ### Author Response · Authors · 2025-11-29
> > >
> > > We sincerely appreciate reviewer **Pedf** for taking the time to read all our responses and the updated manuscript. We are glad that your concerns regarding the assumptions in our paper and the justification for extending the method to cross-tokenizer settings have been satisfactorily addressed. We also sincerely thank you for again highlighting the strong contribution of our work from both theoretical and empirical perspectives.

---

### Official Review · Reviewer_TcP2 · 2025-11-02

**Soundness:** 3
**Presentation:** 3
**Contribution:** 3
**Rating:** 6
**Confidence:** 3

**Summary:**

The paper proposes Concrete Score Distillation (CSD), a discrete score-matching objective for LLM distillation that avoids softmax-induced smoothing and is invariant to constant logit shifts, which are two limitations of common KD and direct-logit losses. It resolves the quadratic cost of pairwise score matching with an analytic, linear-time gradient and provides formal guarantees. Experiments on instruction following and task-specific settings (e.g., summarization, translation) show CSD consistently outperforms recent KD objectives and delivers favorable fidelity-diversity trade-offs.

**Strengths:**

- The paper proves consistency and that CSD’s optimum set strictly contains DLD’s, leveraging invariance to constant logit shifts

- It derives an Algorithm 1 for keeping compute/memory in line with standard KD

- The method consistently surpasses strong KD baselines across tasks and yields complementary gains when combined with on-policy techniques, suggesting good scalability.

**Weaknesses:**

- The paper claims linear-time gradients, but lacks systematic reporting of GPU memory, tokens/sec, and wall-clock per step versus KL/SKL/DLD across vocabulary sizes (e.g., 32k to 128k). Adding end-to-end cost curves and scaling ablations would strengthen adoption

- While the paper shows that weights modulate mode-seeking/covering, default choices, schedules, or adaptive schemes are not well-justified; more guidance (or learning the weights) would aid practitioners.

- Since CSD replaces softmax normalization with centered logits, its impact on probability calibration warrants measurement. (Related to shift-invariance.)

**Questions:**

Can you report throughput, peak memory, and step time for CSD vs. KL/SKL/DLD on matched setups, plus scaling with vocab size? (This directly tests the linear-time gradient claim and Algorithm 1 practicality.)

Can you provide defaults, schedules, or an adaptive/learned weighting (e.g., driven by confidence or gradient variance)?

---

> ### Author Response · Authors · 2025-11-21
> **Official Comment by Authors for Reviewer TcP2 (1/3)**
>
> We sincerely appreciate **reviewer TcP2** for the valuable feedback and for recognizing both the **theoretical contributions** and **empirical scalability** of our work. We address the reviewer’s comments below.
>
> &nbsp;
>
> ---
>
> ### **Weakness 1 & Question 1. (Efficiency reporting)**
>
> *Q. The paper claims linear-time gradients, but lacks systematic reporting of GPU memory, tokens/sec, and wall-clock per step versus KL/SKL/DLD across vocabulary sizes (e.g., 32k to 128k). Adding end-to-end cost curves and scaling ablations would strengthen adoption. // Can you report throughput, peak memory, and step time for CSD vs. KL/SKL/DLD on matched setups, plus scaling with vocab size? (This directly tests the linear-time gradient claim and Algorithm 1 practicality.)*
>
> &nbsp;
>
> ### **Response to Weakness 1 & Question 1.**
>
> Thank you for suggesting this important analysis. We have added measurements of GPU memory usage and wall-clock training time for the distillation procedure. (Because tokens/sec was measured under the same sequence length and batch size, it can be directly inferred from sec/batch; therefore, we omit it.) These measurements include the teacher forward pass, student forward pass, loss computation, and the student backward pass. The loss function can influence the overall computational cost by affecting both the complexity of the loss computation itself and the structure of the gradient computation graph used during the student’s backward pass. We keep all other experimental conditions fixed across the five backbone configurations. (We also added the results in **Table 9** of Section D in the revised manuscript.)
>
>
> ||||Configuration|||
> |-|:-:|:-:|:-:|:-:|:-:|
> |**Models**|**GPT-2**|**GPT-2**|**OpenLLaMA**|**Qwen2.5-Instruct**|**Gemma2-Instruct**|
> | **Teacher**| 1.5B | 1.5B | 7B | 7B | 9B |
> | **Student**|0.1B | 0.3B | 3B | 1.5B | 2B |
> | **LoRA**   | x | x| o| o| o|
> | **Vocab Size**    |50,257| 50,257| 32,000| 151,665| 256,000|
> | **Sequence Length**| 512  | 512  | 512| 1024| 1024|
> | **Microbatch$\times$Grad_Accu.**|32 (32$\times$1)|32 (32$\times$1)|32 (32$\times$1)|32 (2$\times$16)| 32 (1$\times$32)|
> |||||||
> ||||**Peak Memory (MB) ($\downarrow$)**|||
> | **SKL**| 39,130 | 60,083  | 41,542  | 52,235| 52,197|
> | **CSD (MC)**| 30,490 | 50,657 | 35,342 | 47,502 | 48,201|
> | **CSD (Analytic)**| **28,920** | **49,085** | **34,542** | 42,766 | 44,205|
> | **DLD**| 30,490 | 50,657 | 35,342 | 45,134 | 46,203|
> | **KL**| 32,846 | 49,871 | 35,042 | **38,033** |**40,208**|
> |||||||
> ||||**Elapsed Time (Sec. / batch) ($\downarrow$)**|||
> | **SKL**|  0.803|  1.094|  4.077 |  27.88 |  43.92 |
> | **CSD (MC)**|  0.789|  1.061|  4.063|  27.78| 43.06|
> | **CSD (Analytic)**|  0.764 |  1.033 |  4.041 |  26.64| **42.12**|
> | **DLD**|  **0.758**|  1.033|  **4.035**|  26.62| 44.50|
> | **KL**|  0.770|  **1.027**|  4.044|  **25.84** | 43.34|
>
>
> Both the student and teacher produce logits over the full vocabulary simultaneously, and all losses in the above table are computed using standard linear-time tensor operations. Therefore, changing the loss function results in a less significant difference in training speed. However, SKL requires representing intermediate distributions between the student and teacher, which leads to slightly higher memory usage.
>
> For CSD (Analytic), following **Algorithm 1** of the manuscript, the backward pass is computed with respect to the raw student logits, and the gradient coefficient is detached and applied afterward. This effectively precomputes part of the L2 loss gradient, which makes the computational graph for the student’s backward pass smaller. As a result, CSD (Analytic) achieves better memory and time efficiency compared to CSD (MC), which performs the backward pass on the full L2 loss output, and also compared to DLD.

---

> ### Author Response · Authors · 2025-11-21
> **Official Comment by Authors for Reviewer TcP2 (2/3)**
>
> ### **Weakness 2 & Question 2. (Adaptive weighting)**
>
> *Q. While the paper shows that weights modulate mode-seeking/covering, default choices, schedules, or adaptive schemes are not well-justified; more guidance (or learning the weights) would aid practitioners // Can you provide defaults, schedules, or an adaptive/learned weighting (e.g., driven by confidence or gradient variance)?*
>
> &nbsp;
>
> ### **Response to Weakness 2 & Question 2.**
>
> Diversity and fidelity form an inherent trade-off in generative modeling, and the preferred balance often depends on the task or the user’s goals. If strong fidelity is desired, we recommend using CSD (ss), whereas CSD (ts) is more suitable when strong diversity is preferred. However, we agree that providing a well-balanced default option is important. Because an adaptive loss can better reconcile this trade-off, we conducted additional experiments on the adaptive loss as you suggested.
>
> We provide the results of two naïve scheduling and one confidence-based adaptive loss that interpolates CSD (ss) and CSD (ts) by defining $w_1$ as an interpolation of $p_s$ and of $p_T$ using $\alpha$. We found that the following geometric interpolation performs better than linear interpolation in balancing the fidelity–diversity trade-off:
> $$ w_1(x) \propto p_s(x)^{\alpha}p_T(x)^{1-\alpha}, \quad w_2(x) = p_s(x) $$
> - 1: CSD (ts $\rightarrow$ss): increases $\alpha$ from 0 to 1 during training iterations.
> - 2: CSD (ss $\rightarrow$ts): decrease $\alpha$ from 1 to 0 during training iterations.
> - 3: CSD (Entropy): $\alpha = clip(\frac{ H(p_s(x)) - H(p_T(x))}{H(p_s(x))}, 0, 1)$
>
> CSD (ts $\rightarrow$ss), CSD (ss $\rightarrow$ts), and CSD (Entropy) combine the strengths of both CSD (ss) and CSD (ts), and therefore achieve better performance at intermediate trade-off points as shown in **Figure 12 (Appendix D.2)** of the modified manuscript. Because the learning rate is high in the early stages and decreases over time, the loss used at the beginning of training tends to have a stronger influence on the final trade-off position. For example, CSD (ts $\rightarrow$ss) behaves similarly to CSD (ts) because its early-stage loss is closer to CSD (ts).
>
> Among these variants, we propose CSD (Entropy) as the default choice. Unlike iteration-based scheduling, CSD (Entropy) adaptively sets $\alpha$ at each token every step. Early in training, $H(p_s(x))$ is typically larger than $H(p_T(x))$, so $\alpha$ becomes close to 1, making the loss similar to CSD (ss). Since $p_s$ is more diverse than $p_T$ at the early stage, the CSD (ss) weighting provides richer feedback over a larger set of vocabulary indices compared to CSD (ts).
>
> In the later stages of training, $p_s$ ideally becomes closer to $p_T$, making $\alpha$ close to 0. In a well-trained situation, CSD (ss) and CSD (ts) will show similar behavior, so the exact value of $\alpha$ becomes less important. However, when training does not progress well and $p_s$ becomes overconfident without matching $p_T$, focusing solely on CSD (ss) weighting is undesirable. In such cases, stronger teacher guidance from CSD (ts) is needed, which is why CSD (Entropy) is designed as above. With this design, the model showed more balanced performance compared to CSD (ts $\rightarrow$ss) and CSD (ss $\rightarrow$ts). We have also included this analysis in Appendix D.2 of the modified manuscript. In particular, CSD (Entropy) showed improvements in both fidelity (Rouge-L) and diversity (Self-BLEU) compared to CSD (us), which previously represented the balance point between diversity and fidelity, as shown below.
>
> | | Rouge-L ($\uparrow$)|Self-BLEU ($\downarrow$) |
> |----------|-|-|
> |   CSD (us)      |    19.88   | 0.31  |
> |  CSD (Entropy)     |    **19.95**   |**0.29** |

---

> ### Author Response · Authors · 2025-11-21
> **Official Comment by Authors for Reviewer TcP2 (3/3)**
>
> ### **Weakness 3. (Probability Calibration)**
>
> *Q. Since CSD replaces softmax normalization with centered logits, its impact on probability calibration warrants measurement. (Related to shift-invariance.)*
>
> &nbsp;
>
> ### **Response to Weakness 3.**
>
> First, as proven in **Proposition 1**, the CSD loss attains its optimum when the student matches the teacher’s categorical distribution for every input context in theory. However, we agree that probability calibration is an important aspect of model behavior and should be empirically assessed as well. Therefore, we refer to the Expected Calibration Error (ECE) formulation used in  [A1], which defines ECE under a stochastic student LLM setting, and we adapt it to our case by treating the teacher probabilities as the target distribution:
>
> $$ ECE (p_T, q_{\theta}) = E_{(\mathbf{c},\mathbf{y}) \sim D}\left[\sum_{y_t}  |p_T(y_t|\mathbf{c},\mathbf{y}_{<t}) - q _{\theta} (y_t|\mathbf{c},\mathbf{y} _{<t}) |  q _{\theta} (y_t|\mathbf{c},\mathbf{y} _{<t})\right] $$
>
> The results are shown in **Figure 5.b** of the modified manuscript. In all cases, the ECE decreases over training iterations. This confirms that replacing softmax normalization with centered logits does not harm the model’s calibration behavior in practice.  Notably, CSD (ss), which has a mode-seeking tendency, is more overconfident and therefore achieves a relatively smaller reduction in ECE. In contrast, CSD (ts), which exhibits a mode-covering behavior, reduces ECE more effectively. This is expected, since overconfidence generally leads to worse ECE. For metrics other than ECE, such as KL or MSE, please refer to **Figure 10** in Appendix D. Below is a comparison of the probability calibration results between CSD (ts) and the representative baselines of DLD at the last epoch.
>
> | Loss | ECE $\downarrow$ |KL $\downarrow$ |MSE $\downarrow$ |
> |----------|--|--|--|
> |   DLD (s)      |    0.23     | 3.29|0.59|
> |   CSD (ts)      |   **0.10**    | **0.44** | **0.16**|
>
>
>
> [A1] [ICLR 2024] Beyond Reverse KL: Generalizing Direct Preference Optimization with Diverse Divergence Constraints
>
> ---
>
> Thank you again for your time and thoughtful feedback. Please feel free to ask if you have any further questions.

---

### Author Response · Authors · 2025-11-21
**Global Reply to All Reviewers**

# **Revision Summary**

We sincerely thank all the reviewers for their valuable feedback. We have made our best efforts to incorporate all questions, suggestions, and comments into the revised version of our paper. All modifications in the manuscript are highlighted in blue. For newly added rows in the tables, only the method names are marked in blue. The captions of newly added figures are marked in blue. The text describing the added or modified content has also been updated accordingly.

----

### **Revisions in the Main Paper**

-	**Table 2, Table 3**: Added DLD baselines.

-	**Table 4 (Section 4.3)**: New experiments using Qwen2.5-7B-IT and Gemma2-9B-IT for general chat capabilities

-	**Table 5, Figure 5.a**: Added DLD-variant baselines

-	**Figure 5.b**: Calibration error ablation results

-	**Figure 5.c**: Convergence comparison results
----

### **Revisions in the Appendix**

- **Section A.4**: Mathematical formulation of DLD variants

- **Section A.5**: Mathematical formulation of the weighted divergence objective

- **Section C.3**: Experimental details for the Qwen2.5-7B-IT and Gemma2-9B-IT experiments

- **Algorithm 2**: Algorithms for Monte Carlo estimation

- **Section D.1**, **Figure 9**, **Table 6**, **Figure 11**: Analysis of logit offsets

- **Figure 10**: Probability calibration error evaluated with additional metrics

- **Figure 12 (Section D.2)**: Adaptive weighting loss and related experiments

- **Figure 13 (Section D.3)**: Analysis on gradient coefficient diversity

- **Table 8**: Empirical evaluation of weighting strategies for divergence-based objectives
- **Table 9**: Comprehensive evaluation of training speed and memory usage

- **Table 10**: Added DLD baselines for task-specific experiments

----

### **Highlighting new experimental results**

We would like to emphasize once again that **Table 4** in the modified manuscript presents newly added experiments evaluating general chat capability using recent, stronger models.

|| | Qwen2.5-IT (7B$\rightarrow$1.5B) | | |Gemma2-IT (9B$\rightarrow$2B) | |
|-|:-:|:-:|:-:|:-:|:-:|:-:|
| Benchmark| MT-Bench| MT-Bench | AlpacaEval | MT-Bench | MT-Bench| AlpacaEval |
| Judge| GPT4| GPT4-Turbo | GPT4-Turbo | GPT4| GPT4-Turbo | GPT4-Turbo |
| | |  |  | |  |  |
| Teacher | 8.59 | 7.52 | 88.69 | 8.91 | 7.66 | 94.60 |
| | |  |  | |  |  |
| DPKD (Li et al., 2024) | 1.04 | 1.09 | 0.32 | 6.30 | 4.89 | 71.18 |
| DistiLLM2 (Ko et al., 2025) | 7.28 | 5.75 | **70.42** | 7.81 | 6.45 | 89.91 |
| DLD (T) | 7.25 | 5.56 | 69.80 | 5.85 | 4.45 | 31.24 |
| DLD (S) | 7.28 | 5.74 | 67.67 | 7.58 | 6.53 | 89.84 |
| **CSD (T, S) (Ours)** | 7.42 | 5.90 | **70.42** | **7.85** | **6.55** | 89.92 |
| **CSD (S, S) (Ours)** | **7.69** | **5.95** | 69.64 | 7.77 | 6.43 | **90.05** |

----

### **11/29 Updates**
- Fixing minor typos.
- **Table 15**: Added generated results on DLD baseline.

---

### Meta-Review · Area_Chair_gKA2 · 2026-01-07

**Summary:**

This paper introduces Concrete Score Distillation (CSD), a distillation objective for autoregressive language models based on discrete score matching, addressing information loss in softmax-based KD and the lack of shift invariance in direct logit distillation. Reviewers find the method well-motivated and technically solid, with clear theoretical guarantees and consistent empirical gains across multiple distillation settings. The rebuttal and revision strengthened the paper by addressing concerns on baselines, efficiency, weighting design, and calibration.

**Reviewer Concerns:**

**Concerns adequately addressed**
- Efficiency and scalability. Added runtime and memory benchmarks show that analytic CSD is practically comparable to, and sometimes more efficient than, KL and DLD objectives.
- Strength of DLD baselines. The authors include stronger DLD variants and clarify their relationship to CSD, alleviating concerns that gains stem from weak baselines.
- Weighting strategy. The revision provides clearer guidance via scheduled and adaptive weighting schemes, improving robustness and usability.
- Probability calibration. Additional calibration results indicate that CSD does not degrade, and can improve upon, DLD calibration.
- Theoretical clarity. Minor issues in notation and assumptions were clarified without affecting the core claims.

**Remaining considerations**
- Scalability to extremely long contexts and very large vocabularies may benefit from further approximation strategies, but this does not undermine the current results.
- Cross-tokenizer distillation is not empirically validated, though this limitation is acknowledged and within scope.

**Reviewer Scores:**

- Reviewer TcP2: Likely unchanged (6)
- Reviewer Pedf: Unchanged (8)
- Reviewer 8hoT: Unchanged (4)
- Reviewer 5noZ: Likely improved (from 4 to around 6)

---

### Decision · Program_Chairs · 2026-01-26

Accept (Poster)